# Tissue-selective COPII modulator SEC16B aggravates cardiovascular disease by promoting lipid export

Xiao Wang 1,2,10✉, Yating Hu[1,2,10], Lu Liu[1,2,10], Runze Huang[2], Yawei Wang[3], Bing Liu[2,4], Yifei Zhao[1,2], Yuangang Zhu[2], Jia Lv[5], Liang Liu 3, Huimin Wang[3], Lingzhi Wu[2,6], Xinxuan Xu[1,2], Yaxin Li 7, Guanlin Wang 8 & Xiao-Wei Chen 1,2,3,9✉

## Abstract

The biogenesis and transport of lipoproteins are essential for systemic homeostasis and cardiometabolic health, yet how the secretory pathway acquires specialization to support high-capacity lipoprotein export remains unclear. Here, we report SEC16B as a tissue-selective modulator of the COPII machinery, critical for the efficient secretion of APOB-containing lipoproteins. Integrative bioinformatic analyses identify that *SEC16B* co-emerges with core genes involved in lipoprotein biogenesis. Functional studies, coupled with AI-driven prediction, reveal that SEC16B acts as a molecular brake to fine-tune COPII condensation for lipoprotein export. Mining of UK biobank data links SEC16B to metabolic traits in humans and suggests HNF4A-dependent regulation of SEC16B expression. Hepatic deletion of SEC16B in mice markedly reduces circulating APOB, triglycerides and cholesterol, while conferring robust protection against atherosclerosis and cardiac dysfunction and maintaining liver health. Collectively, these findings position SEC16B as a specialized modulator of lipoprotein export via the general secretory (SEC) pathway in the liver, suggesting potential therapeutic avenues for combating cardiometabolic diseases.

**Keywords** Lipid Metabolism; Lipoprotein Secretion; COPII; *SEC16B*; Cardio-metabolic Health
**Subject Categories** Cardiovascular System; Membranes & Trafficking; Metabolism

## Introduction

Atherosclerotic cardiovascular diseases (ASCVDs) and related metabolic disorders remain the leading cause of morbidity and mortality worldwide. A central driver of ASCVD pathogenesis is dyslipidemia, or elevated levels of plasma lipids, affecting over 2 billion people globally (Goldstein and Brown, 2015; Yasin et al, 2019). Bulk lipids, including cholesterol and triglycerides (TGs), are transported into the circulation by APOB-containing lipoproteins (Ference et al, 2020; Zhang et al, 2025). While medications such as statins and PCSK9 inhibitors have effectively lowered atherogenic cholesterol levels, approaches to reduce high levels of both TG and cholesterol, in so-called mixed dyslipidemia, remain limited (Ray et al, 2025; Rifai & Ballantyne, 2021). Of note, plasma APOB levels represent a stronger predictor of ASCVD risks than lipid levels per se (Behbodikhah et al, 2021), highlighting the need to elucidate the itinerary of the lipid carrier and its regulation, particularly with the aid of mega-data from consortium studies and recent advances of AI-driven discoveries (Abramson et al, 2024; Gao et al, 2024; Jumper et al, 2021).

APOB, along with microsomal triglyceride transfer protein (MTTP), belongs to a specialized family of lipid-ferrying proteins that may evolve from a common ancestor, coinciding with the increasing demand for high-capacity lipid transport in higher organisms (Smolenaars et al, 2007). In mammals, the expression and deployment of APOB is strikingly tissue-restricted, with the liver producing very-low-density lipoproteins (VLDLs) to distribute endogenously synthesized lipids, and the intestine generating chylomicrons to deliver dietary lipids (Feingold, 2022; Fisher and Ginsberg, 2002). The biogenesis of APOB-containing lipoproteins within the ER lumen requires highly coordinated steps, including MTTP-mediated co-translational lipidation to stabilize nascent APOB, a prerequisite for the subsequent assembly into lipid-loaded lipoproteins (Hussain et al, 2012; Read et al, 2000). Despite reliance on the universal ER-to-Golgi secretory pathway (Fisher and Ginsberg, 2002; Zanetti et al, 2011), APOB-containing lipoproteins engage a specialized export program, orchestrated by the transmembrane cargo receptor SURF4 (Ginsberg, 2021; Tang et al, 2022; Tao et al, 2023; Wang et al, 2021a). SURF4 selectively bridges luminal APOB-containing lipoproteins to the cytosolic COPII transport machinery, enabling efficient ER exit to embark on the secretory itinerary (Tang et al, 2022; Tao et al, 2023; Wang et al, 2021a). Notably, genetic inactivation of hepatic SURF4 leads to grossly healthy mice with near-complete depletion of plasma lipids under fasting conditions (Shen et al, 2022; Tang et al, 2022; Wang et al, 2021a), revealing a selective transport program coping with

[1]State Key Laboratory of Membrane Biology, Peking University, Beijing, China. [2]Institute of Molecular Medicine, College of Future Technology, Peking University, Beijing, China. [3]Center for Life Sciences, Peking University, Beijing, China. [4]Chongqing Institute for Brain and Intelligence, Chongqing, China. [5]Department of Orthopedics, The Fourth Medical Center of Chinese PLA General Hospital, Beijing, China. [6]Institute of Advanced Clinical Medicine, Peking University, Beijing, China. [7]Guangzhou National Laboratory, Guangzhou, China. [8]Institute of Metabolism and Integrative Biology, Fudan University, Shanghai, China. [9]Beijing Advanced Center of RNA Biology (BEACON), Peking University, Beijing, China. [10]These authors contributed equally: Xiao Wang, Yating Hu, Lu Liu. ✉E-mail: Xiao_WANG@pku.edu.cn; xiaowei_chen@pku.edu.cn

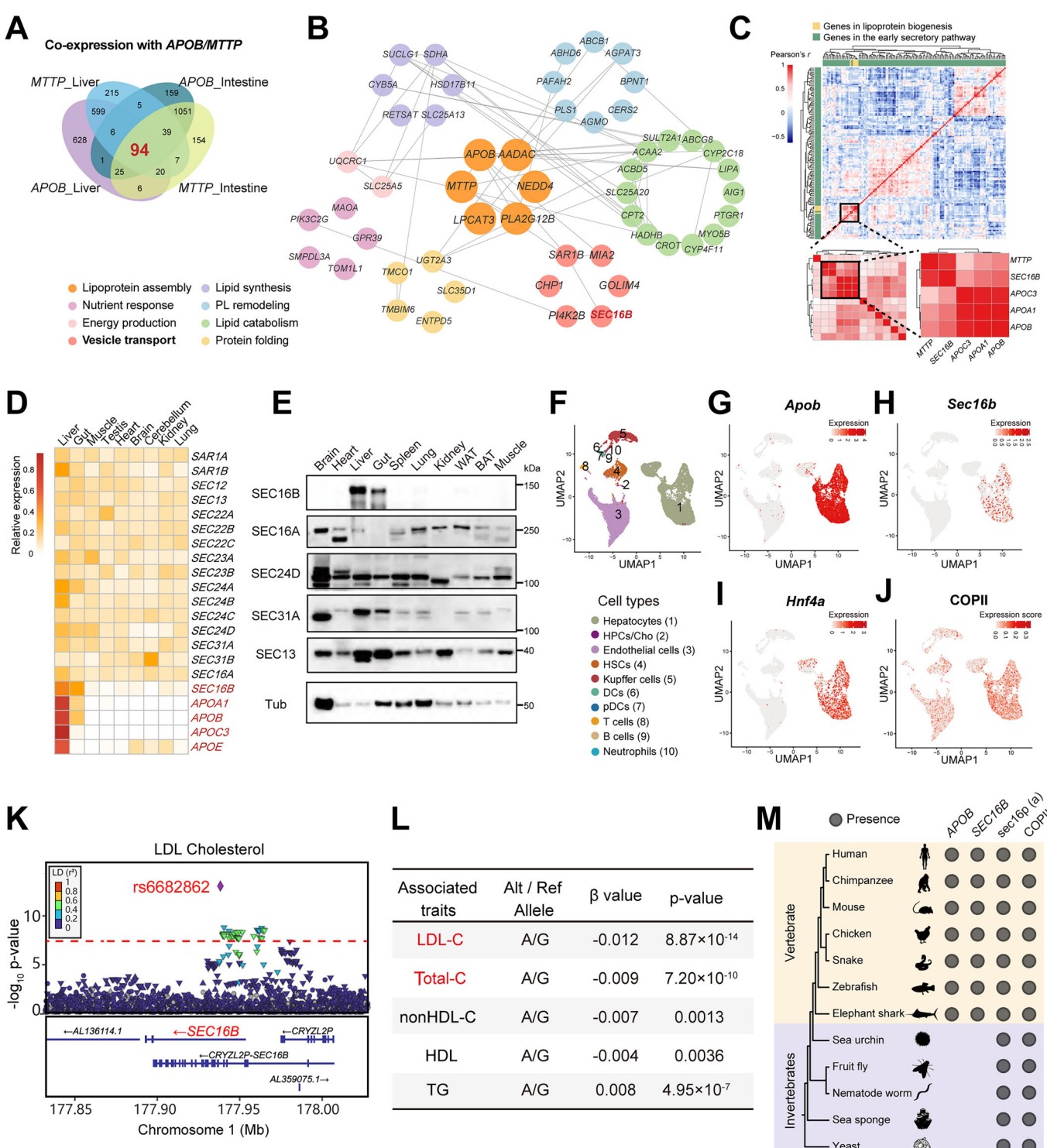

the general COPII machinery to operate high-capacity lipid transport (Ginsberg, 2021; Tang and Ginsburg, 2023; Wang and Chen, 2023; Zhang et al, 2025).

The assembly of COPII-coated vesicles begins with the activation of the small GTPase SAR1, which sequentially recruits the inner-coat heterodimer SEC23/24, followed by the outer-coat heterotetramer SEC31/13 onto the ER surface (Barlowe and Miller,

2013; Miller and Schekman, 2013). Recent studies reveal that the COPII employs self-constrained condensation to balance dynamics and assembly, thereby maximizing the efficiency for lipoprotein export (Qiu et al, 2024; Wang et al, 2023a). Specifically, the inner COPII coat SEC23/24 initiates IDR-dependent condensation to promote complex assembly. Subsequent recruitment of the outer coat SEC31/13 serves to restrain excessive condensation, thereby

◄ **Figure 1.   Emergence of *SEC16B* of as a liver/gut-specific, lipid-related *Sec* gene.**

(A) Venn diagram of co-expressing genes of *APOB* and *MTTP* in the liver and intestine, defined by Pearson correlation of RNA expression above 0.6. (B) The functional clustering of 94 prioritized candidates in (A) based on GO annotation. Nodes represent genes, and edges indicate functional interactions derived from the STRING database. (C) Heatmap of pairwise correlation matrix of mRNA tissue distribution patterns for ER-to-Golgi transport and lipoprotein-related genes. Lower: a zoom-in cluster containing genes encoding apolipoproteins and *MTTP*. The color scale represents Pearson correlation coefficients with positive in red and negative in blue. (D) Heatmap of RNA expression levels of genes among the *SEC* and *APO* genes across different human tissues. (E) Immunoblot (IB) analysis of SEC protein levels across different mouse tissues. WAT, white adipose tissue; BAT, brown adipose tissue. Representative of three independent experiments is shown. (F) UMAP visualization of mouse liver cell clusters based on published annotations. HPCs/Cho hepatic progenitor cells/cholangiocyte, HSCs hepatic stellate cells, DCs dendritic cells, pDCs plasmacytoid DCs. Data are obtained from GEO GSE166504. (G–J) UMAP visualization showing the expression of *Apob* (G), *Sec16b* (H), *Hnf4a* (I), and the AUCell score (J) of COPII genes. (K) Regional plot of *SEC16B* associated with plasma LDL-cholesterol levels in humans. Red line indicates the threshold of genome-wide significance ($P = 5 \times 10^{-8}$). Data are obtained from the Global Lipids Genetics Consortium (GLGC). (L) Summary of the GLGC genome-wide association data between the leading SNP rs6682862 and plasma lipids. The *P* and β values are obtained from GLGC. See "Methods" section "Association analysis of genetic variants in SEC16B with lipid traits" for details. LDL-C LDL-cholesterol, Total-C total cholesterol, nonHDL-C nonHDL-cholesterol, TG triglycerides. (M) Evolutionary co-occurrence of *SEC16B* with *APOB*. Source data are available online for this figure.

maintaining optimal dynamics for efficient transport (Wang et al, 2023a). Notably, divalent manganese acts as a signal messenger to promote SEC23/24 condensation, adding an additional layer of control on lipoprotein secretion (Wang et al, 2023a; Wang et al, 2023b). Intriguingly, the dual-specificity kinase DYRK3, known for dissolving various biomolecular condensates, also regulates COPII puncta on the ER surface, via directly phosphorylating SEC16A, further highlighting the tunable nature of COPII condensation (Gallo et al, 2023). Moreover, these findings raise the intriguing possibility of a potential "molecular brake" that prevents the over-progression of COPII condensation in response to cellular lipid demand.

Here, we report that, using integrative bioinformatic analyses, we identified SEC16B as a tissue-selective *Sec* gene regulating lipoprotein secretion and lipid homeostasis. Mechanistically, SEC16B restrains COPII condensation at ER exit sites, fine-tuning the general transport machinery to support efficient lipoprotein export. A lipid-associated SNP (rs6682862) in the *SEC16B* locus potentially links its gene expression to HNF4α, establishing a genetic basis for selective expression. Pathologically, SEC16B is upregulated in obesity and hyperlipidemia. Hepatic inactivation of SEC16B markedly counters mixed hyperlipidemia, prevents atherosclerosis, and preserves coronary flow reserve, with remarkably little hepatotoxicity. Together, our findings uncover SEC16B as a selective, human-relevant regulator of lipoprotein transport and highlight its therapeutic potential for treating dyslipidemia and ASCVD.

## Results

### Emergence of *SEC16B* as a liver/gut-specific, lipid-related *Sec* gene

The unique physiological demands of lipoprotein secretion led us to hypothesize that specialized regulatory factors have evolved in the liver and small intestine to cooperate with APOB-containing lipoproteins, supporting their efficient export. To test this hypothesis, we conducted systematic bioinformatic analysis, integrating genome-wide correlation, tissue-specific distribution, and single-cell transcriptomic analysis, to identify potential regulators selectively involved in lipid transport.

We first ranked all human protein-coding genes based on their expression correlation to either *APOB* and *MTTP*, two lipid-

ferrying ER proteins, in either the liver or small intestine, using bulk RNA-seq data from the GTEx consortium. Genes co-expressed with both *APOB* and *MTTP* (defined by Pearson $r > 0.6$) in both tissues were retained, yielding 94 candidates for further analysis (Fig. 1A). Importantly, *SMLR1* and *ERICH4* were identified as being significantly co-expressed with both *MTTP* and *APOB* in the liver or the gut, consistent with prior co-expression analyses (van Zwol et al, 2023; Visser et al, 2024). Based on Gene Ontology (GO) annotations, 52 genes of this prioritized set were assigned to 8 modules with confirmed roles in processes including lipoprotein assembly, lipid metabolism, and vesicle-mediated transport (Fig. 1B). Notably, two genes involved in COPII-mediated ER export were enriched in this group: *SAR1B*, encoding a known regulator of lipoprotein secretion in mice and humans (Tang et al, 2024; Wang et al, 2021a), and *SEC16B* (Fig. EV1A–D), encoding a much less understood paralog of the COPII scaffold SEC16A (Bhattacharyya and Glick, 2007; Budnik et al, 2011; Shi et al, 2023; Yorimitsu and Sato, 2012).

We further focused on genes annotated under the GO term "ER-to-Golgi vesicle-mediated transport" along with several lipid-binding apolipoproteins (Fig. 1C, upper panel), and performed pairwise correlation analysis of their expression across multiple tissues under hierarchical clustering. One of the clustered modules prominently featured several apolipoproteins alongside *APOB* and *MTTP*, with *SEC16B* emerging as the top candidate (Fig. 1C, lower panels). RNA-seq data and protein expression immunoblotting (IB) further illustrated that, while most COPII-related genes, including SEC16A, were broadly expressed, SEC16B showed highly restricted expression in the liver and small intestine, mirroring the distribution of APOB-containing lipoprotein production (Fig. 1D,E).

To further ascertain cell-type specificity, we analyzed single-cell RNA-seq (scRNA-seq) data from healthy mouse livers (Su et al, 2021). Dimensionality reduction by UMAP revealed ten major hepatic cell populations (Figs. 1F and EV1E). As expected, *Apob* expression was confined to hepatocytes (Fig. 1G). Notably, *Sec16b* expression mirrored this pattern, showing exclusive hepatocyte expression with negligible levels in non-parenchymal cells (Fig. 1H). A similar distribution was observed for *Hnf4a* (Fig. 1I), a hepatocyte-specific transcriptional regulator of lipid metabolism likely contributing to the transcriptional co-regulation between *Sec16b* and lipoprotein-related genes. In contrast, canonical COPII components were broadly expressed across all liver cell types, as visualized by expression signature enrichment scores (Fig. 1J).

Hence, among the early secretory pathway genes analyzed, *Sec16b* was the only SEC component with hepatocyte-specific expression resembling that of *Apob* (Fig. EV1F).

Analysis of data from Global Lipids Genetics Consortium (GLGC) (Graham et al, 2023) uncovered a single nucleotide polymorphism (SNP), rs6682862, located in the first intron of the *SEC16B* gene, which was significantly associated with plasma LDL cholesterol (LDL-C, $P < 10^{-13}$) and total cholesterol (TC, $P < 10^{-9}$), implicating *SEC16B* as potential regulator of lipid metabolism in humans (Fig. 1K,L). To contextualize these findings evolutionarily, we noted that *APOB* is a vertebrate-specific gene central to complex lipid transport. Intriguingly, *SEC16B* appears to have similarly emerged in vertebrates, suggesting its possible co-evolution with *APOB* (Figs. 1M and EV1G). In contrast, *SEC16A* and its yeast ortholog *Sec16p* represent an evolutionarily conserved scaffold essential for general COPII vesicle formation (Fig. 1M). Furthermore, SEC16B contains only a central conserved domain (CCD) that exhibits homology to the ubiquitous SEC16A (Fig. EV1H), indicating that it may have evolved as a paralog adapted to meet the specialized demands of vertebrate lipid secretion. Together, the above analysis identified SEC16B as a tissue-selective and evolutionarily aligned factor for lipoprotein secretion in vivo.

## SEC16B selectively regulates hepatic lipid secretion in a dose-sensitive manner

To examine SEC16B function in vivo, we generated a floxed allele of murine *Sec16b* (Fig. 2A). Acute liver-specific gene inactivation was achieved by AAV-mediated hepatic expression of Cre recombinase (Fig. 2B). Specifically, mice bearing homozygous (*Sec16b^{flox/flox}*, hereafter LKO) or heterozygous (*Sec16b^{flox/+}*, LHets) alleles were compared to littermate controls receiving AAV-luciferase (CTL). IB of liver proteins confirmed complete SEC16B depletion in LKO livers and partial (~50%) reduction in LHets, relative to CTLs (Fig. 2C).

Remarkably, loss of hepatic SEC16B reduced fasting plasma TGs by >60% compared to controls, whereas LHets also exhibited ~30% decrease in plasma TG levels, together revealing a critical, dose-dependent role of SEC16B in hepatic lipid transport (Fig. 2D). Fractionation of plasma by size exclusion chromatography further confirmed corresponding reductions of TG content in the atherogenic, APOB-containing VLDL and LDL fractions, upon hepatic *Sec16b* haplo-insufficiency or completion deficiency (Fig. 2E, quantified in 2F). Similarly, total plasma cholesterol levels, particularly those within atherogenic lipoproteins, were also reduced by SEC16B depletion in a dose-dependent fashion (Fig. 2G–I).

Consistent with these lipid-lowering effects by *Sec16b* inactivation, IB analysis confirmed ~70% reduction of APOB in *Sec16b* LKO mice compared to controls (Fig. 2J, quantified in 2K), along with a modest, ~50% reduction in APOA1. In contrast, plasma levels of albumin (Fig. 2J, quantified in 2K), representing approximately 90% of hepatic secretome, and the soluble secretory protein PCSK9 (Fig. EV2A), remained unaffected. Both silver staining and quantitative plasma proteomics further confirmed the overall unaltered plasma protein profiles in *Sec16b* LKO mice compared to controls, including the abundant albumin (Figs. 2L and EV2B), suggesting a selective role of SEC16B on lipoprotein transport over general protein secretion.

To directly assess hepatic TG secretion, we performed tyloxapol-based assays that block lipoprotein clearance via inhibition of lipoprotein lipase. *Sec16b* LKO mice exhibited markedly reduced TG secretion over time, compared to controls (Fig. 2M). Taken together, the murine studies aligned with our analysis of human data, collectively revealing SEC16B as a key regulator in lipoprotein transport and lipid homeostasis.

## SEC16B loss preserves liver homeostasis and triggers SREBP suppression

The unique emergence and selective function of SEC16B implies a lipid-specific *Sec* gene, leading us to assess its role in overall liver architecture and homeostasis. Transmission EM (TEM) revealed well-preserved ER structures in control or SEC16B-deficient hepatocytes, with the latter exhibited slight dilation and more frequent accumulation of lipid-laden particles stained with imidazole (Fig. 3A), indicative of a defect in lipoproteins export (Rong et al, 2015; Wang et al, 2021a). The Golgi apparatus in control hepatocytes was packed with lipid-laden particles (Fig. 3B, upper), whereas in SEC16B-deficient hepatocytes it showed markedly reduced accumulation (Fig. 3B, lower). Notably, the Golgi cisternae and their stacking appeared intact in the absence of SEC16B. Biochemical fractionation further confirmed an accumulation of APOB in the ER-enriched fractions and accordingly a reduction in the Golgi-enriched fraction isolated from the SEC16B-deficient livers, compared to controls (Fig. EV2C,D). Moreover, mitochondria and associated ER regions appeared comparable between SEC16B-deficient and control hepatocytes (Fig. 3C), consistent with SEC16B being a selective regulator for specialized lipoprotein secretion.

Histology of hepatic samples from overnight fasted mice further showed that parenchymal architectures (H/E) (Fig. 3D, left panels) were well-preserved in SEC16B-deficient livers, with mildly elevated lipid deposition (Oil red O staining) compared to controls (Fig. 3D, middle panels). However, no elevation in hepatic inflammation were observed between SEC16B-deficient or control mice (Fig. 3D, right panels). Biochemical measurements confirmed lipid accumulation in SEC16B-deficient livers under fasting conditions, in line with a blockade in hepatic lipoprotein export. However, no detectable difference in lipid contents were observed between SEC16B-deficient livers and WT control from mice in refeeding conditions, in which hepatic lipid synthesis is upregulated (Fig. 3E). Importantly, liver injury markers (plasma AST and ALT concentrations) remained unelevated in *Sec16b* LKO mice compared to controls (Fig. 3F), further confirming the overall liver health upon *Sec16b* inactivation.

To systematically assess hepatic alterations upon SEC16B deficiency, we performed liver transcriptome analysis by RNA-seq in both fasted and refeeding conditions, a paradigm known to switch metabolic gene expression programs in the liver (Trefts et al, 2017). PCA analysis confirmed robust transcriptomic shifts between fasted and refeeding conditions in controls (Fig. 3G). No such global alterations and little changes in the expression of inflammatory genes were observed in *Sec16b* LKO mice compared to controls in matching dietary conditions (Figs. 3G and EV2E), suggesting global liver metabolism and health were preserved. This allowed us to focus on genes that exhibited differential responses to the fasting/refeeding switch, between control and *Sec16b* LKO

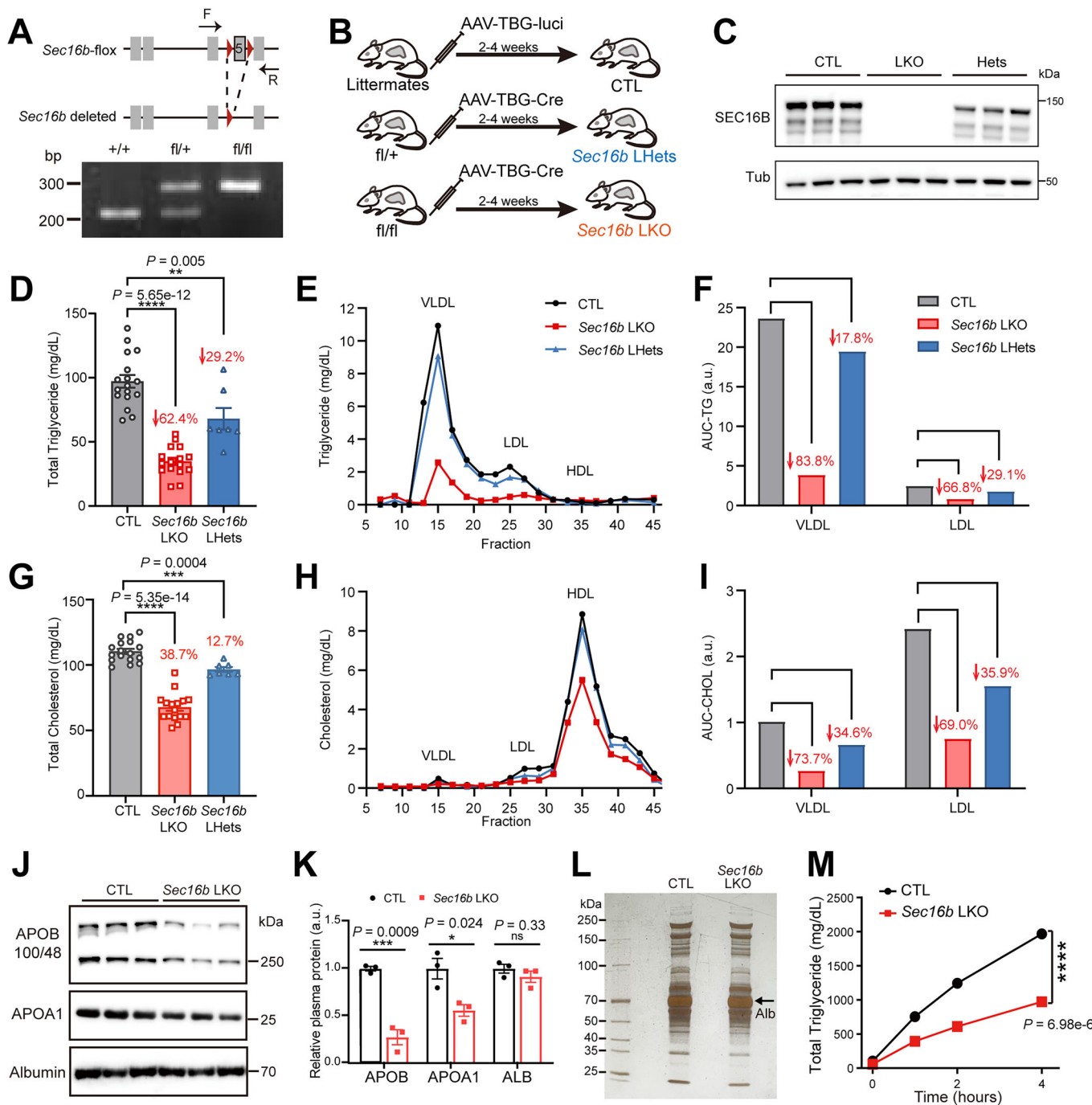

mouse. Among the fasting/refeeding responsive genes (1565 genes with >2-fold changes) in control mice, 217 showed altered regulation in *Sec16b* LKO mice (Fig. 3H). Interestingly, ~85% of these 217 genes exhibited diminished responses upon SEC16B deficiency, congregating to lipid biosynthetic pathways as the top-enriched term (Fig. 3I), suggesting a feedback mechanism suppressing lipogenesis in SEC16B-deficient livers. These results thus led to a focus on the refeeding condition known to activate hepatic lipid synthesis. Consistently, gene set enrichment analysis (GSEA) revealed significantly downregulated expression programs in fatty acid and sterol biosynthesis in *Sec16b*-deficient liver compared to controls (Fig. 3J,K).

The dampened lipogenic programs by SEC16B inactivation led us to examine SREBP1, a master transcription factor of lipid synthesis largely responsible for feeding-activated lipogenesis. Consistent with the gene expression data, fasting-induced processing and activation of SREBP was blunted in *Sec16b* LKO mice compared to controls. Similar reduction was also observed in SREBP1 targets, such as the lipogenic enzymes FASN and ACC (Fig. 3L). The data collectively revealed a selective blockade of

◄ **Figure 2. SEC16B selectively regulates hepatic lipid secretion in a dose-sensitive manner.**

(A) Schematics of the floxed murine *Sec16b* allele (upper) and PCR genotyping of the indicated alleles (lower). (B) Schematics of liver-specific *Sec16b* homozygous knockout (LKO) and heterozygous knockout (LHets) via AAV-mediated hepatic expression of Cre recombinase into *Sec16b* $^{fl/fl}$ mice and *Sec16b* $^{fl/+}$ mice, respectively. Littermates injected with AAV carrying TBG-*Luciferase* were allocated to wild-type controls (CTL). (C) IB of hepatic SEC16B in CTL, *Sec16b* LKO and *Sec16b* LHets mice. (D) Loss of hepatic SEC16B decreases plasma triglycerides in a dosage-dependent manner. Plasma triglycerides were measured in CTL ($n = 17$), *Sec16b* LKO ($n = 16$), and *Sec16b* LHets ($n = 7$). Data are shown as mean ± SEM. **$P < 0.01$, ****$P < 0.0001$ (two-tailed Student's $t$ test). (E) Triglyceride concentrations in VLDL, LDL, and HDL fractionated from pooled plasma samples in (D). Representative of three independent experiments is shown. (F) Quantification of APOB-containing VLDL and LDL triglyceride levels in (E). a.u. arbitrary units, AUC area under the curve. (G) Plasma cholesterol levels in mice from (D). Data are shown as mean ± SEM. ****$P < 0.0001$ (two-tailed Student's $t$ test). (H) Cholesterol concentrations of samples in (E). (I) Quantification of APOB-containing VLDL and LDL cholesterol levels in (H). a.u.: arbitrary units; AUC, area under the curve. (J) IB analysis of plasma samples from CTL and *Sec16b* LKO mice. Representative of three independent experiments is shown. (K) Quantification of relative IB signals in (J). Data are shown as mean ± SEM. *$P < 0.05$; ***$P < 0.001$; ns, no significance (two-tailed Student's $t$ test). a.u., arbitrary units. (L) Silver staining of plasma proteins from mice with the indicated genotypes. Representative of three independent experiments is shown. (M) Loss of hepatic SEC16B decreases triglyceride secretion into the circulation. CTL ($n = 4$) and *Sec16b* LKO ($n = 4$) mice were injected with a lipoprotein lipase (LPL) inhibitor to block lipolysis, and plasma triglycerides were measured at the indicated time points. Data are shown as mean ± SEM. ***$P < 0.001$ (two-tailed Student's $t$ test). Source data are available online for this figure.

lipoprotein secretion by SEC16B inactivation that triggers feedback suppression of lipid synthesis and preserves liver homeostasis and function, together suggesting a unique cellular mechanism operated by the lipid-selective *Sec* gene.

## SEC16B balances cellular COPII coalescence on the ER surface

To dissect the mechanism by which SEC16B selectively regulates lipid secretion, we introduced EGFP-tagged SEC16B into Huh7 hepatoma cells. Confocal microscopy showed that SEC16B coalesced into discrete puncta that co-localized with endogenous SURF4 (Fig. EV2F). Sequence analysis revealed that SEC16B contains a conserved central domain (CCD), a previously established SEC13-interacting domain (Hughes et al, 2009; Whittle and Schwartz, 2010). Alphafold3-based structural prediction further indicated that the SEC16B CCD interacts selectively with SEC13, through an interface mimicking the canonical binding between SEC13 and SEC31 in the outer COPII coat (Fig. 4A). Specifically, the N-terminus of SEC16B CCD adopts a triple β-strand conformation that occupies the vacant seventh blade position in the β-propeller of SEC13, thereby dominantly bridges the interaction between SEC16B and SEC13. The C-terminus of SEC16B CCD folds into an α-solenoid architecture that resembles the corresponding region in SEC31, and stacks against the β-propeller of SEC13. Immunoprecipitation confirmed that, while the ubiquitous paralog SEC16A interacts with all COPII components in line with its scaffold function, SEC16B interacts exclusively with SEC13 (Figs. 4B and EV2G), further implying a unique regulatory role on the outer coat SEC31/13.

When SEC31/13 were ectopically expressed in Huh7 cells, they exhibited diffusive cytosolic localization. However, co-introduction of SEC16B effectively relocated the outer coat proteins from the diffusive pattern in the cytosol to coalesced structures on the ER surface (Figs. 4C and EV2H). Given that the recruitment of outer coat proteins may counterbalance inner coat condensation, we then examined SEC24A, a component of the inner COPII coat. Confocal microscopy revealed that cellular SEC24A exhibited characteristic punctate localization on the ER surface, and ectopic expression of SEC16B reduced the punctate signal of SEC24A (Figs. 4D and EV2I,J).

Genetic inactivation in hepatocytes corroborated the above restraining role of SEC16B in COPII condensation. Endogenous SEC24A also exhibited a condensed puncta pattern in wild-type

primary hepatocytes. However, *Sec16b*-null hepatocytes displayed increased coalescence of SEC24A on the ER surface (Fig. 4E, left, quantified in 4F), a phenotype further amplified upon Brefeldin A (BFA) treatment, which inhibits ER–Golgi transport (Fig. 4E, right, quantified in 4F). Consistently, *Sec16b*-null hepatocytes exhibited markedly reduced SEC31A puncta compared to controls, both at steady state and following BFA treatment (Fig. 4G, quantified in 4H). Similarly, SEC16B inactivation in the hepatoma cell line Hepa 1–6 resulted in enhanced SEC24A coalescence and reduced SEC31A signals in puncta (Fig. EV3A–C). Taken together, the data indicate that SEC16B facilitates the constraint of inner coat SEC23/24 coalescence by modulating the outer coat SEC31/13 assembly, constituting a molecular brake on the COPII condensation.

The action of SEC16B on COPII coalescence led us to delineate its relationship with the $Mn^{2+}$ signal in lipoprotein export. Consistent with prior findings (Wang et al, 2023a), $Mn^{2+}$ treatment in WT murine primary hepatocyte dose-dependently enhanced COPII condensation and accordingly led to a bell-shaped curve for APOB secretion (Fig. EV3D–G). Intriguingly, in *Sec16b*-KO primary hepatocyte, $Mn^{2+}$ treatment further promoted COPII condensation and consequently produced an additive regulation on APOB secretion (Fig. EV3D–G). Of note, ALB secretion was largely unaffected across all conditions (Fig. EV3F,G), reflecting a preferential consumption of transport efficiency by APOB-containing lipoproteins under the control of balanced COPII condensation. Furthermore, we observed a cooperative lipid-lowering effect between SEC16B loss and the $Mn^{2+}$ signal in vivo. While daily oral $Mn^{2+}$ at 20 mg/kg reduced plasma LDL-C by ~26% ($P = 0.0098$) in WT mice, and SEC16B loss alone reduced it by nearly half ($P < 0.0001$), the combination of SEC16B loss and $Mn^{2+}$ administration led to a further reduction, culminating in a total decrease of ~77% ($P < 0.0001$, Fig. EV3H). Similar cooperative effects were also evident in plasma total cholesterol and triglyceride levels (Fig. EV3I,J).

## Abnormal COPII assembly affects ER morphology and secretion

The above results intrigued us to generate a SEC16B mutant lacking the CCD domain ($^{\Delta CCD}$SEC16B) to selectively uncouple its regulation on the outer coat (Fig. EV4A). Strikingly, the $^{\Delta CCD}$SEC16B mutant induced protein over-aggregation to form abnormal assemblies (1–10 μm in diameter), which excluded

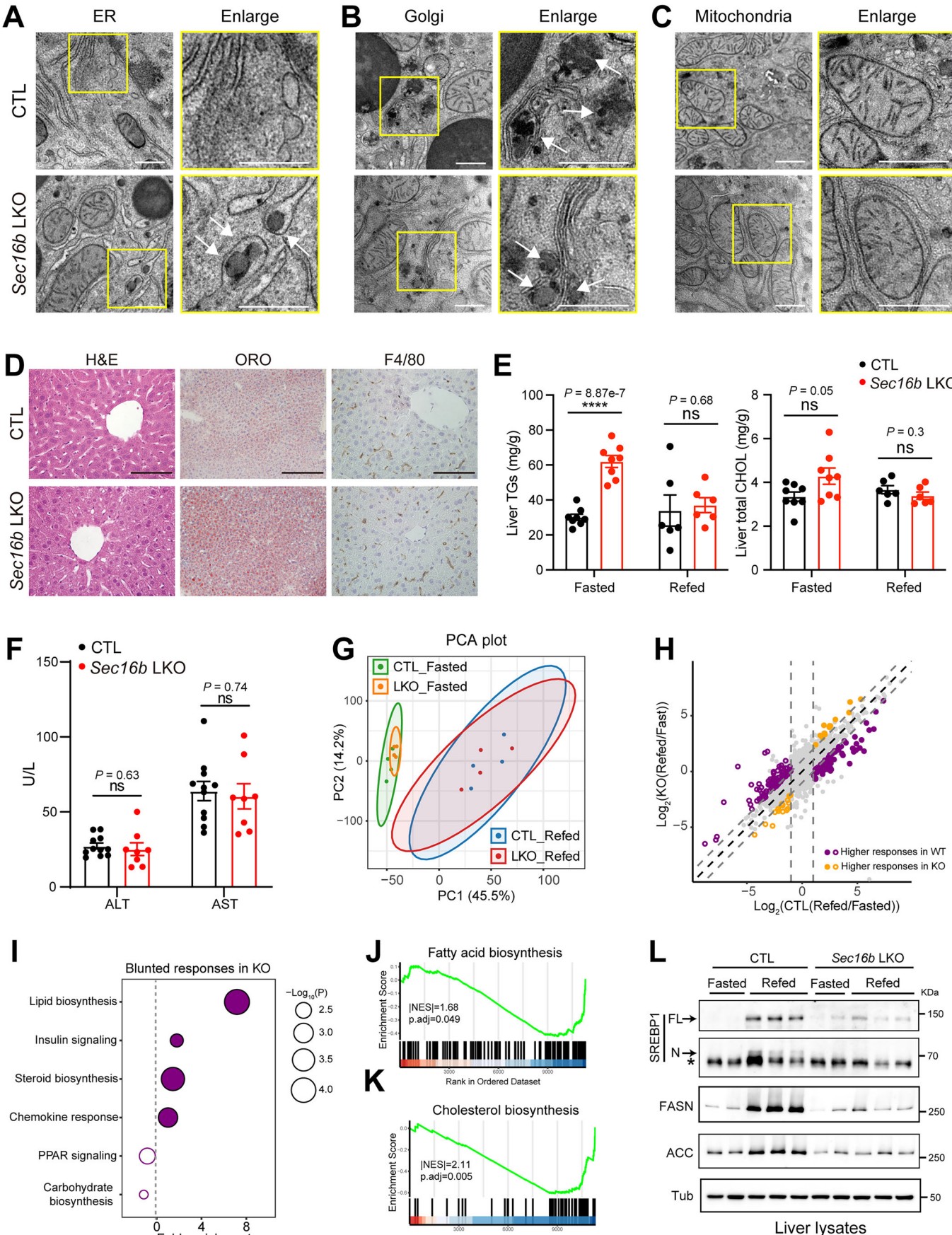

**Figure 3. SEC16B loss preserves liver homeostasis and triggers SREBP suppression.**

(A–C) Ultrastructural analysis by TEM on the ER (A), the Golgi (B), and mitochondria (C) of liver samples from control (upper) and *Sec16b*-deficient (lower) mice under fasting conditions. White arrows indicate lipid-laden particles in the ER lumen and Golgi. Scale bars, 500 nm. (D) Representative H&E, Oil Red O (ORO), and F4/80 staining of liver sections from CTL and *Sec16b* LKO mice. Scale bars = 100 μm except 200 μm for ORO. (E) Quantification of TG and total cholesterol contents in lipid extracts of liver samples from indicated genotypes under fasting or refeeding conditions. n = 8 and 6 biological independent samples for fasting conditions and refeeding conditions, respectively. ****P < 0.0001; ns, no significance (two-tailed Student's *t* test). (F) Plasma alanine aminotransferase (ALT) and aspartate aminotransferase (AST) levels of the indicated mice. n = 11 and 8 for CTL and *Sec16b* LKO mice, respectively. Data are shown as mean ± SEM, ns, no significance (two-tailed Student's *t* test). (G) Principal component analysis (PCA) of mRNA expression profiles in the liver samples with the indicated genotypes under fasting or refeeding conditions. Ellipses: 95% confidence regions assuming bivariate normal distribution. Overlapping ellipses between CTL and *Sec16b* LKO under the same feeding conditions indicate similar transcriptional profiles. n = 4 for each group. (H) Scatter plot of fold changes for mRNA transcript levels between fasting and refeeding states in *Sec16b*-deficient (y axis) versus control livers (x axis). Vertical dashed lines indicate genes with >twofold changes between fasting and refeeding states in controls. Upregulated and downregulated genes in the refeeding state are presented by filled circles and open circles, respectively. Diagonal dashed lines present positions of a twofold difference between the refeeding-to-fasting response in CTL and that in *Sec16b*-deficient livers. Circles colored with purple and yellow highlight genes with blunted and enhanced response in *Sec16b* KO livers, respectively. (I) Bubble plot illustrating GO enrichment of blunted refeeding-to-fasting responsive genes in *Sec16b*-deficient livers, compared to controls. Filled bubbles and open circles represent activated pathways and suppressed pathways under the refeeding condition, respectively. Enrichment analysis was performed using Metascape, which tests for overrepresentation of GO terms by the hypergeometric test. P values were adjusted for multiple comparisons via the Benjamini-Hochberg correction. (J, K) GSEA plots showing the significant downregulation in fatty acid biosynthesis (J) and Cholesterol biosynthesis (K) in *Sec16b* LKO mice versus controls under refeeding conditions. Significance was determined by normalized enrichment score (ES) with FDR q value < 0.05, calculated from an empirical phenotype-based permutation test. (L) IB analysis of hepatic SREBP1 cleavage and lipogenic enzymes in liver samples from mice with the indicated genotypes and refeeding-fasting conditions. n = 2 and 3 biological independent samples for fasting conditions and refeeding conditions, respectively. Representative of three independent biological replicates is shown. Source data are available online for this figure.

soluble proteins such as GFP (Fig. 5A, upper). These abnormal assemblies were highly enriched with the inner COPII coats SEC23/24 (Fig. 5A, middle) but largely excluded the outer COPII coat subunit SEC31A, which instead decorated their periphery (Fig. 5A, lower). Consistent with a role of IDR-mediated multivalency in COPII condensation (Wang et al, 2023a), SEC24 mutants lacking its IDR exhibited less partition into the mega-aggregates. A chimeric SEC24 bearing the orthogonal IDR from FUS, prone to over-condensation as previously reported (Banani et al, 2017), displayed exaggerated aggregation (Fig. 5B, quantified in 5C).

To further investigate COPII over-aggregation in a more controllable manner, we developed an inducible system using FKBP–FRB mediated dimerization (Inobe and Nukina, 2016). We fused the N-terminal fragment SEC16B mutant to FKBP and the C-terminal fragment to FRB. Upon addition of rapamycin, the two fusion proteins were brought together (Fig. 5D), sufficient to induce abnormal COPII assemblies in 30 min (Fig. 5E). Live-cell imaging revealed that the outer coat SEC31A was progressively excluded from these inducible assemblies enriched with the inner coats (Fig. 5F). Consistently, co-immunoprecipitation confirmed uncoupling of the inner coat from the outer coat by the ^ΔCCDSEC16B mutant despite normal expression of these coat components (Fig. 5G), further suggesting a disruption in the proper assembly and function of the COPII transport machinery. Of note, the abnormal COPII assemblies induced by ^ΔCCDSEC16B gradually lost fluidity and molecular mobility, as evidenced by their resistance to 1,6-hexanediol (1,6-Hex) and impaired fluorescence recovery after photobleaching (FRAP) (Fig. EV4B–D).

We thus further investigated the functional consequence of abnormal COPII assemblies induced by the ^ΔCCDSEC16B mutant. Surprisingly, ultrastructural analysis by TEM revealed that these SEC23/24-enriched assemblies were populated by reorganized, crystalloid ER structures (Fig. 5H) in which parallel membranes were closely zipped together. The "zipper" was characterized by a narrow cytosolic gap (~20 ± 5 nm) filled with electron-dense signals indicative of protein enrichment (arrows in the lower panel, Fig. 5H), resembling ER re-organization induced by weak yet multivalent protein-protein interactions on the ER surface

(Lingwood et al, 2009; Snapp et al, 2003; Yu et al, 2021). Functionally, ^ΔCCDSEC16B introduction inhibited the ER exit of COPII cargos, including SURF4 and LMAN1, in an in vitro budding assay (Fig. 5I). In addition, hepatic expression of ^ΔCCDSEC16B in adult mice significantly reduced plasma lipids and provoked liver injury (Fig. EV4E–I). Collectively, the imaging and functional data reveal that uncontrolled COPII coalescence on the ER surface disrupts the dynamics and function of COPII, furthering a unique role of SEC16B tailored for the delicate balance in lipoprotein secretion and systemic lipid control.

## HNF4A-mediated *SEC16B* regulation associates with human hepatic phenotypes

The molecular actions of SEC16B reported above further intrigued us to search for its potential relevance. Analysis of data from UK Biobank (UKBB) revealed that the minor allele of rs6682862 in human *SEC16B* exhibited a significant association with elevated plasma alkaline phosphatase (ALP) levels, albeit with a small positive β value, in populations consisting of ~344,000 European ancestry individuals and ~118,000 East Asian ancestry individuals (Fig. 6A,B). Elevation of circulating ALP levels often reflects impaired liver health, leading us to further examine the involvement of SEC16B as a hepatic gene. The minor allele A of rs6682862 is associated with an increase in *SEC16B* gene expression and the correlation is dose-dependent (Fig. 6C). This intronic region also displayed DNase I hypersensitivity and enrichment of histone modifications, including H3K27Ac and H3K4me3 (Fig. 6D), indicating this region with regulatory roles in gene expression.

We further cloned a 150-bp intronic fragment flanking the rs6682862 SNP upstream of a luciferase reporter to test its transcriptional regulation (Fig. 6E). The fragment containing the major allele substantially enhanced luciferase expression compared to control, confirming the regulatory activity of this intronic region. Moreover, the fragment containing the alternative allele led to about 50%, additional transcription activity than the major allele (Fig. 6F).

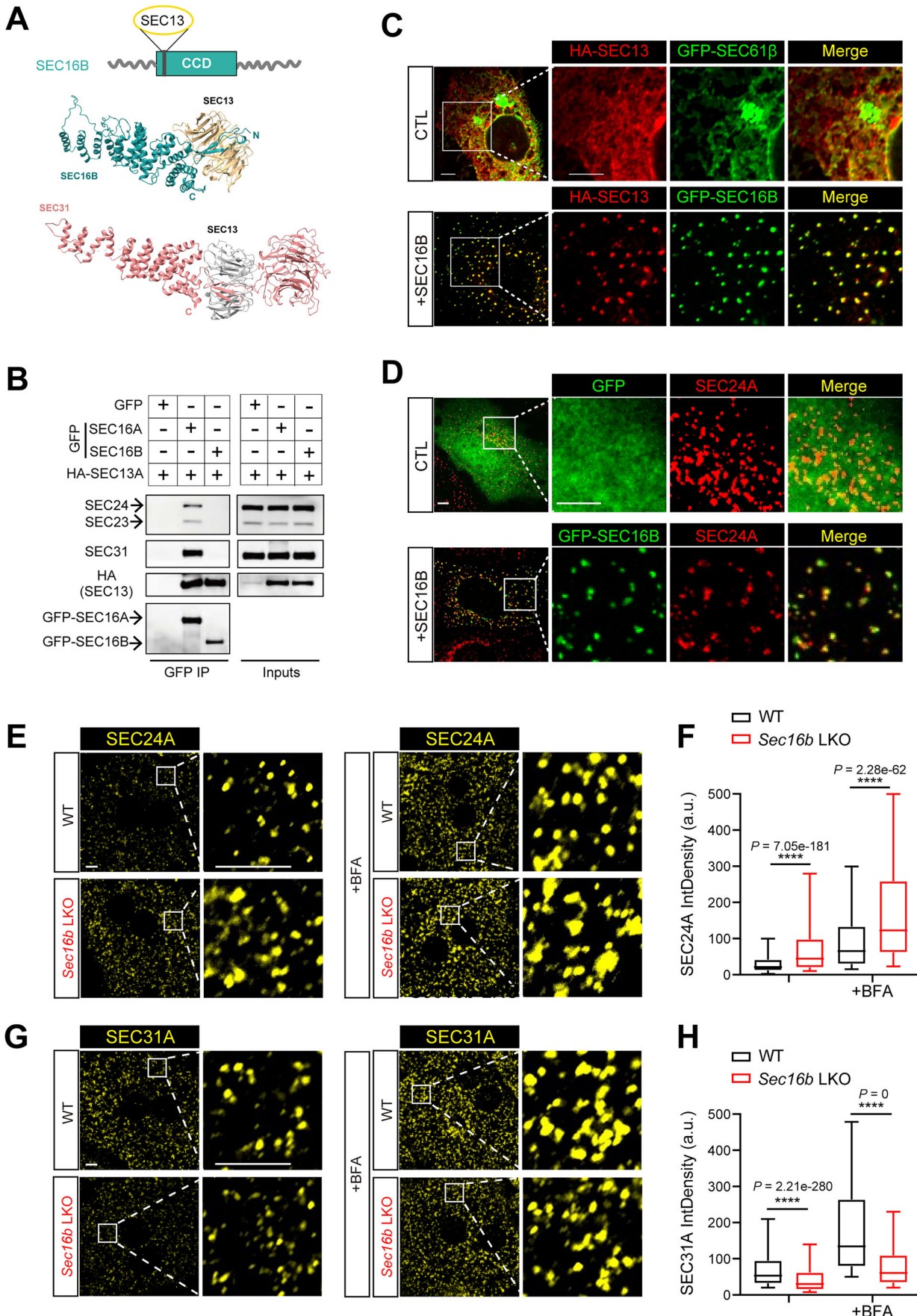

**Figure 4.   SEC16B balances cellular COPII coalescence on the ER surface.**

(A) Upper: molecular schematic of SEC16B. CCD, Central conserved domain. Lower: Alphafold3-predicted interacting interface between SEC16B and SEC13, resembling that of SEC31/SEC13 complex (PDB ID: 2PM6). (B) Coimmunoprecipitation (co-IP) of GFP-SEC16A or GFP-SEC16B with COPII inner and outer coats. Lysates from Huh7 cells transfected with the indicated constructs were subjected to anti-GFP IP prior to IB with the indicated antibodies. (C) SEC16B promotes membrane localization of outer coat proteins. Huh7 cells transfected with Flag-SEC31A and HA-SEC13 (red), plus GFP-SEC61β or GFP-SEC16B WT (green) were analyzed by confocal microscopy following HA staining. Scale bars = 5 μm. (D) SEC16B reduces COPII coalescence on the ER surface. Huh7 cells transfected with GFP (upper), or GFP-SEC16B WT (lower) were fixed and stained with an anti-SEC24A (red) antibody prior to confocal microscopy. Scale bars = 5 μm. (E) Loss of hepatic SEC16B enhances SEC24A coalescence. Primary hepatocytes isolated from control (WT) or *Sec16b* LKO mice were stained with the anti-SEC24A antibody prior to confocal microscopy. Scale bars = 5 μm.
(F) Quantification of the integrated fluorescence signal density of SEC24A puncta in (E). Data are shown as a box plot with 5–95% distribution, with the line inside the box and box edges representing median and the interquartile ranges, respectively. n = 2831, 1741, 2170, and 2823 puncta for WT, *Sec16b* LKO, WT with BFA treatment, and *Sec16b* LKO with BFA treatment, pooled from 10 cells per condition, respectively. ****$P < 0.0001$ (two-tailed Student's t test). a.u. arbitrary units. (G) Loss of hepatic SEC16B decreases SEC31A signals in puncta. Primary hepatocytes isolated from control (WT) or *Sec16b* LKO mice were stained with the anti-SEC31A antibody prior to confocal microscopy. Scale bars = 5 μm. (H) Quantification of the integrated fluorescence signal density of SEC31A puncta in (G). Data are shown as a box plot with 5–95% distribution, with the line inside the box and box edges representing median and the interquartile ranges, respectively. n = 4370, 8347, 3980, and 5396 puncta for WT, *Sec16b* LKO, WT with BFA treatment, and *Sec16b* LKO with BFA treatment, pooled from 10 cells per condition, respectively. ****$P < 0.0001$ (two-tailed Student's t test). a.u. arbitrary units. Source data are available online for this figure.

HNF4α, a master transcription factor controlling hepatic gene regulation, displayed two ChIP-seq binding peaks within the *SEC16B* promoter region in both human liver tissue and HepG2 cells, one of which overlapped the rs6682862-containing region (Fig. 6G). Indeed, the hepatic mRNA expression levels of *Sec16b* were reduced by nearly 80% in HNF4α liver-specific knockout mice, by analyzing public RNA-seq data (Van Dender et al, 2024) (Fig. 6H). Likewise, treatment of primary hepatocytes with an HNF4α inhibitor (Kiselyuk et al, 2012) significantly decreased SEC16B protein levels (Figs. 6I, quantified in 6J and EV5A). As expected, protein levels of the direct target HNF4α (Guo and Lu, 2019) and its known downstream target MTP were also reduced, confirming effective inhibition (Fig. 6I, quantified in 6J).

Complementing these human genetic findings, hepatic SEC16B protein levels were significantly elevated in high-fat diet (HFD)-induced obese mice compared to lean controls (Fig. 6K, quantified in 6L). Similarly, hepatic SEC16B was also elevated in hyperlipidemia (Fig. 6K, quantified in 6L). Accordingly, hepatic overexpression of SEC16B in adult mice induced an upward trend in plasma lipid levels, although not significant (Fig. EV5B–D). Together with the association data obtained from UKBB study, these findings suggest that hepatic SEC16B expression is upregulated under pathological lipid-loading conditions, further indicating its relevance as a potential therapeutic target for lipid-related diseases.

## Therapeutic potentials of SEC16B targeting in dyslipidemia and ASCVD

The above results intrigued us to test the therapeutic of SEC16B in pathological conditions. To this end, we used AAV-mediated delivery of PCSK9 to induce hyperlipidemia and atherosclerosis in *Sec16b fl/fl* mice, with co-delivery of liver-specific *Cre* to inactivate *Sec16b* or GFP as a control (Fig. 7A). Strikingly, hepatic deletion of *SEC16B* led to a dramatic reduction in both plasma cholesterol (Fig. 7B) and triglycerides (Fig. 7C), despite severe hyperlipidemia in control mice. Lipoprotein fractionation confirmed that most of the cholesterol reduction occurred in atherogenic lipoproteins— VLDL and LDL (Fig. 7D), with ~85% and ~80% decreases in VLDL and LDL cholesterol, respectively (Fig. 7E). Similarly, triglyceride reductions were driven by marked decreases in APOB-containing VLDL (~80%) and LDL (~50%) particles (Fig. 7F,G).

Importantly, coronary flow reserve (CFR), a metric of cardiac function, was significantly preserved in SEC16B-deficient mice (Fig. 7H–J), indicating protection against atherosclerosis-associated vascular dysfunction. Consistent with these changes, *en face* Oil Red O staining revealed severe atherosclerosis in control mice but an ~80% reduction in plaque area in *Sec16b* liver-specific knockout (LKO) mice (Fig. 7K, quantified in 7L). Plaque burden and macrophage infiltration in aortic sinus were also profoundly attenuated in *Sec16b* LKO mice (Fig. EV5E,F).

Notably, hepatic *Sec16b* inactivation in the hyperlipidemic conditions did not induce overt liver toxicity, indicated by unchanged serum ALT and AST levels (Fig. 7M), as well as no elevation in inflammatory gene expression (Fig. EV5G). Liver histology showed no major differences in morphology (H&E), lipid accumulation (Oil Red O) (Fig. 7N). Biochemical measurements of lipid extracts confirmed no elevation in contents of TGs and total cholesterol in SEC16B-deficient livers (Fig. 7O), likely stemming from a compensatory suppression in SREBP activation (Fig. EV5H,I). Sirius Red staining further indicated no overt fibrosis induced by hepatic SEC16B inactivation (Fig. 7N, quantified in 7P, left). Interestingly, immunohistochemistry for F4/80 revealed reduced macrophage infiltration in SEC16B-deficient livers (Fig. 7N, right panels, quantified in 7P, right), suggesting dampened hepatic inflammation. Together with the previous findings, our study highlights SEC16B as a novel, translationally relevant therapeutic target with both safety and specificity for the treatment of pathogenic dyslipidemia and atherosclerosis.

## Discussion

Through an integrative pipeline combining large-scale bioinformatics, molecular modeling, and in vivo functional studies, we identify SEC16B as a tissue-selective regulatory node that tailors the early secretory (*SEC*) pathway for lipid-bearing lipoproteins. Specifically, SEC16B balances COPII condensation at the ER, for the efficient export of abundant and bulky cargos such as lipid-rich lipoproteins. Functional disruption of hepatic *SEC16B* selectively impairs lipoprotein secretion under both basal and in particular pathological conditions, profoundly lowering atherogenic factors including APOB, triglycerides, and cholesterol in the circulation. In

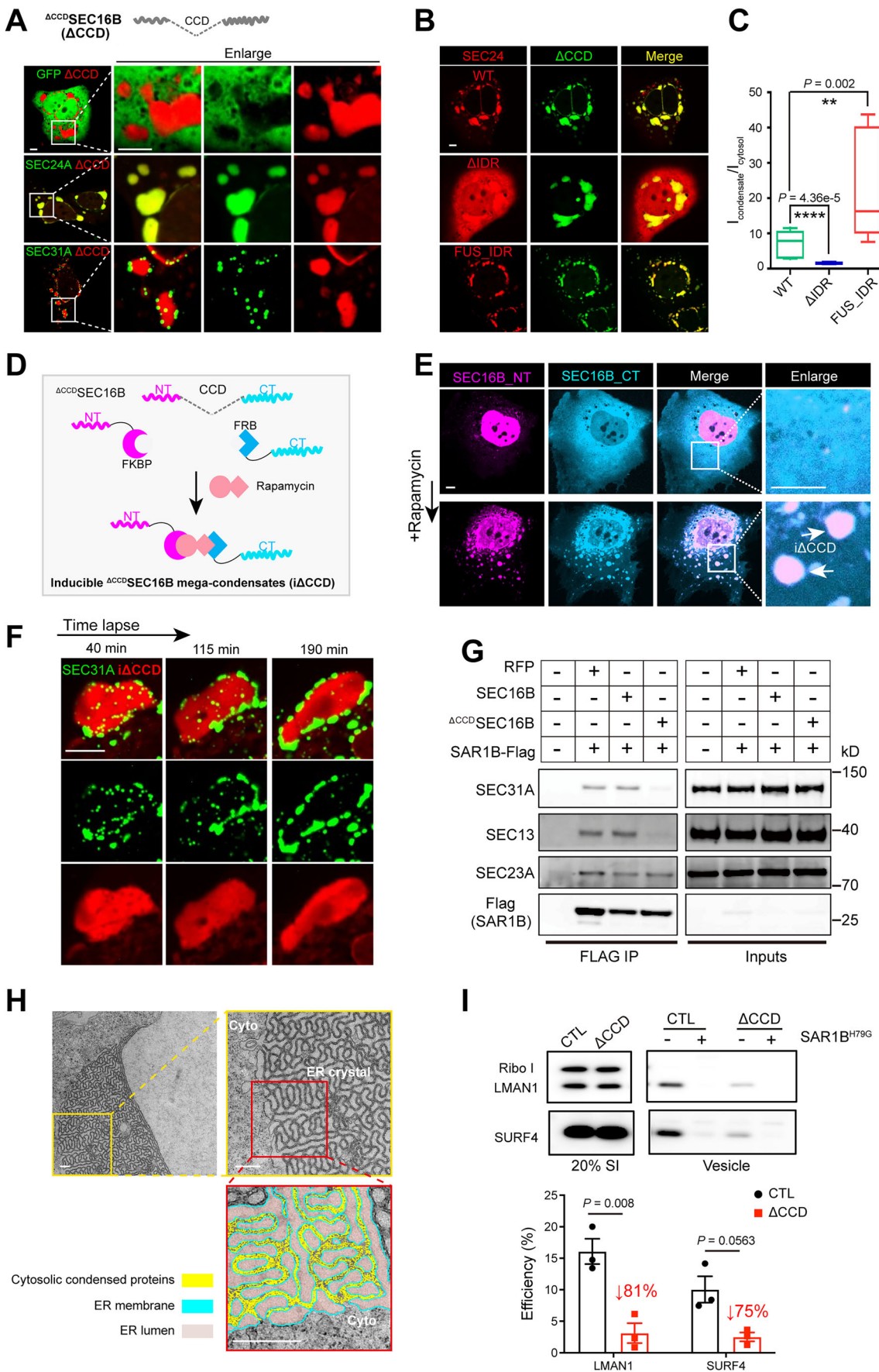

**Figure 5. Abnormal COPII assembly affects ER morphology and secretion.**

(A) Composition of the $^{\Delta CCD}$SEC16B induced abnormal assemblies. Huh7 cells co-transfected with the indicated GFP or GFP-tagged COPII constructs and RFP-$^{\Delta CCD}$SEC16B were imaged with confocal microscopy. Upper: a molecular schematic of $^{\Delta CCD}$SEC16B. Scale bars = 5 μm. (B) Confocal images of Huh7 cells co-transfected with RFP tagged SEC24 mutants and GFP-$^{\Delta CCD}$SEC16B. WT: wild-type form of SEC24. $^{\Delta IDR}$SEC24: SEC24 lacking N-terminal IDR. $^{FUS\_IDR}$SEC24: IDR of FUS re-introduced into $^{\Delta IDR}$SEC24. Scale bars = 5 μm. (C) Quantification of the fluorescent signal ratios of different SEC24s between condensed abnormal assemblies and the cytosol in (B). Data are shown as a box plot with the line inside the box and box edges representing the median and the interquartile ranges, respectively. $n = 11, 9, 13$ cells for WT, $^{\Delta IDR}$SEC24, and $^{FUS\_IDR}$SEC24, respectively. **$P < 0.01$; ****$P < 0.0001$ (two-tailed Student's $t$ test). (D) Schematics of an inducible $^{\Delta CCD}$SEC16B mega-aggregate system (i$\Delta$CCD). mRFP tagged N-terminal domain of SEC16B was fused to FKBP (SEC16B_NT), and ECFP tagged C-terminal domain of SEC16B was fused to FRB (SEC16B_CT). Rapamycin induces the interaction between FKBP and FRB, triggering the engagement of SEC16B_NT and SEC16B_CT and subsequent mega-aggregate formation. (E) Confocal images of Huh7 cells co-transfected SEC16B_NT and SEC16B_CT, before (upper) and after (lower) rapamycin treatment. Scale bars = 5 μm. (F) Time-lapse imaging of induced $^{\Delta CCD}$SEC16B (i$\Delta$CCD) by rapamycin and GFP-SEC31A on Huh7 cells. Scale bars = 5 μm. (G) The abnormal assemblies induced by $^{\Delta CCD}$SEC16B mutant uncouples the inner and outer COPII coats. HEK 293 A cells co-transfected with FLAG-tagged SAR1B$^{H79G}$ and GFP, SEC16B WT or $^{\Delta ACE}$SEC16B were lysed and subjected to anti-FLAG IP. Proteins present in the immune complex (FLAG IP) or total cell lysates (inputs) were detected by IB following SDS-PAGE. (H) TEM morphology of the $^{\Delta CCD}$SEC16B abnormal assemblies on the ER surface. Huh7 cells expressing $^{\Delta CCD}$SEC16B were fixed and subjected to TEM analysis. The color rendering based on the electron density signal indicates the ER membrane, cytosolic condensed protein, and ER lumen. Scale bars = 0.5 μm. Cyto cytosol. (I) Abnormal COPII assembly impairs cargo packaging efficiency. Semi-intact cells prepared from Huh7 cells expressing RFP (CTL) or RFP-$^{\Delta CCD}$SEC16B ($\Delta$CCD) were incubated with cytosol from mouse liver, and indicated reagents. Proteins present in the vesicle fractions or donor membrane (SI) were detected by IB following SDS-PAGE. Lower: Quantification of LMAN1 and SURF4 packaging efficiency. Data are shown as mean ± SEM. $n = 3$ biologically independent replicates. *$P < 0.05$; **$P < 0.01$ (two-tailed Student's $t$ test). Source data are available online for this figure.

murine models, hepatic SEC16B targeting results in robust protection against atherosclerosis and preserved coronary flow reserve, with minimal if any hepatotoxicity. An accompanying independent study corroborates these findings, reinforcing SEC16B's potential as a safe and effective therapeutic target for atherosclerotic cardiovascular disease (ASCVD).

The advent of high-quality transcriptomic and genomic resources, coupled with AI-enhanced analytics, offers unprecedented opportunities to study complex physiology (Abramson et al, 2024; Gao et al, 2024). One such example, as illustrated in the current study, center on the lipid-ferrying lipoproteins that integrate both metabolic and transport homeostasis in a spatial-and-temporal-specific manner. We thus leverage a combinatory approach, with co-variance in expression to pinpoint the candidate SEC16B, GWAS analysis to identify human relevance and regulation by HNF4A, as well as AlphaFold-assisted structure prediction to formulate the hypothesis that SEC16B tempers COPII condensation, acting as a molecular "brake" to prevent excessive or dysfunctional coat buildup.

The molecular mechanism builds upon our previous work showing that balanced COPII condensation, particularly with the outer coat proteins SEC13/31 restraining the inner coat SEC23/24, plays a key role in high-capacity transport of lipoproteins. SEC16B appears to control this threshold, allowing for sufficient but not excessive COPII clustering, thus maintaining secretion competency while avoiding a traffic jam. This molecular brake function of the SEC16B protein appears to complement the promoting effects by the manganese signal on COPII condensation, further constituting the balance in the alternative assembly mechanism of the COPII machinery for transporting bulky and abundant cargos. In this regard, SEC16B targeting may provide a specific approach to modulate the condensation process of hepatic COPII in vivo, leading to effective yet selective inhibition of lipoprotein export. Although hepatocytes do not actively produce procollagen, the regulatory mechanism uncovered in this study may contribute to the secretion of this bulky and abundant cargo, warranting further investigation into its interplay with factor such as the TANGO family proteins (Malhotra, 2024; Malhotra, 2025; Saito et al, 2009) and additional regulatory mechanisms of the COPII machinery (Gallo et al, 2023; Peotter et al, 2019; van Leeuwen et al, 2022; Zacharogianni et al, 2014).

Existing lipid-lowering therapies targeting APOB-containing lipoproteins, including APOB RNAi and MTTP inhibitors, have proven efficacious in genetic hyperlipidemias like homozygous familial hypercholesterolemia (HoFH) (Gebhard et al, 2013; Rader and Kastelein, 2014; Santos et al, 2015). However, these strategies frequently cause hepatic lipid accumulation and steatosis due to disrupted early lipoprotein assembly (LiverToxTeam, 2012). Recently, independent studies in murine models have revealed that arresting lipoprotein at the ER exit by inactivating hepatic SURF4 enables robust lipid-lowering, while causing only mild lipid accumulation and no detectable hepatoxicity (Shen et al, 2022; Tang et al, 2022; Wang et al, 2021a). This improvement is likely attributable to the inhibition of the SREBP pathway triggered by the selective blockade of the late, post-ER stages in lipoprotein export, unmasking a pivotal regulatory node where lipid biogenesis and transport are delicately coordinated. Since SURF4 is widely expressed, the selective enrichment of *SEC16B* in the liver and intestine offers a more targeted approach with reduced systemic impact. Indeed, this study demonstrated that *Sec16b* LKO mice exhibited little hepatic lipid over-accumulation under fed conditions, which resembles the scenario of nutritional sufficiency in normal human physiology. Consequently, hepatic inactivation of SEC16B in pathological hyperlipidemia models not only prevented ASCVDs, but also preserved or even to some extent improved liver health. This potential protective effect likely reflects a less pro-inflammatory hepatic microenvironment fostered by reduced hepatic SREBP activation and lowered circulating LDL levels.

Despite therapeutic progress, ASCVD remains a global health burden. Residual cardiovascular risks persist even after LDL-cholesterol control, particularly due to elevated triglycerides and lipoprotein(a) [Lp(a)], which escapes LDLR-mediated clearance and is enriched in oxidized phospholipids (Finneran et al, 2021; Saeed et al, 2021b; van Capelleveen et al, 2016). Though mouse models lack *LPA*, the human gene encoding apo(a) (Assini et al, 2023), Lp(a) shares *APOB* as its core scaffold (Jawi et al, 2020; Saeed et al, 2021a). Thus, future studies in non-human primates or engineered models will be needed to assess whether *SEC16B* inhibition modulates Lp(a) levels. Together, our study positions SEC16B as a physiologically essential, vertebrate-specific modulator of COPII dynamics in lipid control, highlighting the potential of targeting lipoprotein secretion to produce lipid-lowering without overt hepatotoxicity.

# Methods

### Reagents and tools table

| Reagent/resource | Reference or source | Identifier or catalog number |
| --- | --- | --- |
| **Experimental models** | | |
| HEK293T | ATCC | Cat# CRL-3216 |
| HEK293A | Thermo Fisher | Cat# R70507 |
| Huh7 | JCRB | Cat# JCRB0403 |
| *Sec16b^{fl/fl}* mice | This paper | N/A |
| **Recombinant DNA** | | |
| DeltaF6 | Addgene | Cat# 112867 |
| pRC2/8 | Addgene | Cat# 112864 |
| AAV-TBG-GFP | Addgene | Cat# 105535 |
| AAV-TBG-CRE | This paper | N/A |
| AAV-TBG-Luciferase | This paper | N/A |
| psPAX2 | Addgene | 12260 |
| pMD.2 G | Addgene | 12259 |
| AAV-TBG-FLAG-SEC16B | This paper | N/A |
| AAV-TBG-FLAG-ΔCCDSEC16B | This paper | N/A |
| CMV- mRFP-SEC13 | This paper | N/A |
| CMV- HA-SEC13 | This paper | N/A |
| CMV-mEGFP-SEC16B | Addgene | Cat# 15775 |
| CMV-mEGFP-SEC16A | Addgene | Cat# 15776 |
| CMV-mEGFP | This paper | N/A |
| CMV-mERFP1-SEC24A | Schekman lab | Chen et al, 2013 https://doi.org/10.7554/eLife.00444 |
| CMV-mEGFP-SEC24A | This paper | N/A |
| CMV-mEGFP-SEC31A | Schekman lab | Chen et al, 2013 https://doi.org/10.7554/eLife.00444 |
| CMV-Flag-SEC31A | This paper | N/A |
| CMV-mRFP-ΔCCDSEC16B | This paper | N/A |
| CMV-mEGFP-ΔCCDSEC16B | This paper | N/A |
| CMV-mRFP-ΔIDRSEC24D | This paper | N/A |
| CMV-mRFP-FUS_IDRSEC24D | This paper | N/A |
| CMV-mRFP-SEC24A | This paper | N/A |
| CMV-ECFP-FRB-SEC16B-CTD | This paper | N/A |
| CMV-FKBP-SEC16B-NTD-mRFP | This paper | N/A |
| CMV-SAR1B^{H79G}-BirA-Flag | Xiao-Wei Chen lab | Nie et al, 2018 https://doi.org/10.1073/pnas.1704639115 |
| Firefly | luciferase Promega | Cat# E1751 |
| Renilla | luciferase Promega | Cat# E2241 |

| Reagent/resource | Reference or source | Identifier or catalog number |
| --- | --- | --- |
| AAV- HCRApoE/hAAT-hPCSK9 D374Y | Addgene | Cat #58379 |
| CMV-HA-SEC13 | This paper | N/A |
| CMV-mGFP-SEC61β | This paper | N/A |
| Lenti-CRISPR-V2 | Addgene | Cat# 52961 |
| **Antibodies** | | |
| Rabbit anti-SEC16B | This paper | N/A |
| Rabbit anti-SEC16A | Proteintech | Cat# 20025-1-AP |
| Rabbit anti-APOB | Proteintech | Cat# 20578-1-AP |
| Rabbit anti-APOA1 | Fitzgerald | Cat# 70R-15769 |
| Rabbit anti-ALB | Proteintech | Cat# 16475-1-AP |
| Rabbit anti-TUBA1B | Proteintech | Cat# 11224-1-AP |
| Rabbit anti-SEC23A | Cell Signaling Technology | Cat# 8162 |
| Rabbit anti-SEC24A | Scheckman lab | Chen et al, 2013 https://doi.org/10.7554/eLife.00444 |
| Rabbit anti-SEC24D | Scheckman lab | Chen et al, 2013 https://doi.org/10.7554/eLife.00444 |
| Rabbit anti-SEC13 | Proteintech | Cat# 15397-1-AP |
| Rabbit anti-SEC31A | Proteintech | Cat# 17913-1-AP |
| Mouse anti-SREBP1 (2A4) | Santa Cruz Biotechnology | Cat# sc-13551 |
| Rabbit anti-FASN | Proteintech | Cat# 10624-2-AP |
| Rabbit anti-ACC | Abcam | Cat# ab45174 |
| Mouse anti-HA | Proteintech | Cat# 66006-2-Ig |
| Rabbit anti-GFP | Proteintech | Cat# 50430-2-AP |
| Rabbit anti-FLAG | Proteintech | Cat# 20543-1-AP |
| Rabbit anti-SURF4 | Xiao-Wei Chen lab | Wang et al, 2021a https://doi.org/10.1016/j.cmet.2020.10.020 |
| Rabbit anti-LMAN1 | Proteintech | Cat#13364-1-AP |
| Mouse anti-MTP | Santa Cruz Biotechnology | Cat# sc-135994 |
| Rabbit anti-HNF4A | Abcam | Cat# ab181604 |
| Rabbit anti-F4/80 (D2S9R) | Cell Signaling Technology | Cat# 70076 |
| Rabbit anti-CD68 | Abcam | Cat# ab125212 |
| Rabbit anti-PCSK9 | Santa Cruz Biotechnology | Cat# sc-66996 |
| Rabbit anti-CNX | Proteintech | Cat# 10427-2-AP |
| Rabbit anti-GM130 | Proteintech | Cat# 11308-1-AP |

| Reagent/resource | Reference or source | Identifier or catalog number |
|---|---|---|
| Goat anti-Rabbit IgG (H + L) Cross-Adsorbed Secondary Antibody, Alexa FluorTM 488 | Invitrogen | Cat# A11008 |
| Goat anti-Rabbit IgG (H + L) Cross-Adsorbed Secondary Antibody, Alexa FluorTM 647 | Invitrogen | Cat# A21244 |
| Goat anti-mouse IgG (H + L) Cross-Adsorbed Secondary Antibody, Alexa FluorTM 568 | Invitrogen | Cat# A11031 |
| HRP-conjugated Goat anti-Rabbit IgG (H + L) Secondary Antibody | Invitrogen | Cat# 31460 |
| HRP-conjugated Goat anti-Mouse IgG (H + L) Secondary Antibody | Invitrogen | Cat# 31430 |
| **Oligonucleotides and other sequence-based reagents** | | |
| Primer sequences for genotyping | This paper | Dataset EV1 |
| Primer sequences for qPCR | This paper | Dataset EV1 |
| Sequences of oligonucleotides for sgRNA constructs | This paper | Dataset EV1 |
| **Chemicals, enzymes, and other reagents** | | |
| Anti-DYKDDDDK Affinity Beads | Smart Lifesciences | Cat# SA042001 |
| Proteinase K | TIANGEN | Cat# 0706 |
| Benzonase | Sigma-Aldrich | Cat# E1014 |
| Optiprep | Sigma-Aldrich | Cat# D1556 |
| FBS | Vistech | Cat# SE100-001 |
| PEI | Polyscience | Cat# 23966-1 |
| Protease Inhibitor | Roche | Cat# 4693132001 |
| 1,6-Hexanediol | Sigma-Aldrich | Cat# 240117 |
| Triton X-100 | Sigma-Aldrich | Cat# T8787 |
| Brefeldin A | Cell Signaling Technology | Cat# 9972S |
| TRIzol Reagent | Thermo Fisher | Cat# 15596018 |
| Tyloxapol | Sigma | Cat# T8761 |
| Phanta Max Super-Fidelity DNA Polymerase | Vazyme | Cat# P505-d2 |
| BI6015 | Cayman | Cat# 12032 |
| Collagenase IV | Sigma-Aldrich | Cat# C5138 |
| Fluorescence Mounting Medium | DAKO | Cat# S196430 |
| Ponceau S | Solarbio | Cat# P0012 |
| **Software** | | |
| GraphPad Prism v8.40 | https://www.graphpad.com/features | |
| Snapgene v4.3.6 | https://www.snapgene.com | |
| ImageJ (Fiji) | https://imagej.net NIH | |

| Reagent/resource | Reference or source | Identifier or catalog number |
|---|---|---|
| Zen2 blue edition | https://www.zeiss.com/microscopy/en/service-support/downloads.html Zeiss | |
| Benchling | https://benchling.com | |
| RStudio 2023.03.1 + 446.pro1 | https://posit.co Posit Software, PBC | |
| Cytoscape v3.7.2 | https://cytoscape.org Cytoscape Consortium | |
| **Other** | | |
| RevertAid First Strand cDNA Synthesis Kit | Thermo Fisher | Cat# K1622 |
| Serum Triglyceride Determination Kit | Sigma-Aldrich | Cat# TR0100 |
| CHO Kit | Zhongsheng beikong | Cat# 000180 |
| ALT/GPT Kit | Zhongsheng beikong | Cat# 000000010 |
| AST/GOT Kit | Zhongsheng beikong | Cat# 000000020 |
| BCA Protein Assay Kit Pierce | Thermo-Pierce | Cat# 23227 |
| SuperReal PreMix Plus (SYBR Green) | TIANGEN | Cat# FP205 |
| Dual-Luciferase Reporter Assay System | Promega | Cat# E1910 |
| ClonExpress Ultra One Step Cloning Kit V2 | Vazyme | Cat# C116-02 |

## Experimental animals

All animal care and experimental procedures were approved by the Animal Care and Use Committee at Peking University (protocol IMM-ChenXW-1), which has been authorized by the Association for Assessment and Accreditation of Laboratory Animal Care (AAALAC). Mice were housed under specific pathogen-free (SPF) conditions with a controlled environment at 22 °C under a 12-h light/12-h dark cycle, and free access to food and water, unless the requirement for food withdrawal in specified experiments. All mice used in this study were bred on a C57BL/6 J background. Mice carrying a floxed allele of *Sec16b* were generated using CRISPR/Cas9 technology by GemPharmatech. *Sec16b*[fl/+] mice were crossed with wild-type C57BL/6J mice for 3 rounds. To generate liver-specific *Sec16b* knockout mice (*Sec16b* LKO), liver-specific *Sec16b* heterozygous (*Sec16b* LHets), 6-week-old *Sec16b*[fl/fl] and *Sec16b*[fl/+] mice were intravenously injected with $4 \times 10^{11}$ viral particles (VP) of AAV8-TBG-Cre, respectively. Age and gender-matched littermates receiving the same dose of AAV8-TBG-Luciferase were allocated as controls. Experiments were conducted 4 weeks post-injection. For the pathological hyperlipidemia model, 6-week-old *Sec16b*[fl/fl] mice were intravenously co-injected with AAV8-TBG-Cre (for liver-specific knockout) or AAV8-TBG-Luciferase (control), along with AAV8-TBG-hPCSK9[D374Y] (Bjorklund et al, 2014) and maintained on a Western diet for 16 weeks. For fasting-refeeding experiments, 6-week-old mice with specified genotypes were randomly assigned to a fasting group (16 h) or a fasting-refeeding group (16 h fasting followed

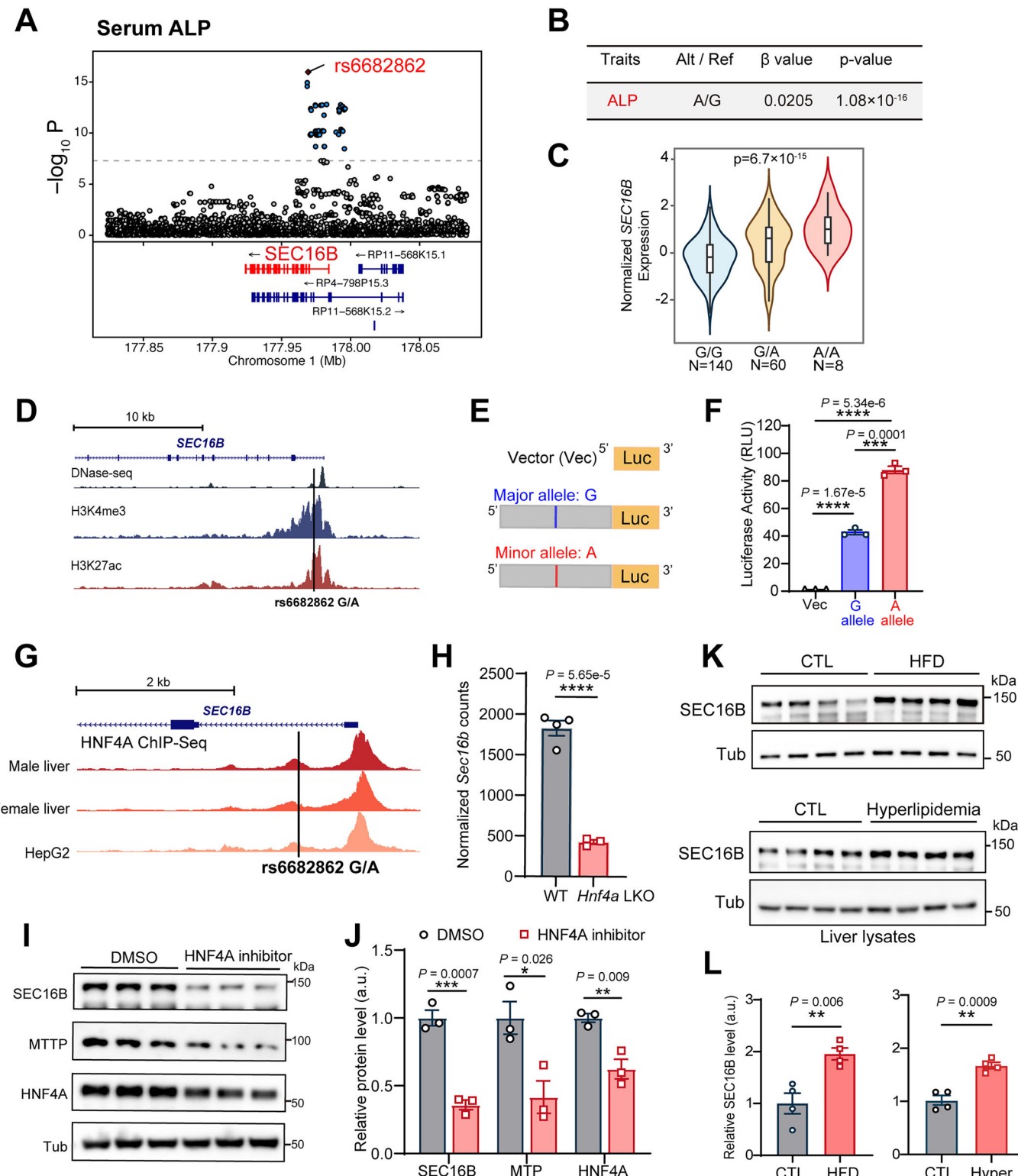

**A** Serum ALP

**B**

| Traits | Alt / Ref | β value | p-value |
|--------|-----------|---------|---------|
| ALP | A/G | 0.0205 | 1.08×10⁻¹⁶ |

**C**

**D**

**E**

**F**

**G**

**H**

**I**

**J**

**K**

**L**

by 12-h food access with a high-fructose diet), with free access to water throughout the experimental periods.

To overexpress SEC16B in the liver, 6-week-old wild-type C57BL/6 J mice were intravenously injected with AAV8-TBG-FLAG-SEC16B at $2 \times 10^{10}$ VP/mice (Lower dose) or $6 \times 10^{10}$ VP/mice (Higher dose). To overexpress ^ΔCCD^SEC16B in the liver,

6-week-old wild-type C57BL/6 J mice were intravenously injected with AAV8-TBG-FLAG-^ΔCCD^SEC16B at $5 \times 10^{11}$ VP/mice (Lower dose) or $2.5 \times 10^{12}$ VP/mice (Higher dose). Age- and gender-matched littermates injected with a respective higher dose of AAV8-TBG-GFP served as controls for both groups. All experiments were performed 2 weeks post-injection.

◄ **Figure 6.  HNF4A-mediated *SEC16B* regulation associates with human hepatic phenotypes.**

(A) Region plot of *SEC16B* associated with plasma alkaline phosphatase (ALP) levels in humans. Data are obtained from UKBB. Red line indicates the threshold of genome-wide significance ($P = 5 \times 10^{-8}$). See "Methods" section "Association analysis of genetic variants in SEC16B with lipid traits". (B) Summary of the genome-wide association data between the leading SNP rs6682862 and plasma ALP. Data are obtained from UKBB. See Methods section "Association analysis of genetic variants in SEC16B with lipid traits". (C) The association between different alleles of rs6682862 and *SEC16B* gene expression in the liver. Data are obtained from the GTEx web portal V8. Violin plot shows the probability density of data at normalized gene expression levels for each genotype, with the center line representing the median and the box spanning the interquartile range (IQR). The whiskers extend to 1.5 times the IQR. The *P* value was tested by a covariate-adjusted linear regression model. (D) Genome active regions marked by DNase I hypersensitive, H3K27ac chip-seq, and H3K4me3 chip-seq signals in the intronic regions adjacent to the rs6682862 SNP. (E) Schematics of the luciferase reporter vector with an inserted 150-bp intronic segment flanking the rs6682862 SNP upstream. (F) rs6682862 A-allele increases the promoter activity of the genome sequence flanking the SNP. Data are shown as mean ± SEM. $n = 3$ biologically independent replicates for each group. ***$P < 0.001$; ****$P < 0.0001$ (two-tailed Student's *t* test). (G) HNF4A chip-seq signals from human liver samples and a hepatoma cell line in the promoter regions of *SEC16B*. Data are obtained from ENCODE and plotted with the UCSC Genome Browser. (H) *Sec16b* mRNA expression in the livers of *Hnf4a* liver knockout and control mice. Data are obtained from GEO GSE246053. $n = 4$ WT and 3 *Hnf4a* LKO mice, respectively. Data are shown as mean ± SEM. ****$P < 0.0001$ (two-tailed Student's *t* test). (I) IB analysis of murine primary hepatocytes treated with DMSO control, or BI6015, a HNF4A inhibitor ($n = 3$ biological replicates). (J) Quantification of the protein expression level in (I). Data are shown as mean ± SEM. **$P < 0.01$; *$P < 0.05$ (two-tailed Student's *t* test). a.u. arbitrary units. (K) IB analysis of SEC16B in liver lysates from wild-type mice fed with chow diet (CTL) and high fat diet (HFD), or from wild-type mice fed with chow diet (CTL) and mice receiving AAV-hPCSK9 fed with western diet (Hyperlipidemia, Hyper). Representative of three independent biological replicates is shown. (L) Quantification of IB signals in (K). Data are shown as mean ± SEM. **$P < 0.01$ (two-tailed Student's *t* test). a.u. arbitrary units. Source data are available online for this figure.

For manganese supplement experiments, *Sec16b* LKO mice and wild-type control mice were divided into subgroups and administered with 0 mg/kg or 20 mg/kg $Mn^{2+}$ daily by oral gavage, respectively. Plasma lipid profiles were assessed one month after the initiation of gavage.

## Cell culture

Huh7 cells (JCRB Cell Ban, JCRB0403; RRID, CVCL_0336), HEK293T (ATCC, CRL-3216; RRID, CVCL_0063), and HEK293A cells (Thermo Fisher, Cat# R70507) were obtained with authentication by STR and morphology. All the cell lines used were confirmed to be free of mycoplasma contamination, tested by the Vazyme Mycoblue Mycoplasma Detector. Mouse primary hepatocytes were isolated from male mouse livers with the indicated genotypes. All cells were cultured in DMEM with 10% fetal bovine serum (FBS) and 1% penicillin/streptomycin at 37 °C under an atmosphere of 5% $CO_2$.

## Antibody preparation

To generate antibodies against SEC16A or SEC16B, peptides containing aa 600T-650A (Uniprot: O15027-1) of SEC16A or aa 687D-781G (Uniprot: Q91XT4) of SEC16B were purified and injected to rabbits as immunogens, respectively. The antibodies were generated by Proteintech.

## Gene co-expression analysis and tissue expression distribution correlation analysis

Tissue expression profile data of human liver and small intestine were obtained from Genotype-Tissue Expression (GTEx) Project (https://gtexportal.org/, GTEx V8). Genes co-expressed with *APOB* and *MTTP* were identified using a Pearson correlation coefficient (r) threshold of >0.6, and both co-expressed in the liver and intestine were selected for functional annotation based on Gene Ontology (GO) enrichment analysis and manual annotation. The intersection of the four datasets was plotted using a Venn diagram package (version 1.7.3). Network was performed by STRING database(https://string-db.org) and visualized in Cytoscape (version 3.7.2).

For tissue expression distribution correlation analysis, expression profiles across multiple tissues were obtained from the Human Protein Atlas (HPA, https://www.proteinatlas.org/). Gene expression data corresponding to ER to Golgi transport processes were extracted and normalized to [0,1] range via min-max scaling across tissues, then calculated pairwise Pearson correlation analysis. And the intergenic Pearson correlation coefficients (r) were visualized using a heatmap. Consensus normalized expression (nTPM) across human major tissues for *SEC* genes and apolipoprotein genes were obtained from HPA and GTEx, followed by proportional normalization across tissues. Heatmap was generated using heatmap R package (version 1.0.12).

## Single-cell transcriptome data analysis

Single-cell transcriptome data of hepatocytes and non-parenchymal cells in the liver of healthy and NAFLD mice (GSE166504) were downloaded. A total of 6 healthy mouse liver data were selected, each including two types of samples: isolated primary hepatocytes and non-parenchymal cells. The following analysis including sample integration, cell clustering and cell type annotation was performed using Seurat package (version 4.1.1) as previously described. The expression of single-genes in different cell types was visualized using uniform manifold approximation and projection (UMAP). The dot plot was performed to visualize the *SEC* genes expression pattern across multi-cell types in the liver. Relative percentage was defined as the ratio of expression in each cell types to all expressed cells. Normalized expression meant that the average expression value of each cell types was extracted and normalized to [0,1] range via min-max scaling.

The R package AUCell (version 1.24.0) was used to calculate the COPII signature score across each cell type. The COPII gene sets were defined as *Sar1a*, *Sar1b*, *Sec23a*, *Sec23b*, *Sec24a*, *Sec24b*, *Sec24c*, *Sec24d*, *Sec13*, *Sec31a*, *Sec16a*.

## Immunoblotting

Mouse liver samples were homogenized with lysis buffer (50 mM Tris, pH 7.5, 150 mM NaCl, 1% NP-40, 10% glycerol) supplemented with protease inhibitors at 4 °C with a tissue homogenizer at 35 Hz for 10 min, followed by centrifugation at 4 °C at 10,000 rpm

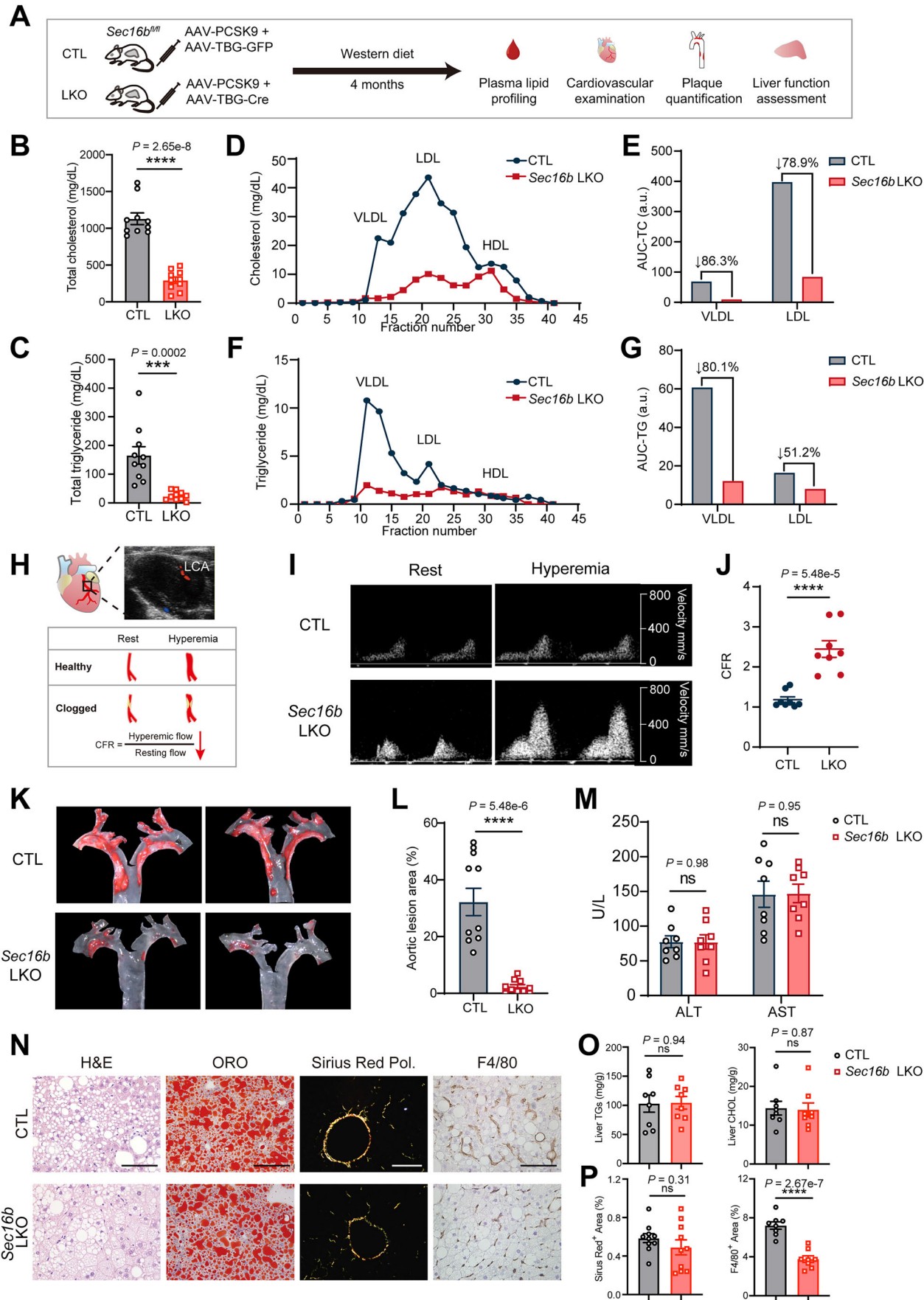

Figure 7.   Therapeutic potentials of SEC16B targeting in dyslipidemia and ASCVD.

(A) Schematics of targeting *Sec16b* for combating hyperlipidemia in mice receiving AAV-TBG-hPCSK9 and western diet (WD). (B, C) Loss of hepatic *Sec16b* decreases pathological plasma cholesterol levels (B) and plasma triglyceride levels (C) in mice from (A). Data are shown as mean ± SEM. $n = 10$ for each group. ***$P < 0.001$; ****$P < 0.0001$ (two-tailed Student's *t* test). (D) Cholesterol measurement from FPLC-mediated lipoprotein profiling with pooled plasma from mice in (B). (E) Quantification of APOB-containing VLDL and LDL cholesterol levels in (D). AUC area under the curve. (F) Triglyceride measurement from FPLC-mediated lipoprotein profiling as in (D). (G) Quantification of APOB-containing VLDL and LDL triglyceride levels in (F). AUC area under the curve. (H) Upper: Blood flow in the left coronary artery (LCA) visualized by Color Doppler. Lower: Schematics illustrating the coronary flow reserve (CFR) in healthy or clogged coronary arteries. (I) Loss of hepatic *Sec16b* improves the coronary flow reserve (CFR) in atherogenic mice from (A). Representative color Doppler view of velocity spectra from the left coronary artery at rest and under 2.5% isoflurane induced hyperemia. (J) Quantification of the CFR in (I). CFR was estimated by the ratio of the diastolic flow velocity during hyperemia to that during rest. Data are shown as mean ± SEM. $n = 8$ for each group. ****$P < 0.0001$ (two-tailed Student's *t* test). (K) Representative images of atherosclerotic plaques in mice from (A). (L) Quantification of atherosclerotic plaques on the thoracic aorta from CTL and *Sec16b* LKO in (K). $n = 10$ for each group. Data are shown as mean ± SEM. ****$P < 0.0001$ (two-tailed Student's *t* test). (M) Plasma ALT and AST levels of mice from (A). Data are shown as mean ± SEM. ns, no significance. $n = 8$ for each group. ns, no significance (two-tailed Student's *t* test). (N) Representative H&E, ORO, Sirius red staining (Sirius Red Pol.), and F4/80 IHC of liver sections from mice in (A). Scale bars = 100 μm, except 200 μm for ORO. (O) Quantification of TG and total cholesterol contents in lipid extracts of WT control and SEC16B-deficient mouse livers. $n = 8$ for each group. Data are shown as mean ± SEM. ns, no significance (two-tailed Student's *t* test). (P) Quantification of Sirius red staining and F4/80 signals in (N). $n = 8$ for each group. Data are shown as mean ± SEM. *$P < 0.05$; ****$P < 0.0001$ (two-tailed Student's *t* test). Source data are available online for this figure.

for 10 min. The pellet and the upper oily layer were discarded, and the middle clear liquid portions were collected. Protein concentrations in lysates were adjusted according to BCA assay-based quantification. Samples were mixed with 4×SDS loading buffer and subjected to SDS-PAGE. Subsequently, proteins were transferred to NC membrane prior to immunoblotting with the indicated antibodies.

For IB analysis of plasma proteins, 3 μL of plasma was mixed with 120 μL of lysis buffer supplemented with protease inhibitors, followed by 4×SDS loading buffer, and subjected to SDS-PAGE.

## RNA-seq analysis

Total RNA was extracted from liver samples using Trizol reagent (Thermo Fisher) following the manufacturer's instructions. mRNA-seq was performed using the Illumina NovaSeq 6000 platform (Novogene, Beijing). Data analysis was carried out as the following pipeline: Raw sequencing data in FASTQ format (150 bp paired-end reads) were subjected to quality assessment using FastQC (version 0.11.9). Quality-passed reads were aligned to the mouse reference genome (mm10) using HISAT2 (version 2.2.1) with default parameters, retaining only uniquely mapped reads for downstream analysis. Gene raw counts were obtained by HTSeq (version 0.12.4) and normalized as Fragments Per Kilobase of exon per Million mapped reads (FPKM) using StringTie (version 2.1.5). Differential gene expression analysis was performed using DESeq2 (version 1.42.0). For functional annotation, Gene Ontology (GO) enrichment analysis was conducted using the Metascape platform (http://metascape.org), focusing on fasting/refeeding responsive genes with altered expression in *Sec16b*-deficient livers. Pathway analysis of genes with differential expression between control and *Sec16b*-deficient livers under refeeding conditions was performed with Gene Set Enrichment Analysis (GSEA) using clusterProfiler R package (version 4.10.0). Data visualizations were generated using ggplot2 (version 3.3.5) in R.

## Gene gain/loss tree

*SEC16B* and *APOB* gene family evolution was analyzed using the Gene Gain/Loss Tree in the Ensembl Genome Browser. This analysis inferred ancestral gene duplications and losses across a wide evolutionary spectrum from yeast to human.

## Preparation of adeno-associated virus (AAV)

AAVs were packaged and produced in HEK293T cells as previously described (Wang et al, 2021b). In brief, ten 15-cm dishes of 293T cells were transfected with 70 μg of AAV shuttle plasmid, 70 μg of Rep/Cap (RC2/8) plasmid, and 200 μg of helper plasmid using polyethyleneimine (PEI). Cells were dislodged and collected at 48 h after transfection. AAV purification was carried out using the Optiprep density gradient centrifugation. The virus titer was determined by qPCR of viral genome DNA using custom SYBR Green assays.

## Plasma analysis

After 16-h fasting, blood was collected from mice tail tips using heparin-coated capillaries. Plasma was obtained by centrifuging twice at 6000 rpm for 5 min at 4 °C. Commercial kits (000180 of Zhongsheng beikong and TR0100 of Sigma, respectively) were used to measure triglyceride and total cholesterol levels according to the manufacturer's protocol. ALT/GPT and AST/GOT were measured using commercial kits (Zhongsheng beikong) according to the manufacturer's protocol.

For fast protein liquid chromatography (FPLC), a Superose 6 column was used to fractionate the pooled plasma samples from mice with the same genotype. The samples were fractionated into 50 fractions with 300 μL/tube at a flow rate of 0.5 mL/min, prior to triglyceride and cholesterol measurements.

## Plasma proteomics and data analysis

Protein samples were solubilized to a concentration of 3 mg/mL in 100 mM TEAB buffer containing 6 M urea. The samples were then reduced with 10 mM dithiothreitol (DTT) at 37 °C for 30 min and alkylated with 20 mM iodoacetamide (IAA) at 35 °C for 30 min in the dark. Following alkylation, the urea concentration was diluted to 2 M with 100 mM TEAB. The proteins were digested overnight at 37 °C with 0.6 μg of trypsin under constant agitation. The resulting peptides were analyzed using a Thermo Orbitrap Fusion Lumos mass spectrometer, and the raw MS data were processed using Proteome Discoverer software (version 2.2). Protein quality control was performed by retaining only proteins identified by at least one unique peptide and detected across all samples.

Differential expression analysis was conducted using the Limma package (version 3.58.1), with significance defined by a $P$ value < 0.05 and a fold change (FC) > 1.5.

## VLDL secretion experiment

Mice were fasted for 16 h and intravenously administered with tyloxapol at a dose of 500 mg/kg. Blood samples were collected at the indicated time points, and the plasma was separated by centrifugation. Triglyceride measurements were performed as described above.

## Histology

For Hematoxylin and Eosin (H/E) staining and Sirius red staining, liver samples were fixed 48 h in 4% PFA followed by paraffin embedding. For IHC, paraffin-embedded liver samples were sequentially sliced, deparaffinized, rehydrated, and subjected to antigen retrieval. The sections were blocked with 10% goat serum for 30 min and incubated with F4/80 primary antibody diluted in blocking buffer overnight at 4 °C, followed by HRP-conjugated secondary antibodies for 2 h at room temperature and DAB labeling.

For Oil Red O staining, liver samples were embedded in optimal cutting temperature (OCT) followed by snap freezing. Frozen tissues were cryo-sectioned at 10 μm and stained with Oil red O solution following the manufacturer's protocol.

## Immunoprecipitation

Liver or cell lysates were incubated with an antibody specified in the legend and Protein A beads at 4 °C for 2 h. After 4–6 washes with NP-40 buffer, immune complexes were eluted using 1.5× SDS sample buffer and subjected to SDS-PAGE.

## Immunostaining and confocal microscopy

Primary hepatocytes or cultured cells on glass slides were washed by warm PBS twice, fixed by methanol for 5 min and rehydrated in PBS for 10 min. Then, cells were blocked by 2.5% goat serum in PBS for 30 min at room temperature, followed by incubation with primary antibody overnight at 4 °C. Subsequently, cells were washed by PBS and incubated with Alexa-fluor conjugated secondary antibodies for 1 h at room temperature. After that, cells on glass slides were washed by PBS and mounted by coverslips. Zeiss spinning disk microscopy or Zeiss 980 laser scanning microscope with a ×63 oil immersion objective was used to perform imaging. ImageJ Fiji was used to quantify fluorescent signals.

For live-cell imaging, cells were plated on 35 mm glass-bottom dishes (Nest) one day before transfection. About one hour prior to imaging, the medium was replaced with imaging medium. Live-cell imaging was performed using a ×63 oil immersion objective (NA 1.4) on a Zeiss Spinning Disk microscope equipped with an incubator providing a humidified atmosphere with 5% $CO_2$ at 37 °C. Fluorescence intensity and areas of condensates are quantified by ImageJ Fiji and analyzed by Prism.

## Fluorescence recovery after photobleaching (FRAP) analysis

For FRAP experiments, a defined region within the sample was subjected to photobleaching using a 561-nm or 488-nm laser on a Zeiss LSM980 confocal microscope. Image acquisition was initiated immediately following bleaching with 1s-intervals. The fluorescence signals were normalized to the pre-bleach baseline and quantified as the mean intensity within the region of interest using ImageJ Fiji.

## APOB secretion assay

The primary mouse hepatocytes were plated in 6 cm dishes and switched to serum-free DMEM after 6 h. Following 5 μM TPEN treatments for 30 min, hepatocytes were incubated with $MnCl_2$ at the specified concentrations for 8 h. The medium was collected and centrifuged to remove cell debris. Secreted proteins in the medium were analyzed using immunoblotting.

## In vitro COPII-vesicle reconstitution

In vitro reconstitution of COPII vesicles was performed as previously described (Chen et al, 2013; Huang et al, 2021), with some modifications. Semi-intact cells from HEK293A cells expressing RFP or RFP-$^{\Delta\Delta ACE}$SEC16B were prepared freshly by permeabilizing cells with 40 ug/ml digitonin in B88 (20 mM Hepes, pH 7.2, 250 mM sorbitol, 150 mM potassium acetate, and 2 mM magnesium acetate). Each reaction, in a final volume of 100 μL, contained ATP regeneration system (1 mM ATP, 40 mM creatine phosphate and 0.2 mg/mL creatine phosphokinase), GTP (0.2 mM), cytosol from mouse liver at a final concentration of 1 mg/mL protein, and semi-intact cells containing 20 μg of proteins in B88. The reaction mixtures were incubated in siliconized 1.5 mL microcentrifuge tubes for 60 min at 30 °C, and terminated by chilling on ice. After centrifugation at 14,000 × $g$ for 10 min to remove medium-speed pellet (donor membrane), 75 μL of supernatant was taken and centrifuged at 55,000 RPM at 4 °C for 30 min in a Beckman TLA100 rotor. The pellets (vesicle fractions) were used for analysis by immunoblotting.

## Electron microscopy

For liver samples, tissues were processed as previously described. Briefly, samples were pre-fixed in 2.5% glutaraldehyde, 0.8% paraformaldehyde in PB, immersed in 0.1 M imidazole in PB, post-fixed with 2% osmium tetroxide in PB, and stained with 1% uranyl acetate. Subsequently, the samples were gradient dehydrated and embedded in epoxy resin. Samples were sectioned at about 60-nm thick using Leica EM UC7 and placed on copper grids. An FEI Tecnai G2 20 Twin electron microscope fitted with a CMOS camera XAROSA (EMSIS; Germany) was used to record images.

## Negative staining

Grids were incubated on 3.33 μl of samples containing isolated lipoproteins samples for 1 min on ice. Excess solutions were removed using a filter paper. The grids were then floated on three

drops of 1% Ua (20 µl). Excess Ua solution was removed using a filter paper. The grids were kept in the dark and air-drying at 4 °C for 10 min. Images were recorded on a FEI Tecnai G2 20 Twin electron microscope fitted with an Eagle™ 4k CCD digital camera (FEI; USA).

## Association analysis of genetic variants in *SEC16B* with lipid traits

GWAS analysis of SNPs in the *SEC16B* gene and plasma lipid levels was performed using LocusZoom (https://my.locuszoom.org/), based on published GWAS data published (Graham et al, 2023). SNP rs6682862 genotypes and normalized gene expression data in the liver were retrieved from the GTEx database, and the violin plot of normalized SEC16B expression levels in populations with respective rs6682862 genotypes was drawn by ggplot2 (version 3.4.4). Epigenetic modifications and HNF4α ChIP-seq data of the regions adjacent to rs6682862 were obtained from the ENCODE database. GWAS analysis for *SEC16B* and serum alkaline phosphatase (ALP) levels was using data obtained from a large cross-population meta-analysis including 344,292 individuals of European ancestry and 118,886 individuals of East Asian ancestry (Sakaue et al, 2021). For each genetic variant, association statistics were estimated using linear regression models under an additive genetic model. The reported *P* values represent the statistical significance of the association between genotype dosage and serum ALP levels.

## Dual-luciferase reporter system

To validate the function of rs6682862 region, 150 bp segments surrounding the SNP were cloned into the firefly luciferase vector using standard restriction-enzyme cloning. The firefly luciferase was used as the experimental group, and the Renilla luciferase was used as the internal reference group. In brief, Huh7 cells were plated in a six-well plate with a seeding density of about 30%, and transfected with 1 µg of firefly luciferase plasmid with SNP alleles specified in the figure, 0.5 µg of Renilla luciferase plasmid using polyethyleneimine (PEI). Twenty-four hours post transfection, the cells were lysed and subjected to the firefly and renilla luciferase activity measurement using the Dual-Luciferase Reporter Assay System (Promega) according to the manufacturer's protocol.

## Analysis of atherosclerosis

*Sec16b*<sup>fl/fl</sup> mice receiving AAV-hPCSK9<sup>D374Y</sup> with AAV-TBG-GFP or without AAV-TBG-CRE were fed with western diet for 4 months. Mice were sacrificed after anesthetized, followed by perfusion with PBS and 4% paraformaldehyde (PFA) sequentially. Liver samples were collected for histology analysis and lipid content measurements. The aortas are isolated and stained with Oil red O, photographed with Olympus stereo microscope equipped with a SCMOS camera. The lesion areas in thoracic aortic were quantified with ImageJ.

Atherosclerotic lesion analysis at the aortic root was performed following established methodology (Lavillegrand et al, 2024; Martinez-Hervas et al, 2014; Takaoka et al, 2024; Zhang et al, 2023). In brief, collected hearts were fixed overnight in 4% PFA, cryoprotected in PBS-30% sucrose solution, embedded in OCT compound, and stored at −80 °C. Sequential 10-µm cross-sections of the aortic sinus containing valve leaflets were prepared using a cryostat(Centa et al, 2019). Histological analyses were performed using H&E staining for plaque area quantification, Oil Red O for lipid deposition, and Masson's Trichrome for assessment of collagen content and necrotic core area (defined as acellular regions). Quantification of fibrous cap thickness was performed by measuring the layer of cells and extracellular matrix overlying necrotic cores at five different locations, with the results expressed as the mean thickness per necrotic core.

For immunohistochemical analysis, cryosections were equilibrated to room temperature for 20 min and rehydrated in PBS before being blocked with 10% goat serum containing 0.1% Triton X-100 for 30 min. Sections were then incubated with anti-CD68 primary antibody in blocking buffer at 4 °C overnight, followed by appropriate HRP-conjugated secondary antibody incubation for 1 h at room temperature and subsequent DAB labeling.

All images were photographed using an Olympus BX51 Microscope and then quantified using Image-Pro Plus 6.0 software.

## Measurement of coronary flow reserve

Mice were anesthetized with 3% isoflurane in oxygen in a closed chamber, then secured in a supine position on ECG electrodes attached to a heated procedure board maintained at 37 °C. Once positioned, the mice were maintained under full sedation using 1% isoflurane administered via a nose cone. Electrocardiographic measurements were obtained using a high-frequency ultrasound system (Vevo 3100), equipped with a 70 MHz transducer. Blood flow in the left coronary artery (LCA) was assessed using pulsed-wave color Doppler.

Resting blood flow was recorded at the 1% isoflurane dose, while hyperemic flow was induced with a 2.5% isoflurane dose. Coronary flow reserve (CFR) was calculated as the ratio of hyperemic flow to resting flow.

## Quantification and statistical analysis

Quantifications of IB signals, fluorescence signals in imaging, and plaque area were performed using ImageJ. No statistical methods were used to predetermine sample size. Statistical analysis was performed using GraphPad Prism 8. Statistical analysis was conducted using Student's two-tailed *T* tests, or One-Way ANOVA multiple comparisons test, as being specified in the figure legends. The results of *P* value < 0.05 were considered significant, as being indicated in the figure legends. Experimental results shown are representative of at least three biologically independent experiments. Mouse experiments were randomized. Imaging and histology were evaluated in a blinded fashion.

# Data availability

This study includes no data deposited in external repositories.

The source data of this paper are collected in the following database record: biostudies:S-SCDT-10_1038-S44318-026-00754-8.

# Peer review information

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

## Acknowledgements

The work is supported by National Science Foundation of China (NSFC) grants, 32422037 (to XW), 32125021, 92254308, 92357307 (to X-WC), 32571360 (to XW), 32400985 (to LL); National Key R&D Program grant no. 2024YFA1802802 and 2021YFA0804802 (to X-WC); Beijing Outstanding Young Scientist Program JWZQ20240102002 (to X-WC); Beijing Advanced Center of RNA Biology (BEACON) at Peking University; and the Yunnan Provincial Science and Technology Project at Southwest United Graduate

School 202302AO370004 (to X-WC). The authors thank the Core Facilities of Life Sciences at Peking University in Beijing, China for assistance with EM, and Dr. Wanqiu Ding from the Bioinformatics Core Facility of College of Future Technology at Peking University in Beijing, China for assistance with bioinformatic analysis.

## Author contributions

**Xiao Wang**: Conceptualization; Formal analysis; Funding acquisition; Validation; Investigation; Visualization; Writing—original draft; Writing—review and editing. **Yating Hu**: Formal analysis; Investigation; Visualization; Writing—original draft; Writing—review and editing. **Lu Liu**: Funding acquisition; Investigation; Visualization. **Runze Huang**: Investigation. **Yawei Wang**: Investigation. **Bing Liu**: Investigation. **Yifei Zhao**: Investigation. **Yuangang Zhu**: Investigation. **Jia Lv**: Investigation. **Liang Liu**: Investigation. **Huimin Wang**: Investigation. **Lingzhi Wu**: Investigation. **Xinxuan Xu**: Investigation. **Yaxin Li**: Investigation. **Guanlin Wang**: Resources. **Xiao-Wei Chen**: Conceptualization; Resources; Supervision; Funding acquisition; Investigation; Writing—original draft; Project administration; Writing—review and editing.

Source data underlying figure panels in this paper may have individual authorship assigned. Where available, figure panel/source data authorship is listed in the following database record: biostudies:S-SCDT-10_1038-S44318-026-00754-8.

## Disclosure and competing interests statement

The authors declare no competing interests.

# Expanded View Figures

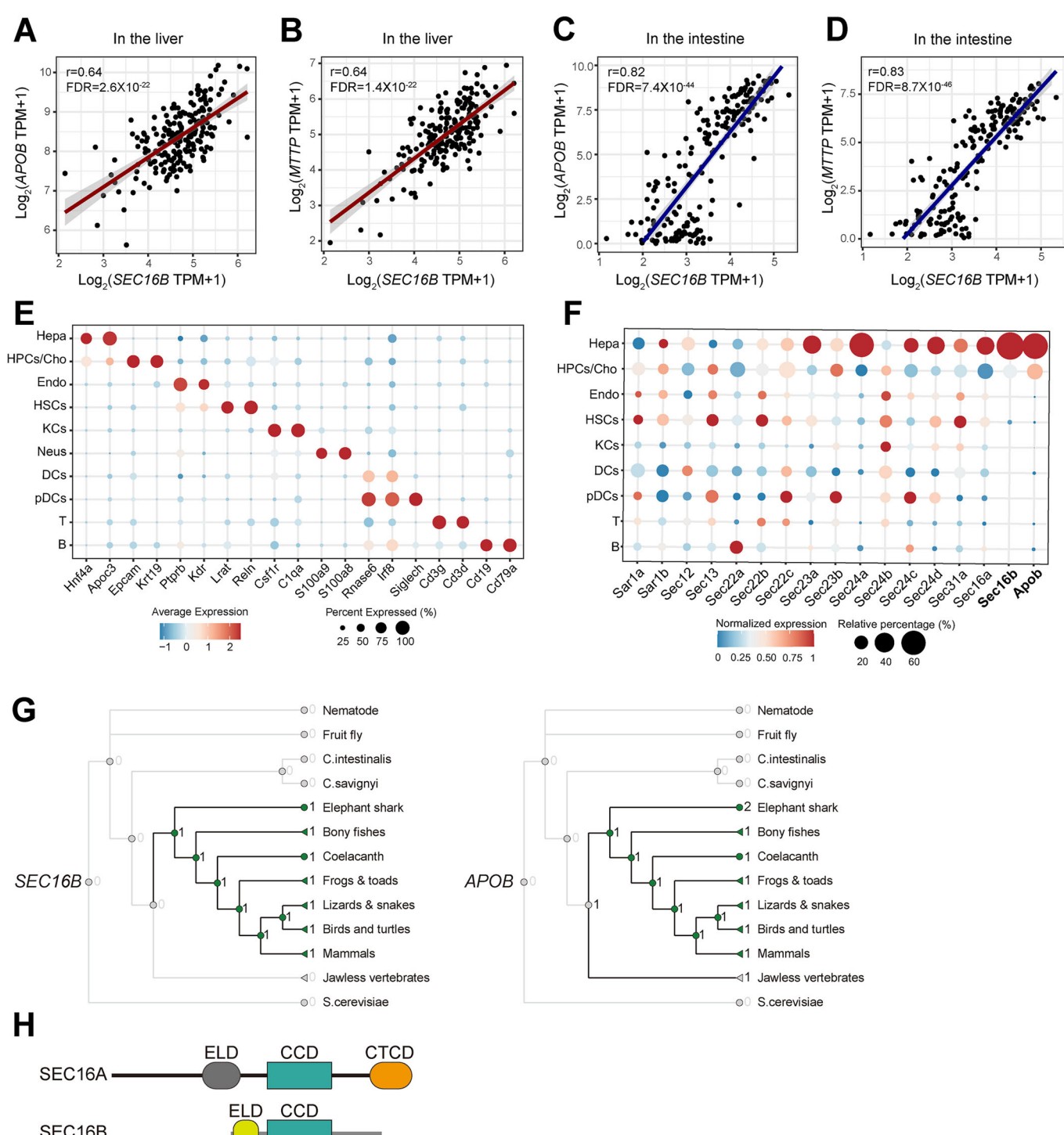

**Figure EV1.  Identification of a tissue-selective *SEC* gene *SEC16B*.**

(A, B) Correlation analysis of *SEC16B* with *APOB* expression (A) and *MTTP* expression (B) in human liver. (C, D) Correlation analysis of *SEC16B* with *APOB* expression (C) and *MTTP* expression (D) in human intestine. (E) Bubble heatmap of marker gene expression for each identified cell cluster in Fig. 1F. Hepa hepatocytes, hPCS/Cho hepatic progenitor cells/cholangiocyte, Endo endothelial cells, HSCs hepatic stellate cells, KCs kupffer cells, Neus neutrophils, DCs dendritic cells, pDCs plasmacytoid dcs, T T cells, B B cells. (F) Bubble heatmap of *Sec* gene expression patterns across cell types from Fig. 1F. Sizes represent the percentage of cells expressing the indicated gene within each cluster. Color represents the normalized expression levels in each cell cluster. The expression of *Sec31b* was not detected. Neutrophils were excluded from the analysis due to low cell counts (*n* < 10) to avoid potential statistical bias. (G) Gene gain and loss events of *SEC16B* and *APOB* by using the Ensembl browser. (H) Domain schematics of SEC16A and SEC16B. ELD ER localizing domain, CCD central conserved domain, CTCD C-terminal conserved domain.

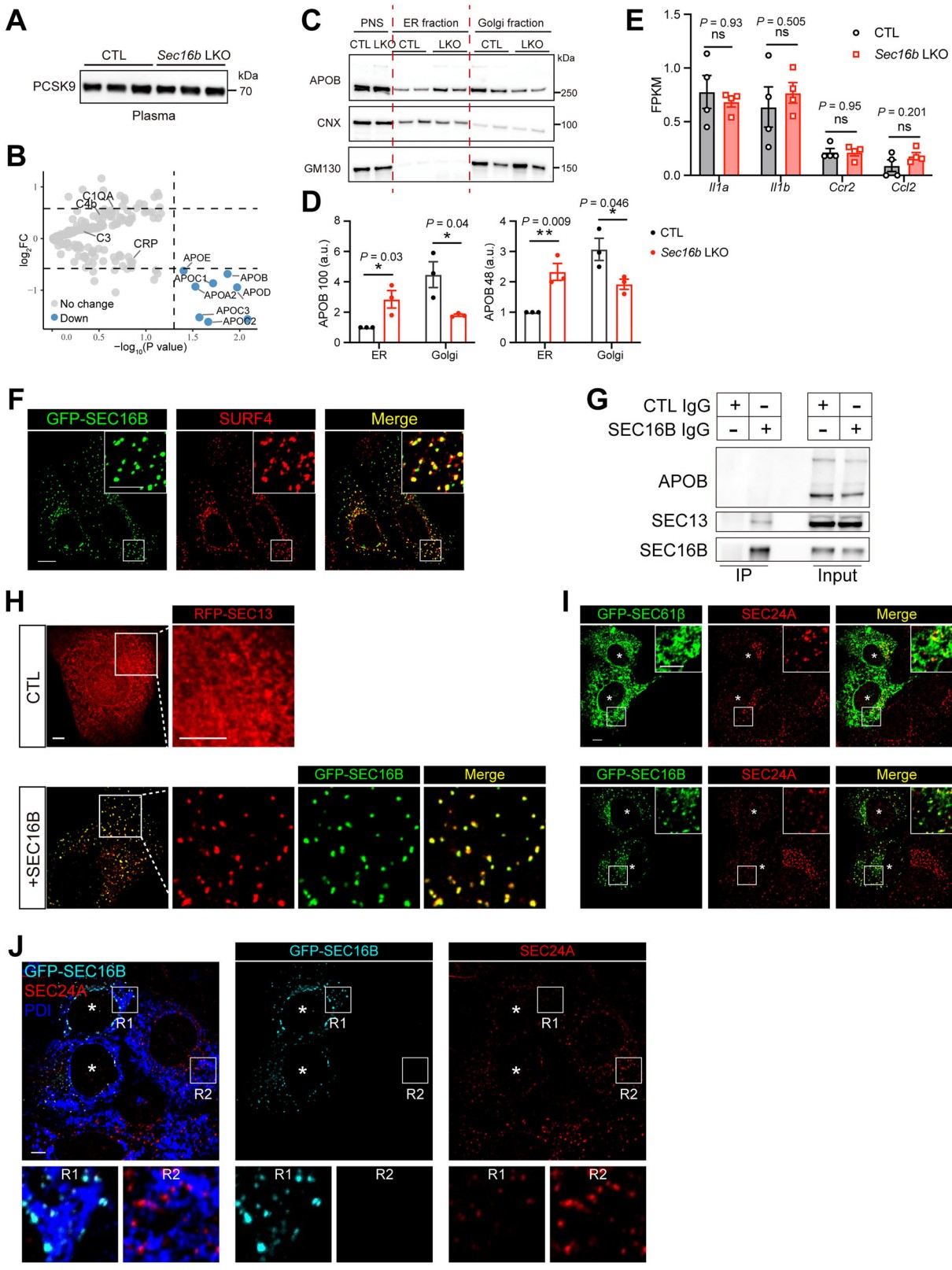

◀ **Figure EV2. SEC16B selectively regulates hepatic lipid secretion.**

(A) IB analysis of PCSK9 in plasma samples from CTL and *Sec16b* LKO mice. Representative of three independent experiments is shown. (B) Volcano plot of quantitative proteomics in plasma samples from CTL and *Sec16b* LKO mice. Significantly altered proteins with |FC| > 1.5 and adjusted *P* value < 0.05 are colored in blue. *n* = 3 for each genotype. (C) IB analysis of ER-enriched fractions and Golgi-enriched fractions isolated from CTL and *Sec16b*-KO mouse livers. Representative of three independent experiments is shown. (D) Quantification of IB signals in (C). *n* = 3 biological independent replicates for each group. Data are shown as mean ± SEM. *$P$ < 0.05; **$P$ < 0.01 (two-tailed Student's *t* test). (E) Analysis of inflammatory gene expression using hepatic RNA-seq data from Fig. 3G with CTL and *Sec16b* LKO mice under fasting conditions. *n* = 4 mice for each group. Statistic was analyzed using the Wald test in DESeq2 with adjustment by Benjamini-Hochberg. Data are shown as mean ± SEM. ns, no significance. (F) Co-localization of SEC16B with SURF4. Huh7 cells transfected with GFP-SEC16B were fixed and stained with an anti-SURF4 (red) antibody, prior to confocal microscopy. Scale bars = 5 μm. (G) Co-IP of endogenous SEC16B and SEC13 in mouse liver samples. Liver lysates were subjected to anti-SEC16B or CTL IgG IP, followed by SDS-PAGE and IB with the indicated antibodies. (H) SEC16B promotes membrane localization of outer coat proteins. Huh7 cells transfected with HA-SEC31A and RFP-SEC13, plus empty vector (upper row) or GFP-SEC16B WT (lower row) were analyzed by confocal microscopy. Scale bars = 5 μm. (I) SEC61β overexpression exhibits little effect on COPII coalescence compared to neighboring cells. Huh7 cells transfected with GFP-SEC61β (upper) or GFP-SEC16B (lower) were fixed and stained with an anti-SEC24A (red) antibody, prior to confocal microscopy. Scale bars = 5 μm. Asterisks, positive cells with GFP-SEC61β or GFP-SEC16B expression. (J) Similar localization on the ER surface of SEC24A between SEC16B-expressing and neighboring control cells. Huh7 cells transfected with GFP-SEC16B were fixed and co-stained with anti-SEC24A (red) and anti-PDI (blue) antibodies, prior to confocal microscopy. Scale bars = 5 μm. Asterisks, positive cells with GFP-SEC16B expression. Source data are available online for this figure.

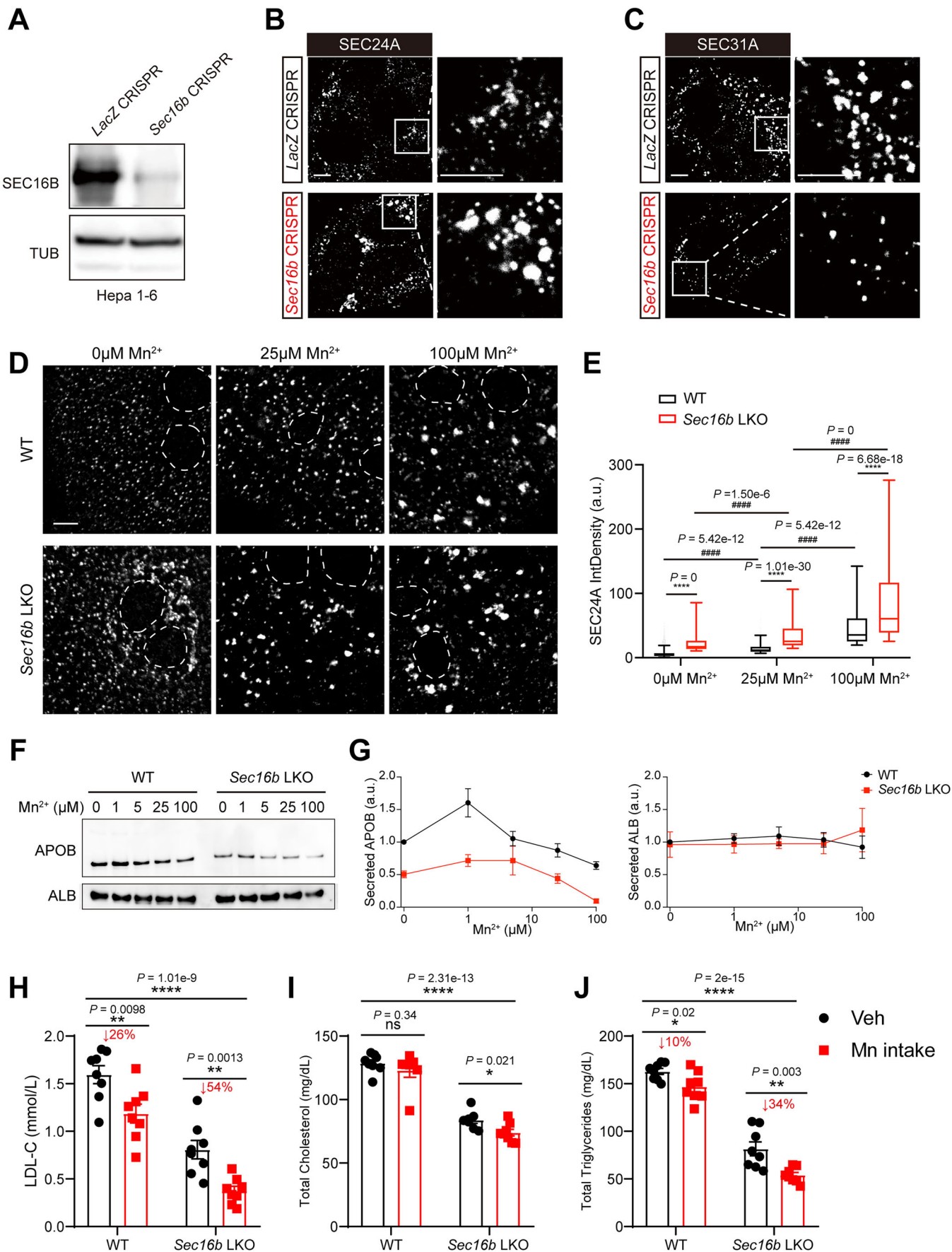

◄

**Figure EV3.  A cooperative effect between SEC16B loss and Mn²⁺ signals on COPII coalescence and lipid secretion.**

(**A**) IB analysis of SEC16B protein depletion in Hepa 1-6 with CRISPR-mediated gene editing. (**B**) Loss of SEC16B enhances SEC24A coalescence in Hepa 1-6. CTL (*LacZ* CRISPR) or *Sec16b* KO (*Sec16b* CRISPR) Hepa 1-6 cells were stained with the anti-SEC24A antibody prior to confocal microscopy. Scale bars = 5 μm. (**C**) Loss of SEC16B decreases SEC31A signals in puncta. The same set of cells as in (**B**) were stained with the anti-SEC31A antibody prior to confocal microscopy. Scale bars = 5 μm. (**D**) SEC16B loss enhances the promoting effect of Mn²⁺ signals on COPII coalescence. Primary hepatocytes isolated from control (WT) or *Sec16b* KO mice were treated with $MnCl_2$, prior to SEC24A staining. Dashed lines: nucleus. Scale bars = 5 μm. (**E**) Quantification of the integrated fluorescence signal density of SEC24A puncta in (**D**). Data are shown as a box plot with 5–95% distribution, with the line inside the box and box edges representing median and the interquartile ranges, respectively. $n$ = 2615, 747, 896, 512, 425, and 271 puncta for WT, *Sec16b* KO, WT with 25 μM $MnCl_2$ treatment, *Sec16b* KO with 25 μM $MnCl_2$ treatment, WT with 100 μM $MnCl_2$ treatment, and *Sec16b KO* with 100 μM $MnCl_2$ treatment, respectively. Data were pooled from 10 cells per condition. Statistical analysis between WT and *Sec16b* KO under the same $MnCl_2$ treatment was determined by two-tailed Student's *t* test. ****$P$ < 0.0001. Statistical analysis among different $MnCl_2$ treatments within the same genotype was determined by ANOVA with the Tukey post hoc test. ####$P$ < 0.0001. a.u. arbitrary units. (**F**) Cooperative regulation of SEC16B loss and Mn²⁺ signals in APOB secretion. Primary hepatocytes isolated from WT or *Sec16b* LKO mice were treated with TPEN for 30 min, followed by $MnCl_2$ treatment for 8 h with indicated doses. APOB and albumin in the medium were analyzed by IB. Representative of three independent experiments is shown. (**G**) Quantification of IB signals in (**F**). The IB signals were normalized to the secretion level of WT at 0 μM $MnCl_2$ treatment. $n$ = 3 biologically independent replicates for each genotype at each treatment. Data are shown as mean ± SEM. a.u. arbitrary units. (**H–J**) Cooperative effects between SEC16B loss and Mn²⁺ signals in lowering plasma lipids. Total LDL-C (**H**), total cholesterol (**I**) and triglyceride (**J**) levels in plasma of WT and *Sec16b* LKO mice with vehicle or $MnCl_2$ intake were measured. $n$ = 8 for each group. Data are shown as mean ± SEM. *$P$ < 0.05; **$P$ < 0.01; ****$P$ < 0.0001. ns no significance (two-tailed Student's *t* test). Source data are available online for this figure.

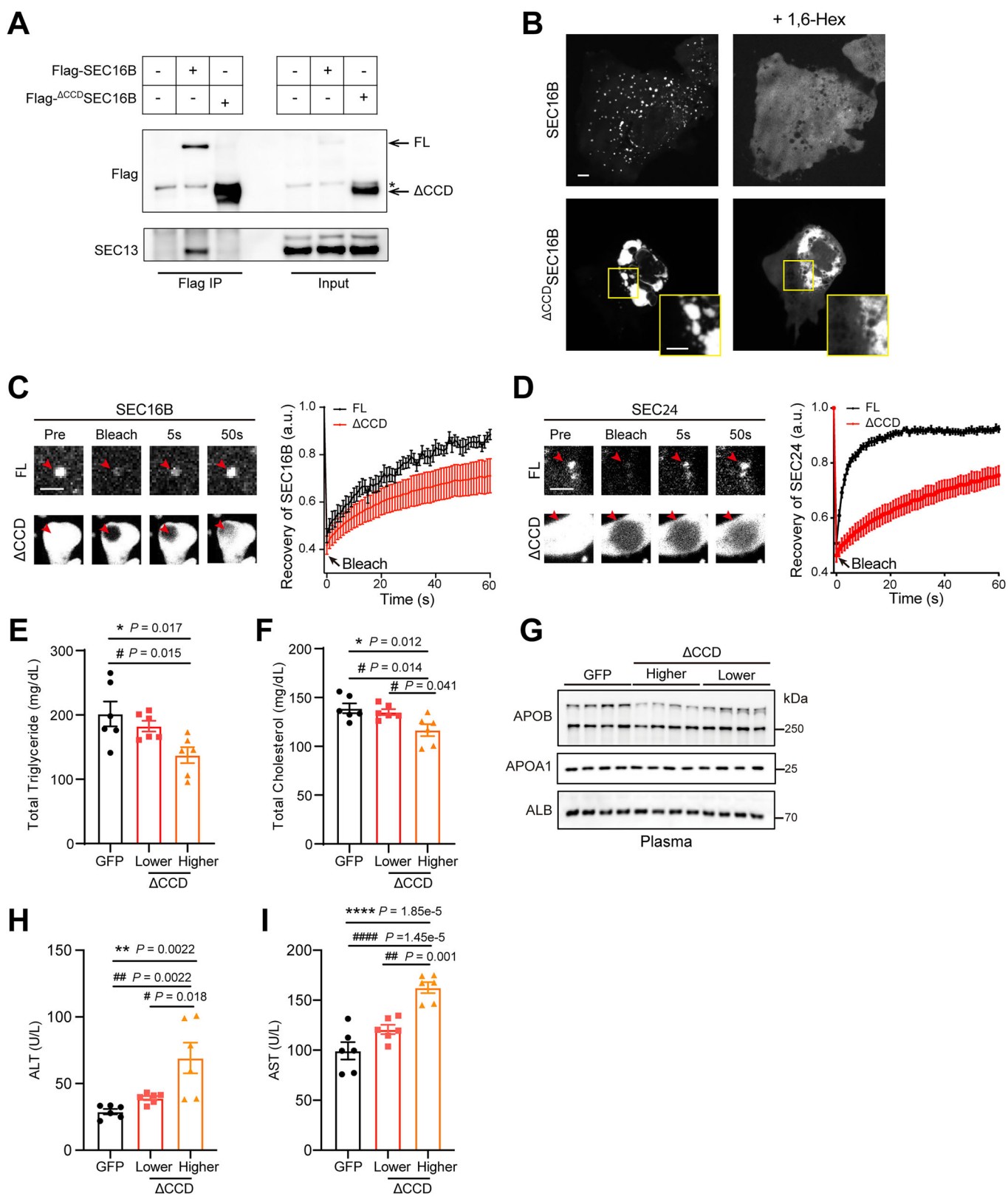

◀

**Figure EV4.   Abnormal COPII assembly impairs secretion.**

(A) SEC16B lacking CCD domain ($^{\Delta CCD}$SEC16B) fails to interact with SEC13. Mouse livers expressing Flag-SEC16B or Flag-$^{\Delta CCD}$SEC16B were lysed and subjected to anti-Flag Co-IP, followed by SDS-PAGE and IB with the indicated antibodies. (B) Confocal image of Huh7 cells transfected with GFP-SEC16B or GFP-$^{\Delta CCD}$SEC16B before or after 1,6-Hex treatments for 30 s. Scale bars $=$ 5 μm. (C) FRAP comparison of GFP-SEC16B coalescence and $^{\Delta CCD}$SEC16B over-aggregates in Huh7 cells. Arrowheads: selected foci for photobleaching in each condition. Scale bars $=$ 2.5 μm. Right, quantification analysis, shown as mean ± SEM. $n=9$ for GFP-SEC16B coalescence and 7 for GFP-$^{\Delta CCD}$SEC16B aggregates, respectively. (D) FRAP comparison of RFP-SEC24D colocalized with GFP-SEC16B coalescence or $^{\Delta CCD}$SEC16B over-aggregates in Huh7 cells. Arrowheads: selected foci for photobleaching in each condition. Scale bars $=$ 2.5 μm. Right, quantification analysis, shown as mean ± SEM. $n=11$ and 9 for RFP-SEC24D colocalized with GFP-SEC16B coalescence or $^{\Delta CCD}$SEC16B aggregates, respectively. (E) Plasma triglyceride levels of mice receiving AAV-TBG-GFP or AAV-TBG-$^{\Delta CCD}$SEC16B. Data are shown as mean ± SEM. $n=6$ mice for each group. $^*P < 0.05$ determined by ANOVA; $^\#P < 0.05$ determined by the Tukey post hoc test. (F) Plasma total cholesterol levels of mice in (E). Data are shown as mean ± SEM. $^*P < 0.05$ determined by ANOVA; $^\#P < 0.05$ determined by the Tukey post hoc test. (G) IB analysis of plasma samples from mice in (E). Representative of three independent experiments is shown. (H) Plasma ALT levels of mice in (E). Data are shown as mean ± SEM. $^{**}P < 0.01$ determined by ANOVA; #, $P < 0.05$; $^{\#\#}P < 0.01$ determined by the Tukey post hoc test. (I) Plasma AST levels of mice in (E). Data are shown as mean ± SEM. $^{****}P < 0.0001$ determined by ANOVA; $^{\#\#}P < 0.01$; $^{\#\#\#\#}P < 0.0001$ determined by the Tukey post hoc test. Source data are available online for this figure.

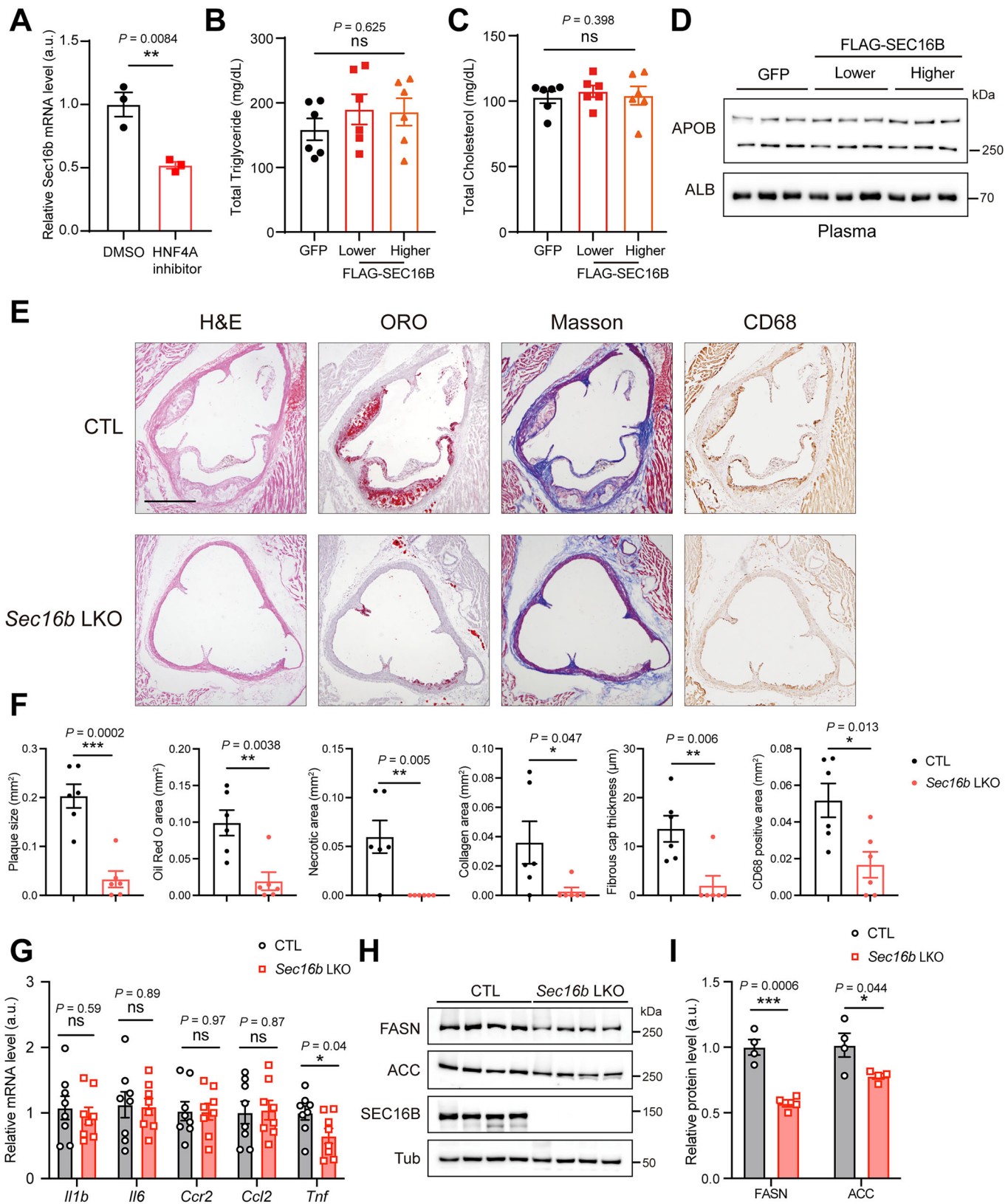

**Figure EV5. Therapeutic potentials of SEC16B targeting in dyslipidemia and ASCVD.**

(A) Relative *Sec16b* mRNA levels in murine primary hepatocytes treated with DMSO control, or BI6015, a HNF4A inhibitor, as measured by reverse transcription-quantitative PCR (RT-qPCR). Data are shown as mean ± SEM. $n = 3$ biological replicates. *P* values were determined by two-tailed Student's *t* test. (B) Plasma triglyceride levels of mice receiving AAV-TBG-GFP or AAV-TBG-SEC16B. $n = 6$ mice for each group. Data are shown as mean ± SEM. ns, no significance determined by ANOVA test. (C) Plasma total cholesterol levels of mice in (B). Data are shown as mean ± SEM. $n = 6$ mice for each group. ns, no significance determined by the ANOVA test. (D) IB analysis of plasma samples from mice in (B). Representative of three independent experiments is shown. (E) Representative of H&E, ORO, Masson and CD68 IHC at aortic root cross-sections from mice in Fig. 7A. Scale bars = 100 μm. (F) Quantification of plaque size, ORO positive area, necrotic area, collagen area, fibrous cap and CD68 positive are using aortic root sections in (E). Data are shown as mean ± SEM. $n = 6$ mice for each group. *$P < 0.05$; ****$P < 0.0001$ (two-tailed Student's *t* test). (G) Relative expression levels of hepatic inflammatory genes in mice from Fig. 7A. Hepatic mRNA from the indicated mouse liver samples were subjected to RT-qPCR using primers as listed in Dataset EV1. $n = 8$ mice for each group. *P* values were determined by two-tailed Student's *t* test. (H) IB analysis of liver samples from mice in Fig. 7A. (I) Quantification of IB signals in (H). Data are shown as mean ± SEM. $n = 4$ mice for each group. *$P < 0.05$; **$P < 0.01$ (two-tailed Student's *t* test). Source data are available online for this figure.

