## [Peer Review File · The EMBO Journal]

Tissue-selective COPII modulator SEC16B aggravates cardiovascular disease by promoting lipid export

Xiao Wang, Yating Hu, Lu Liu, Runze Huang, Yawei Wang, Bing Liu, Yifei Zhao, Yuangang Zhu, Jia Lv, Liang Liu, Huimin Wang, Lingzhi Wu, Xinxuan Xu, Yaxin Li, Guanlin Wang, and Xiao-Wei Chen

Corresponding authors: Xiao-Wei Chen (xiaowei_chen@pku.edu.cn) , Xiao Wang (Xiao_WANG@pku.edu.cn)

Review Timeline:

Submission Date:	19th Jun 25
Editorial Decision:	24th Sep 25
Revision Received:	1st Dec 25
Editorial Decision:	16th Jan 26
Revision Received:	3rd Feb 26
Accepted:	26th Feb 26

Editor: Ieva Gailite

Transaction Report:

Dear Xiao-Wei,

Thank you for submitting your manuscript for consideration by the EMBO Journal. We have now received comments from four reviewers, which are included below for your information.

As reflected in their reports, the reviewers appreciate the findings, but also find that substantial further experimental analysis will be needed to convincingly support the findings. Based on the overall interest expressed in the reports of all reviewers and your willingness to engage in a major revision as expressed in the preliminary revision plan provided during the pre-decision consultation, I invite you to revise the manuscript in response to the comments by all reviewers. As discussed, from the editorial side addressing point 4 from reviewer #2 will not be needed, similarly to a full reconciliation of the impacts of manganese signalling vs SEC16B on COPII condensation as requested by reviewer #3 in their paragraph 4.

We generally allow three to six months as standard revision time. Due to the extent of the required revisions, I have currently set the deadline to six months. Should you foresee a problem in meeting this deadline, please let me know in advance to discuss an extension. I should add that it is The EMBO Journal policy to allow only a single major round of revision and that it is therefore important to resolve the main concerns at this stage.

As a matter of policy, competing manuscripts published during this period will not negatively impact on our assessment of the conceptual advance presented by your study. However, please contact me as soon as possible upon publication of any related work to discuss the appropriate course of action.

When preparing your letter of response to the referees' comments, please bear in mind that this will form part of the Review Process File and will therefore be available online to the community. For more details on our Transparent Editorial Process, please visit our website: <https://www.embopress.org/page/journal/14602075/authorguide#transparentprocess>. Please also see the attached instructions for further guidelines on preparation of the revised manuscript.

Please feel free to contact me if you have any further questions regarding the revision. Thank you for the opportunity to consider your work for publication. I look forward to receiving your revised manuscript.

With best regards,

leva

leva Gailite, PhD
Senior Scientific Editor
The EMBO Journal
Meyerhofstrasse 1
D-69117 Heidelberg
Tel: +4962218891309
i.gailite@embojournal.org

- a point-by-point response to the referees' comments, with a detailed description of the changes made (as a word file).
- a word file of the manuscript text.
- individual production quality figure files (one file per figure)

- a complete author checklist, which you can download from our author guidelines (<https://www.embopress.org/page/journal/14602075/authorguide>).

- Expanded View files (replacing Supplementary Information)

We realize that it is difficult to revise to a specific deadline. In the interest of protecting the conceptual advance provided by the work, we recommend a revision within 3 months (23rd Dec 2025). Please discuss the revision progress ahead of this time with the editor if you require more time to complete the revisions.

Referee #1:

In this manuscript, Wang et al. integrated comprehensive bioinformatics analysis with experiments on both molecular cell and tissue metabolism scale to uncover Sec16B as a modulator for lipoprotein export in hepatocytes. The therapeutic potential of Sec16B was further validated in mice models. The computational analysis is solid and persuasive, the experiments are carefully designed and well explained, and the data is presented in a very clear way. The manuscript is well-written. There are just a few points that may need further clarification to improve the clarity of the message the manuscript would like to deliver to the general audience.

Major points:

1. In Fig. 4c, the difference of Sec13 localization between vector and Sec16B expression is very striking. Which promoter is Sec16B under in this case and could it be an issue with overexpression? Similar question may be asked for Fig. 4d.
2. For Fig.4d, GFP alone may not be the best control, especially if the authors conclusion is focused on "Sec24A on ER surface". Potentially, the ER can be stained with a different dye, or a different ER marker might be used.
3. The evidence provided in Fig. 4 is convincing in that Sec16B modulates COPII assembly, and that Sec16B interacts exclusively with Sec13. The decrease in Sec31A assembly and increase in Sec24A coalescence is consistent with what is already known mechanistically in other systems about how Sec16 might "regulate" assembly of the inner and outer coats: yeast Sec16 competes for binding of Sec31 to the inner coat (Kung et al, 2012), likely via PPP-motifs in Sec16 that compete for binding to Sec23 (Ma and Goldberg 2016). This should probably be incorporated into the authors "condensate" model. Here, knowing whether these features are shared between Sec16A and Sec16B might be helpful, relating to the question of whether there something special about Sec16B or is it only gene expression differences that account for the functional specificity.
4. In the section "COPII over-aggregation disrupts ER dynamics", the formation of large, irregular-shaped COPII assembly in cells were described as "over-aggregation", "over-condensation"; the assembly themselves as "mega-condensation", "mega-aggregates", interchangeably. A uniform wording here may help reduce confusion of the nature of the observation. As the authors provided beautiful EM images showing the assembly is actually a mixture of protein assembly and abnormally-shaped ER membranes, they might consider calling the observations as "abnormal COPII assembly". The term "ER dynamics" also has its specific meaning, and the data showed here may not be the best to support the claim. The authors might consider changing the section title to "Abnormal COPII assembly affects ER morphology and secretion"

Minor:

1. It would be helpful to have a diagram of the protein domains of Sec16A and Sec16B so that the reader can appreciate how these proteins differ - or is the gene expression pattern the sole specific difference between these proteins?
2. For Fig. 1m, the authors may consider leveraging the "gene gain/loss tree" in ensembl gene browser to further support their claim on the evolutionary history of Sec16B and ApoB, as it is believed that two major whole genome duplication has happened in vertebrates (<https://pubmed.ncbi.nlm.nih.gov/16128622/>)
3. In Fig. 3g, the meaning of the circle in the PCA analysis was not explained in the figure legend.
4. The data in Fig. 3h is beautiful and potentially very useful for the general community. The authors might consider posting a list of significant genes as a useful resource to the public.
5. Regarding the Sec16B-Sec13 interaction (line 261), this feature has been well-established in other systems: first for Sec16ACCD-Sec13 using yeast 2-hybrid from David Stephens lab, then by structural analysis for yeast Sec16-Sec13 by Thomas Schwartz. These studies should be cited. The functional importance of this interaction hasn't been established, and

might well be independent of the Sec13/Sec31 interaction.

Referee #2:

Overall comments:

In this manuscript, Wang, et al investigated the role of SEC16B in lipid export and evaluated the potential of developing SEC16B as a therapeutic target in cardiovascular disease. Their work includes huge amount of work in both cell culture and animal studies. They also provide a unique mechanism by which SEC16B affects COPII condensation through regulating SEC23/24 and SEC13 in hepatocytes. However, despite their admirable efforts, the key mechanisms was not supported well due to the lack of critical in vivo experiments. In addition, the fact that atheroprotective effect of SEC16B may involve different mechanisms caused further confusion. I will discuss some of this below:

Major points:

- 1) Authors have done a great job to dissect the detailed mechanism by which SEC16B regulated ER dynamics. However, beyond the molecular mechanism, there was no further investigation in how SEC16B affects hepatocyte function, especially ApoB secretion. For instance, authors found SEC16B affected COPII condensation through regulating SEC23/24 and SEC13. So whether the Δ CCD299 SEC16B affects hepatocyte phenotype and ApoB secretion should also be studied in the revision. In addition, the utilization of animal models expressing Δ CCD299 SEC16B will help supporting their hypothesis.
- 2) One previous study (PMID: 23580231) indicated SEC24A deficiency lowered plasma cholesterol through reduced PCSK9 secretion. Considering authors found SEC16B deficiency inhibited ER-Golgi transport by increasing coalescence of SEC24A, they also need to determine whether PCSK9 secretion is affected in Sec16b LKO mice.
- 3) In Fig. 3, authors revealed sec16b deficiency led to the down-regulation in fatty acid biosynthesis and cholesterol biosynthesis in mice under refeeding conditions. However, there was no significance of TC and TG level between Sec16b LKO mice and control under refeeding conditions (Fig. 3f). Therefore, it seems that the downregulation of cholesterol biosynthesis is an independent phenotype and not related to COPII machinery. Additional experiments are needed to characterize which mechanism results in the protective effect against ASCVD in Sec16b LKO mice.
- 4) It's interesting that authors found SREBP pathway is blunted in Sec16b LKO mice. However, whether SREBP pathway was involved in the protective effect of Sec16b deficiency on ASCVD is not explored in the current manuscript. To support their hypothesis, a mouse model with inactive SREBP pathway in hepatocytes is need.
- 5) Similarly, authors demonstrated HNF4 α accounted for the regulation of Sec16b expression. However, they did not provide any evidence confirming their finding in vivo. At least the expression and activity of HNF4 α in hepatocytes during HFD-induced obese are needed in the revision.

6) For the measurement of atherosclerosis, authors only provided the quantification of atherosclerotic plaques on the thoracic aorta. To do a better job, it is recommended to provide the quantification of burden size, lipid content, and plaque complexity (macrophage amount, cell death, and fibrous cap, etc.) of atherosclerotic plaques on the aortic roots or brachiocephalic artery.

Minor points:

- 1) Please provide the experimental information of data in Fig. 3a. Are these data from mice under fasting or refeeding conditions?
- 2) It is an interesting finding that no difference in hepatic inflammation was observed between SEC16B deficient or control mice under control and refeeding conditions, while reduced macrophage infiltration was found in SEC16B-deficient livers during plaque progression (Fig. 7n and 7p). Authors may need a few sentences to explain this phenotype in the revision.

Referee #3:

This manuscript by Wang et al investigates SEC16B as a regulator of lipoprotein secretion, proposing it acts as a "molecular brake" on COPII assembly. While the findings present interesting observations, the work contains methodological limitations and interpretive issues that weaken its conclusions. A particular concern is that the same laboratory recently published a comprehensive characterization of COPII condensation regulated by manganese (Wang et al., Nature Cell Biology 2023, PMID: 37884645), yet this manuscript does not integrate or address this highly relevant mechanism.

The laboratory's prior work established that COPII undergoes phase separation driven by SEC24 IDRs, with manganese as a physiological regulator, supported by FRAP dynamics, phase diagrams, and 1,6-hexanediol sensitivity tests. The current manuscript proposes that SEC16B also regulates COPII condensation but does not examine potential interactions between these mechanisms or apply comparable experimental standards. The evidence for condensation regulation relies primarily on changes in puncta intensity without the biophysical characterization employed in their previous publication.

A critical concern involves the Δ CCD mutant experiments. The electron microscopy data in Figure 5h reveal structures more consistent with protein aggregation than physiological condensation - specifically, the ~20nm membrane spacing and electron-dense material between ER membranes suggest crystalloid organization rather than liquid-like condensates. This distinction is important for interpreting the proposed mechanism.

Given the laboratory's established expertise in COPII condensation, several key experiments would strengthen this work: (1) testing whether SEC16B modulates manganese-induced condensation, (2) examining if manganese affects SEC16B function,

(3) determining the relative contributions of each mechanism under physiological conditions, and (4) explaining the biological rationale for multiple condensation regulators of the same process.

Additional experimental gaps include the absence of dose-response relationships, kinetic analyses of secretion, and validation of the AlphaFold3-predicted SEC16B-SEC13 interaction through mutagenesis or biochemical approaches. The in vitro budding assays show modest effects (5-20% differences) that require additional validation.

The human genetic evidence, while suggestive, shows small effect sizes ($\beta = -0.004$ to -0.012 for lipid traits, $\beta = 0.0205$ for ALP) that warrant cautious interpretation. Direct demonstration that the identified SNP affects SEC16B expression in hepatocytes would strengthen these associations.

To advance this work, I recommend either: (1) comprehensively examining how SEC16B relates to the established manganese-regulated condensation mechanism with comparable experimental rigor, or (2) focusing on SEC16B's regulatory functions independent of condensation claims. This would provide clearer mechanistic insight and better integrate with the laboratory's existing body of work on COPII regulation.

Referee #4:

The authors identified SEC16B as a novel protein involved in VLDL secretion, specifically required for the ER-Golgi transport of lipidated APOB particles. Its hepatic expression appears to be regulated by HNF4A, and a genetic variant in the promoter region of SEC16B is associated with plasma cholesterol levels in humans. The manuscript is well written, and the data are clearly presented.

While the results support the main conclusions, several points should be addressed:

Figure 1: The authors show an association between rs6682862 and plasma cholesterol levels. Since VLDL is rich in triglycerides, did the authors also examine the association between this variant and plasma triglyceride levels? If no association was found, how do they explain this discrepancy?

Figure 1. The authors performed a co-expression analysis and identified several genes. This approach has also been used recently by others (Zwol et al., Visser et al.). Did the authors also detect SMLR1 and ERICH4 in their datasets?

Figure 3: The authors suggest that EM analysis visualizes lipoprotein particles. How do they confirm that these structures are indeed lipoprotein particles? A common limitation of EM data is that typically only a single representative image is shown without quantification.

Figure 3: The authors claim that SEC16B specifically regulates the ER-Golgi transport of lipidated APOB particles, but this conclusion is based solely on albumin levels. Did they assess other secretory proteins, such as PCSK9? Alternatively, was a proteomics analysis of plasma proteins performed to support this specificity?

If SEC16B is so specific for APOB, does SEC16B bind directly to APOB?

Figure 3: The authors show that loss of hepatic SEC16B does not increase hepatic inflammation. Is this conclusion based solely on the marker F4/80? Did the authors also examine other inflammatory markers and cytokines to support this finding? (see also Figure 7, only F4/80 was used as an inflammation marker).

Figure 6: The authors suggest that HNF4A regulates lipid metabolism, potentially via SEC16B. Does HNF4A also regulate APOB secretion? Was this examined in the HNF4A knockout or overexpression models? Does overexpression of SEC16B increase VLDL production? This is relevant information, as SEC16B expression is increased in hyperlipidemia and under pathological lipid-loading conditions.

Impaired VLDL secretion is often associated with hepatic steatosis and can exacerbate diet-induced steatosis; however, this does not appear to occur with the loss of SEC16B. Do the authors have an explanation for this observation?

Since SEC16B is also expressed in the gut, can the authors speculate on whether SEC16B might be involved in chylomicron secretion, and how this process might be regulated? Additionally, does the studied variant have any effect in the gut?

Point-to-point response

We sincerely thank the reviewers for their time and insightful comments. As detailed in the following point-to-point response, we have followed these constructive comments to revise our manuscript, with additional experimental data and textual revisions. Moreover, some of the comments have inspired experiments that have greatly helped our ongoing investigations, which are also provided in the response. Herein, the reviewers' comments are quoted in blue and italicized. Changes to the manuscript are highlighted with yellow background in the revised manuscript file.

Referee #1 (Report for Author)

“In this manuscript, Wang et al. Integrated comprehensive bioinformatics analysis with experiments on both molecular cell and tissue metabolism scale to uncover Sec16B as a modulator for lipoprotein export in hepatocytes. The therapeutic potential of Sec16B was further validated in mice models. The computational analysis is solid and persuasive, the experiments are carefully designed and well explained, and the data is presented in a very clear way. The manuscript is well-written. There are just a few points that may need further clarification to improve the clarity of the message the manuscript would like to deliver to the general audience.”

Authors' response: We thank the reviewer for these encouragements and insightful suggestions, which have guided us to carry out additional experiments and textual revisions to strengthen the manuscript.

“Major points:

- 1. In Fig. 4c, the difference of Sec13 localization between vector and Sec16B expression is very striking. Which promoter is Sec16B under in this case and could it be an issue with overexpression? Similar question may be asked for Fig. 4d.*
- 2. For Fig.4d, GFP alone may not be the best control, especially if the authors conclusion is focused on "Sec24A on ER surface". Potentially, the ER can be stained with a different dye, or a different ER marker might be used.”*

Authors' response: We thank the reviewer for these critical questions. As guided by the reviewer, we introduced an additional control by expressing the ER transmembrane protein GFP-SEC61 β under the same CMV promoter used for SEC16B. In cells expressing GFP-SEC61 β , SEC13 remains diffusive in the cytosol, in stark contrast to its re-localization into coalesced structures on the ER surface in GFP-SEC16B expressing cells (please see Fig. 3C in the revised manuscript). Consistently, the inner coat condensates marked by SEC24A were unaffected by GFP-SEC61 β overexpression (Fig. EV2I). Furthermore, as the reviewer suggested, we performed co-staining of SEC24A with an endogenous ER marker PDI. While the coalescence of SEC24 were significantly reduced in SEC16B-expressing cells compared to their neighboring cells,

the remaining puncta maintained their fundamental association with the ER membrane (Fig. EV2J). Collectively, the new control experiments helped strengthen our original interpretation that the effects observed are specific to SEC16B.

“3. The evidence provided in Fig. 4 is convincing in that Sec16B modulates COPII assembly, and that Sec16B interacts exclusively with Sec13. The decrease in Sec31A assembly and increase in Sec24A coalescence is consistent with what is already known mechanistically in other systems about how Sec16 might "regulate" assembly of the inner and outer coats: yeast Sec16 competes for binding of Sec31 to the inner coat (Kung et al, 2012), likely via PPP-motifs in Sec16 that compete for binding to Sec23 (Ma and Goldberg 2016). This should probably be incorporated into the authors "condensate" model. Here, knowing whether these features are shared between Sec16A and Sec16B might be helpful, relating to the question of whether there something special about Sec16B or is it only gene expression differences that account for the functional specificity.”

Authors' response: We thank the reviewer for this insightful question regarding the molecular mechanism by which SEC16B regulates COPII assembly. The reviewer rightly points out that Sec16/SEC16A interacts with multiple COPII components via distinct domains to coordinate assembly and dynamics. In contrast, SEC16B lacks inner COPII-binding domains for SEC23 and SEC24, but only retains the conserved CCD domain for SEC13 interaction (as shown in Fig. 4B). On the hand, we have also generated *Sec16a* liver specific knock mice in the lab. Notably, SEC16A deficiency largely abolished COPII condensation and consequently disrupted Golgi organization (right figure). This is in sharp contrast to the increased COPII condensation and overall intact secretory pathway observed upon loss of hepatic SEC16B. Hence, we propose that ubiquitous SEC16A and tissue-restrictive

Figure for reviewers removed

SEC16B play apposing roles in balancing COPII condensation. we fully agree with the reviewer that the precise relationship between these paralogs warrants further investigation and we are working to unravel these molecular mechanisms.

“4. In the section "COPII over-aggregation disrupts ER dynamics", the formation of large, irregular-shaped COPII assembly in cells were described as "over-aggregation", "over-condensation"; the assembly themselves as "mega-condensation", "mega-aggregates", interchangeably. A uniform wording here may help reduce confusion of the nature of the observation. As the authors provided beautiful EM images showing the assembly is actually a mixture of protein assembly and abnormally-shaped ER

membranes, they might consider calling the observations as "abnormal COPII assembly". The term "ER dynamics" also has its specific meaning, and the data showed here may not be the best to support the claim. The authors might consider changing the section title to "Abnormal COPII assembly affects ER morphology and secretion"'''

Authors' response: We thank the reviewer for the helpful suggestions regarding terminology and section title. We agree that "abnormal COPII assembly" more accurately describes the SEC16B mutant-induced structures and have revised the text accordingly to ensure the consistency. We have also adopted the reviewer's suggested section title, as it precisely reflects our findings. These changes have significantly improved the clarity of the manuscript.

"Minor:

1. It would be helpful to have a diagram of the protein domains of Sec16A and Sec16B so that the reader can appreciate how these proteins differ - or is the gene expression pattern the sole specific difference between these proteins?"

Authors' response: We thank the reviewer for this helpful suggestion. The domain diagrams of SEC16A and SEC16B have been included in the revised manuscript (Fig. EV1H, also see figure below). These diagrams highlight a critical structural distinction: while SEC16A possesses multiple COPII interacting domains (PMID: 22675024), SEC16B has a more limited architecture, only containing the conserved CCD domain responsible for binding SEC13 (confirmed in Fig. 4B). Combined with the opposing effects on COPII coalescence observed in SEC16A KO and SEC16B KO hepatocytes, we propose that the specific function of SEC16B stems not only from its expression pattern, but also from its intrinsic capacity to balance COPII condensation. The reviewer's thoughtful suggestions here and in point #3 above are very helpful and directly guiding our ongoing investigation into molecular foundations underlying the distinct functions of SEC16A and SEC16B.

Domain diagrams of SEC16A and SEC16B

"2. For Fig. 1m, the authors may consider leveraging the "gene gain/loss tree" in ensembl gene browser to further support their claim on the evolutionary history of Sec16B and apob, as it is believed that two major whole genome duplication has happened in vertebrates (<https://pubmed.ncbi.nlm.nih.gov/16128622/>)"

Authors' response: We thank the reviewer for the insightful suggestion. Following the reviewer's guidance, we have conducted gene gain/loss tree browsing for APOB and SEC16B (Fig. EV1G), further supporting functional linkage of these two genes.

“3. In Fig. 3g, the meaning of the circle in the PCA analysis was not explained in the figure legend.”

Authors’ response: We thank the reviewer for the careful examination. The requested information has now been specified in the relevant figure legend.

“4. The data in Fig. 3h is beautiful and potentially very useful for the general community. The authors might consider posting a list of significant genes as a useful resource to the public.”

Authors’ response: We thank the reviewer for highlighting this analysis. we have provided the list of genes significantly changed in Fig. 3H in the file of Dataset EV2 with the revised manuscript.

“5. Regarding the Sec16B-Sec13 interaction (line 261), this feature has been well-established in other systems: first for Sec16ACCD-Sec13 using yeast 2-hybrid from David Stephens lab, then by structural analysis for yeast Sec16-Sec13 by Thomas Schwartz. These studies should be cited. The functional importance of this interaction hasn't been established, and might well be independent of the Sec13/Sec31 interaction.”

Authors’ response: We thank the reviewer for highlighting the references and these references have now been cited in the revised manuscript. The reviewer is right about the well-established interaction between SEC13 and the conserved domain of Sec16/SEC16A. These foundational studies stimulate our work on the distinct role of SEC16B in balancing COPII condensation and thereby lipid secretion, in a manner dependent on its CCD-SEC13 interaction. We are grateful for these constructive comments, which have significantly strengthened our manuscript.

Referee #2 (Report for Author)

“Overall comments:

In this manuscript, Wang, et al investigated the role of SEC16B in lipid export and evaluated the potential of developing SEC16B as a therapeutic target in cardiovascular disease. Their work includes huge amount of work in both cell culture and animal studies. They also provide a unique mechanism by which SEC16B affects COPII condensation through regulating SEC23/24 and SEC13 in hepatocytes. However, despite their admirable efforts, the key mechanisms was not supported well due to the lack of critical in vivo experiments. In addition, the fact that atheroprotective effect of SEC16B may involve different mechanisms caused further confusion. I will discuss some of this below:”

Authors’ response: We thank the reviewer for the constructive comments, which help us strengthen our manuscript, as detailed below.

“Major points:

1) Authors have done a great job to dissect the detailed mechanism by which SEC16B regulated ER dynamics. However, beyond the molecular mechanism, there was no further investigation in how SEC16B affects hepatocyte function, especially apob secretion. For instance, authors found SEC16B affected COPII condensation through regulating SEC23/24 and SEC13. So whether the Δ CCD299 SEC16B affects hepatocyte phenotype and apob secretion should also be studied in the revision. In addition, the utilization of animal models expressing Δ CCD299 SEC16B will help supporting their hypothesis.”

Authors’ response: We thank the reviewer for this valuable suggestion. Following the reviewer’s advice, we introduced the Δ CCDSEC16B mutant into murine liver to further elucidate physiological function of SEC16B-regulated COPII dynamics. As shown in Fig. EV4E-G in the revised manuscript, hepatic introduction of Δ CCDSEC16B led to a dose-dependent reduction in plasma lipids and APOB levels. Additionally, plasma ALT and AST levels were slightly elevated in these mice, indicative of liver damage, which aligns well with our prior observation that the Δ CCDSEC16B mutant impairs ER dynamics by inducing excessive COPII aggregation. We are grateful to the reviewer for the constructive comment, which has greatly improved our manuscript.

“2) One previous study (PMID: 23580231) indicated SEC24A deficiency lowered plasma cholesterol through reduced PCSK9 secretion. Considering authors found SEC16B deficiency inhibited ER-Golgi transport by increasing coalescence of SEC24A, they also need to determine whether PCSK9 secretion is affected in Sec16b LKO mice.”

Authors’ response: We thank the reviewer for this important point regarding PCSK9. In response, we have now performed IB analysis of PCSK9 in plasma samples from fasted CTL and *Sec16b* LKO mice. The results revealed no detectable difference in

plasma PCSK9 between these two genotypes (please see Fig. EV2A in the revised manuscript). We would like to draw attention to the genetic evidence presented in Fig. 7A-G, which demonstrated hepatic *Sec16b* inactivation still exerted a profound lipid-lowering effect in mice expressing a gain-of-function PCSK9 mutant. These data collectively support that SEC16B regulates lipid homeostasis independent of PCSK9.

As the reviewer noted, SEC24A knockout in the liver caused a blockade PCSK9 secretion, which was exaggerated by additional loss of SEC24B but not SEC24C/D (PMID: 23580231). The above studies unveiled paralog-specific functions among the SEC24s. Distinct from the direct inactivation of SEC24A, SEC16B deficiency increased COPII coalescence as an alternative assembly mechanism for lipoproteins as a class of bulky and abundant cargos (PMID: 37884645). Though we monitor COPII coalescence using SEC24A puncta as an indicator, the altered COPII coalescence in SEC16B inactive-hepatocytes selectively hinders the efficient transport of bulky yet abundant APOB-containing lipoproteins, while the transport of general secretory proteins including PCSK9 (and the abundant Albumin) remains largely unaffected.

We thank the reviewer for raising this critical question, which helped us clarify potential mis-understanding of our study.

*“3) In Fig. 3, authors revealed *sec16b* deficiency led to the down-regulation in fatty acid biosynthesis and cholesterol biosynthesis in mice under refeeding conditions. However, there was no significance of TC and TG level between *Sec16b* LKO mice and control under refeeding conditions (Fig. 3f). Therefore, it seems that the downregulation of cholesterol biosynthesis is an independent phenotype and not related to COPII machinery. Additional experiments are needed to characterize which mechanism results in the protective effect against ASCVD in *Sec16b* LKO mice.*

*4) It's interesting that authors found SREBP pathway is blunted in *Sec16b* LKO mice. However, whether SREBP pathway was involved in the protective effect of *Sec16b* deficiency on ASCVD is not explored in the current manuscript. To support their hypothesis, a mouse model with inactive SREBP pathway in hepatocytes is need.”*

Authors' response: We thank the reviewer for the insightful comments. We propose a unified model that the downregulation of lipid biosynthetic genes stems from stalled lipoprotein ER export, constituting an important negative feedback mechanism that is orchestrated by the suppression of SREBP activation.

Specifically, our data showed that hepatic *Sec16b* inactivation resulted in ER retention of lipid-laden lipoproteins (Fig. 3A, and Fig. EV2C-D). This export blockade reduced the plasma levels of atherogenic lipoproteins and thereby conferred the primary protective effect against ASCVD. Furthermore, our data indicated that this lipoprotein export halted at the late-ER stage triggered compensatory negative feedback on the SREBP pathway (Fig. 3I-L), serving as an additional protective layer that prevents hepatic lipid accumulation (Fig. 3E). Therefore, the concerted effect of reduced

lipoprotein secretion and suppressed lipogenesis cooperatively contribute to the anti-ASCVD benefit and the overall liver health.

While we sincerely thank the reviewer for highlighting these interesting observations on SREBP pathway in our study, we respectfully suggest that generating a new SREBP knockout model on top of the SEC16B deficiency may be beyond the scope of the current manuscript. In particular, considering the blunted SREBP activity in SEC16B-deficient hepatocytes and the well-established protective roles of SREBP suppression in metabolically challenged animal models, additional combinatory genetic inactivation in this context would be unlikely to yield further mechanistic insight. During the revision process, we have also consulted the editors, who agreed that such experiments would be excessive for this study. We nevertheless fully agree with the reviewer that dissecting the precise mechanistic link between ER-retained lipoproteins and SREBP activation represents an interesting direction for future research.

“5) Similarly, authors demonstrated HNF4 α accounted for the regulation of Sec16b expression. However, they did not provide any evidence confirming their finding in vivo. At least the expression and activity of HNF4 α in hepatocytes during HFD-induced obese are needed in the revision.”

Authors’ response: We thank the reviewer for the thoughtful comment. We would like to clarify that we have previously included the *in vivo* evidence supporting this regulation: analysis of public RNA-seq data revealed that *Sec16b* expression is reduced by ~70% in the livers of HNF4 α liver-specific KO mice, compared to control mice (Fig. 6H). We have also made textual revision to avoid any potential confusion (please see page 16, 2nd paragraph). This finding was further supported by the data showing that HNF4 α inhibitor treatment significantly reduced SEC16B protein levels in primary hepatocytes (Fig. 6I-J), collectively establishing the key role of HNF4 α in regulating *Sec16b* expression. Following the reviewer’s suggestion, we also examined HNF4 α in HFD challenged livers. The results showed that no detectable change in HNF4 α expression in livers of HFD-fed mice, compared to chow-fed controls (right figure). This indicates that the upregulation of SEC16B in pathological conditions involves a more intricate regulatory layer, beyond the modulation of HNF4 α protein levels. Additionally, SEC16B is also subjected to post-transcriptional or post-translational regulation. We thank the reviewer for this stimulating suggestion, which highlighted an interesting area for future investigations for us and also for the field.

IB analysis of HNF4A in the obese livers

“6) For the measurement of atherosclerosis, authors only provided the quantification

of atherosclerotic plaques on the thoracic aorta. To do a better job, it is recommended to provide the quantification of burden size, lipid content, and plaque complexity (macrophage amount, cell death, and fibrous cap, etc.) Of atherosclerotic plaques on the aortic roots or brachiocephalic artery.”

Authors’ response: We thank the reviewer for the insightful comment. Following the reviewer’s suggestion, we have conducted a comprehensive histopathological evaluation of atherosclerotic plaques at the aortic root, and accordingly carried out quantitative analysis (please see Fig. EV5D-E in the revised manuscript).

“Minor points:

1) Please provide the experimental information of data in Fig. 3a. Are these data from mice under fasting or refeeding conditions?”

Authors’ response: The experimental information of Fig. 3a is now provided in the corresponding legend. These data are collected from mice under fasting conditions.

“2) It is an interesting finding that no difference in hepatic inflammation was observed between SEC16B deficient or control mice under control and refeeding conditions, while reduced macrophage infiltration was found in SEC16B-deficient livers during plaque progression (Fig. 7n and 7p). Authors may need a few sentences to explain this phenotype in the revision.”

Authors’ response: We thank the reviewer for highlighting this data and for providing the helpful suggestion. We propose that the reduced macrophage infiltration in SEC16B-deficient livers under an atherogenic diet in Figure 7, is a consequence of the improved systemic lipid homeostasis. Specifically, the reduced hepatic SREBP activation and lowered circulating LDL levels collectively foster a less pro-inflammatory hepatic microenvironment. This would, in turn, attenuate monocyte recruitment and macrophage activation. The precise molecular mechanism underlying this observation is interesting for future investigation. We thank the reviewer for highlighting these observations and have added this discussion in the revised text (please see page 20, 2nd paragraph and page 21, 1st paragraph).

Referee #3 (Report for Author)

“This manuscript by Wang et al investigates SEC16B as a regulator of lipoprotein secretion, proposing it acts as a “molecular brake” on COPII assembly. While the findings present interesting observations, the work contains methodological limitations and interpretive issues that weaken its conclusions. A particular concern is that the same laboratory recently published a comprehensive characterization of COPII condensation regulated by manganese (Wang et al., Nature Cell Biology 2023, PMID: 37884645), yet this manuscript does not integrate or address this highly relevant mechanism.”

“The laboratory's prior work established that COPII undergoes phase separation driven by SEC24 idrs, with manganese as a physiological regulator, supported by FRAP dynamics, phase diagrams, and 1,6-hexanediol sensitivity tests. The current manuscript proposes that SEC16B also regulates COPII condensation but does not examine potential interactions between these mechanisms or apply comparable experimental standards. The evidence for condensation regulation relies primarily on changes in puncta intensity without the biophysical characterization employed in their previous publication.

Authors’ response: We thank the reviewer for these insightful comments, which have greatly helped us sharpen the conceptual connection between our current study and our previously published work on manganese-regulated COPII condensation (Wang et al., Nature Cell Biology 2023). In that earlier study, we established manganese as a positive regulator that promotes COPII condensation to support lipoprotein export, whereas the current work focuses on identifying SEC16B as a counterbalancing “molecular brake” that prevents over-condensation and thereby enables selective cargo transport. Thus, the two studies address complementary arms of the same regulatory logic—one promoting and one constraining COPII condensation to maintain optimal flux.

Guided by the reviewer’s recommendations, we have now conducted additional experiments to explicitly integrate these mechanisms, including new assays examining how manganese signaling and SEC16B jointly shape COPII condensation behavior. As detailed below, these new data strengthen the unified model that balanced regulation of COPII condensation is essential for physiological lipoprotein secretion. We have incorporated these results and further discussion into the revised manuscript.

A critical concern involves the Δ CCD mutant experiments. The electron microscopy data in Figure 5h reveal structures more consistent with protein aggregation than physiological condensation - specifically, the ~20nm membrane spacing and electron-dense material between ER membranes suggest crystalloid organization rather than liquid-like condensates. This distinction is important for interpreting the proposed mechanism.”

Authors' response: We thank the reviewer for these insightful comments and apologize for any lack of clarity in our previous manuscript. We fully agree with the reviewer that the protein aggregates induced by the Δ^{CCD} SEC16B mutant are very distinct from regular COPII puncta. We have now uniformly referred these aberrant aggregates to as “abnormal COPII assemblies” in the revised manuscript to avoid confusion, as also suggested by reviewer 1. Following this reviewers' suggestion, we have conducted multiple lines of experiments to further characterize this abnormal COPII assembly.

1. **1,6-hexanediol (1,6-Hex) treatment** (Fig. EV4B): Treatment with 1,6-Hex, which completely dissolved wild-type SEC16B puncta, failed to dissolve aggregates formed by the Δ^{CCD} SEC16B mutant. This differential sensitivity indicates that while normal COPII condensates exhibit liquid-like properties, the mutant induces a transition to a gel-like state.
2. **Fluorescence recovery after bleaching (FRAP) assay** (Fig. EV4C-D): To further assess the material properties of these abnormal assemblies, we performed FRAP analysis. In stark contrast to the dynamic exchange seen in normal COPII condensates, both the Δ^{CCD} SEC16B mutant itself and the co-localized SEC24 within the abnormal assemblies showed markedly impaired fluorescence recovery, confirming their greatly reduced molecular mobility.
3. **Physiological outputs of abnormal COPII assembly induced by the mutant** (Fig. EV4E-G): We expressed Δ^{CCD} SEC16B in the liver of adult mice. The results revealed that Δ^{CCD} SEC16B decreased plasma lipid and APOB levels, in a dose-dependent manner. This observation also aligns with our previous *in vitro* data, showing severely impaired ER export efficiency by Δ^{CCD} SEC16B.

In summary, these data collectively strengthen our model in which the COPII condensation needs to be balanced for supporting efficient transport of bulky and abundant lipoproteins.

“Given the laboratory's established expertise in COPII condensation, several key experiments would strengthen this work: (1) testing whether SEC16B modulates manganese-induced condensation, (2) examining if manganese affects SEC16B function, (3) determining the relative contributions of each mechanism under physiological conditions, and (4) explaining the biological rationale for multiple condensation regulators of the same process.”

Authors' response: We thank the reviewer for these constructive suggestions. In response, we have performed new lines of experiments that significantly strengthen our findings and clarify the complementary regulation of COPII condensation: Mn^{2+} acts as a promoting factor (positive regulation), whereas SEC16B serves a molecular brake (negative regulation).

1. **The role of SEC16B in Mn^{2+} regulated COPII condensation:** We treated primary hepatocytes from control and *Sec16* LKO mice with Mn^{2+} . As expected, Mn^{2+} induced a dose-dependent increase in SEC24 condensation and a corresponding

bell-shape regulation on APOB secretion in control hepatocytes. Strikingly, SEC16B deficiency profoundly sensitized the cells to Mn^{2+} , exacerbating its promoting effect on COPII condensation and leading to a more pronounced suppression on APOB secretion (Fig. EV3D-E).

2. **The effect of Mn^{2+} in SEC16B balanced COPII condensation:** To assess the interplay of Mn^{2+} signal and SEC16B in COPII condensation under physiological condition, we administered to wild-type (WT) and *Sec16b* LKO mice with Mn^{2+} via gavage. In WT mice, Mn^{2+} administration reduced plasma LDL-C levels by ~26%. Notably, while *Sec16b* LKO alone resulted in a nearly half reduction in plasma LDL-C, Mn^{2+} administration in these mice caused a further ~54% reduction. Similar cooperative effects were also observed for plasma triglyceride and total cholesterol levels (Fig. EV3H-J), revealing that Mn^{2+} dramatically potentiates the effects of the losing SEC16B-mediated brake on COPII.

In summary, prompted by the reviewer's constructive suggestion, our new data reveal a cooperative relationship of Mn^{2+} and SEC16B, as two arms of regulation of the balanced COPII condensation. While Mn^{2+} promotes COPII condensation by coordinating inner coat proteins, SEC16B act as a crucial brake by modulating outer coat assembly to prevent COPII over-condensation. This multi-layered regulatory system likely allows for the precise control of lipoprotein production and secretion, representing a specialized process intricately coupled to fluctuating nutrient cues. We believe that other factors, particularly the TANGO family proteins, may also contribute to this regulatory mechanism. These may constitute a fruitful avenue for future research that was inspired by the reviewer's insights.

“Additional experimental gaps include the absence of dose-response relationships, kinetic analyses of secretion, and validation of the alphafold3-predicted SEC16B-SEC13 interaction through mutagenesis or biochemical approaches. The in vitro budding assays show modest effects (5-20% differences) that require additional validation.”

Authors' response: We apologize for any lack of clarity in our previous explanation and appreciate this opportunity to provide further clarification in the revised manuscript. We are sincerely grateful to the reviewer for the attention to the dose-response relationships, which underscores the quantitative nature of COPII condensation in lipid control. We have generated *Sec16b* liver specific knockout (*Sec16b* LKO) and haploinsufficient (*Sec16b* LHets) mice. As shown in Fig. 2A-I, the reduction of plasma lipids levels was in dose-dependent manner. These phenotypes were mechanistically linked to a blockade in lipoprotein export, as confirmed by hepatic triglyceride secretion kinetic assay (Fig. 2M), ultrastructure analysis of the early secretory pathway (Fig. 3A-B), and a new biochemical fractionation experiment (Fig. EV2C-D in the revised manuscript). Conversely, hepatic SEC16B overexpression in WT mice induced a slight elevation in plasma lipids (Fig. EV5A-C). collectively, these experiments demonstrate a dose-dependent role for SEC16B in regulating lipid homeostasis.

We also agree with the reviewer that the experimental validation of the AlphaFold-predicted SEC16B-SEC13 is crucial. In our original manuscript, we have performed co-immunoprecipitation to confirm the interaction between SEC16B and SEC13 (Fig. 4B). In this revision, we further solidified this point by demonstrating the endogenous SEC16B-SEC13 interaction in mouse liver (please see Fig. EV2G). Following the reviewer's advice, we further confirmed the requirement of the CCD domain in this interaction, as the Δ^{CCD} SEC16B mutant failed to co-immunoprecipitate with SEC13 (Fig. EV4A in the revised manuscript).

We apologize for the previous figure presentation that might lead to misunderstanding in Fig. 5I. As noted by reviewer, the Δ^{CCD} SEC16B mutant induces a gel-like, abnormal COPII assembly on the ER surface, causing severe ER export impairment. Specifically, our *in vitro* COPII reconstitution assay showed that Δ^{CCD} SEC16B robustly inhibited the packaging efficiency of LMAN1 and SURF4 by ~75% (drop from ~16% to ~4%) and ~65% (drop from ~10% to 3%). Furthermore, hepatic expression of Δ^{CCD} SEC16B in adult mice significantly lowers plasma lipid and APOB levels. We have revised the data presentation of Fig. 5I for clarity, and incorporated these new *in vivo* data in Fig. EV4E-G.

“The human genetic evidence, while suggestive, shows small effect sizes ($\beta = -0.004$ to -0.012 for lipid traits, $\beta = 0.0205$ for ALP) that warrant cautious interpretation. Direct demonstration that the identified SNP affects SEC16B expression in hepatocytes would strengthen these associations.”

Authors' response: We thank the reviewer for this important comment regarding the interpretation of the human genetic evidence. We agree that the effect sizes of the SNP are modest, which is typical for regulatory variants influencing complex traits in humans like blood lipids. As the variant is located in a non-coding region with an unknown cell-type-specific regulatory function, a direct demonstration of the SNP's effect on hepatocyte SEC16B expression is technically challenging. Therefore, our study pivots to definitively establish the gene's role in lipid regulation, through direct physiological and mechanistic experiments. We have revised the text (please see page 7, last paragraph, and page 15, last paragraph) to more cautiously interpret the GWAS data, framing it as a hypothesis-generating clue.

“To advance this work, I recommend either: (1) comprehensively examining how SEC16B relates to the established manganese-regulated condensation mechanism with comparable experimental rigor, or (2) focusing on SEC16B's regulatory functions independent of condensation claims. This would provide clearer mechanistic insight and better integrate with the laboratory's existing body of work on COPII regulation.”

Authors' response: We sincerely thank the reviewer for these valuable suggestions, which substantially improves our study. We have followed the reviewer's comments and carried out requested experiments, resulting in much strengthened manuscript.

Referee #4 (Report for Author)

“The authors identified SEC16B as a novel protein involved in VLDL secretion, specifically required for the ER-Golgi transport of lipidated APOB particles. Its hepatic expression appears to be regulated by HNF4A, and a genetic variant in the promoter region of SEC16B is associated with plasma cholesterol levels in humans. The manuscript is well written, and the data are clearly presented.

While the results support the main conclusions, several points should be addressed:”

Authors’ response: We sincerely thank the reviewer for the positive comments and for the construction suggestions, which have guided us to carry out additional experiments to strengthen the manuscript.

“Figure 1: The authors show an association between rs6682862 and plasma cholesterol levels. Since VLDL is rich in triglycerides, did the authors also examine the association between this variant and plasma triglyceride levels? If no association was found, how do they explain this discrepancy?”

Authors’ response: We thank the reviewer for raising this insightful point. We have now included the plasma triglyceride data, which shows no significant association with the variant. We interpret this finding as consistent with the distinct metabolic fates of lipids. While cholesterol is more efficiently retained within circulating APOB-containing lipoproteins, triglycerides are rapidly hydrolyzed by lipase and consumed in periphery tissues after secretion. This complex interplay among multiple processes on plasma triglycerides likely masks a genetic association linked to the secretion pathway, which is the primary function of SEC16B.

“Figure 1. The authors performed a co-expression analysis and identified several genes. This approach has also been used recently by others (Zwol et al., Visser et al.). Did the authors also detect SMLR1 and ERICH4 in their datasets?”

Authors’ response: We thank the reviewer for the perceptive comment and also directing us to the elegant analysis in referred studies. Our analysis indeed detected both SMLR1 and ERICH4, with expression patterns closely mirroring the observations from Zwol et al. and Visser et al. (PMID: 36053190 and 39492093). Specifically, our data analysis confirmed that SMLR1 is significantly co-expressed with MTTP in the liver and gut, and with APOB in the gut, whereas ERICH4 is significantly co-expressed with both MTTP and APOB in the gut. These congruent findings have been incorporated in the revised text (please see page 6, 2nd paragraph).

“Figure 3: The authors suggest that EM analysis visualizes lipoprotein particles. How do they confirm that these structures are indeed lipoprotein particles? A common limitation of EM data is that typically only a single representative image is shown without quantification.”

Authors' response: We thank the reviewer for this critical comment regarding the interpretation of our EM data. We agree that definitive identification and quantification of lipoprotein particles by EM alone is limited. To improve scientific rigor, we have revised the text, now referring to these imidazole-stained particles in the ER lumen as “lipid laden-particles” (please see page 9, last paragraph). We have also performed biochemical fractionation of subcellular compartments, followed by immune-blotting. This new experiment demonstrates a significant accumulation of APOB in the ER fractions and a corresponding reduction in the Golgi fractions in SEC16B KO livers (Fig. EV2C-D). This provides independent, quantitative validation of the ER export blockade initially suggested by the EM micrographs.

“Figure 3: The authors claim that SEC16B specifically regulates the ER-Golgi transport of lipidated APOB particles, but this conclusion is based solely on albumin levels. Did they assess other secretory proteins, such as PCSK9? Alternatively, was a proteomics analysis of plasma proteins performed to support this specificity?”

Authors' response: We thank the reviewer for this important question regarding the specificity of SEC16B's role. Our data from multiple approaches collectively support its selectivity in lipoprotein secretion. First, we confirmed the global secretion profile is largely intact in *Sec16b* LKO mice, as shown by unchanged plasma protein patterns on silver staining (Fig. 2L). Second, as the reviewer suggested, we directly assessed PCSK9 and performed plasma proteomics. These new experiments confirmed that SEC16B deficiency specifically reduces apolipoproteins without broadly affecting general secretory proteins, including PCSK9 (Fig. EV2A-B). Furthermore, our genetic data in Fig. 7A-G demonstrated that hepatic *Sec16b* inactivation maintained its profound lipid-lowering effect even in mice expressing a gain-of-function PCSK9 mutant. This establishes SEC16B regulates lipid homeostasis independent of the PCSK9 pathway.

“If SEC16B is so specific for APOB, does SEC16B bind directly to APOB?”

Authors' response: We thank the reviewer for the insightful question. To address this question, we carried out co-immunoprecipitation experiments using liver lysates, which showed no detectable interaction between the SEC16B and APOB (Fig. EV2G in the revised manuscript). This observation is consistent with their distinct subcellular localization: SEC16B localizes on the cytosolic side of the ER, whereas APOB is synthesized and lipidated within the ER lumen. Therefore, a direct physical interaction is topologically improbable. Instead, our study illustrated that SEC16B regulates APOB transport by organizing the COPII machinery. In support of this model, we found that SEC16B co-localizes with the ER transmembrane protein SURF4, a known cargo receptor for lipoproteins (Fig. EV2F in the revised manuscript).

“Figure 3: The authors show that loss of hepatic SEC16B does not increase hepatic

inflammation. Is this conclusion based solely on the marker F4/80? Did the authors also examine other inflammatory markers and cytokines to support this finding? (see also Figure 7, only F4/80 was used as an inflammation marker)."

Authors' response: We thank the reviewer for the critical question. Our conclusion that SEC16B loss does not induce hepatic inflammation is supported by multiple independent lines of evidence beyond F4/80: **1) Functional Markers:** plasma ALT and AST levels, sensitive indicators of liver damage and inflammation, were unchanged between WT and *Sec16b* LKO mice on both normal chow (Fig. 3F) and western diet (Fig. 7M); **2) hepatic transcriptome:** RNA profiling revealed no elevation in the expression of core inflammatory factors in *Sec16b* LKO livers (Fig. EV2E and Fig. EV5F in the revised manuscript); **3) Systemic Proteomics:** the quantitative plasma proteomics as suggested by the reviewer confirmed that key inflammatory proteins, including CRP and complement components were similar between *Sec16b* LKO and control mice (Fig. EV2B). Collectively, these data support that hepatic SEC16B deficiency does not trigger detectable inflammation.

"Figure 6: The authors suggest that HNF4A regulates lipid metabolism, potentially via SEC16B. Does HNF4A also regulate APOB secretion? Was this examined in the HNF4A knockout or overexpression models? Does overexpression of SEC16B increase VLDL production? This is relevant information, as SEC16B expression is increased in hyperlipidemia and under pathological lipid-loading conditions."

Authors' response: We thank the reviewer for insightful questions that help clarify the regulatory network. The reviewer is correct that HNF4A is known as an important regulator in lipoprotein metabolism, primarily through its direct transcriptional regulation of *APOB* itself and other key genes like *MTTP* (PMID: 37600688). Our study adds a new layer to this regulation by identifying *SEC16B* as another HNF4A target gene involved in lipoprotein trafficking, supported by the data showing reduced hepatic *Sec16b* mRNA level in HNF4A liver specific KO mice (Fig. 6H), and decreased SEC16B expression in primary hepatocytes treated with an HNF4A inhibitor (Fig. 6I-J). As advised, we overexpressed SEC16B in mouse liver and observed an upward trend in plasma levels of lipids and APOB, although these changes did not reach statistical significance in this model. We thank the reviewer for the constructive suggestions and have now included these new data as Fig. EV5A-C.

"Impaired VLDL secretion is often associated with hepatic steatosis and can exacerbate diet-induced steatosis; however, this does not appear to occur with the loss of SEC16B. Do the authors have an explanation for this observation?"

Authors' response: We thank the reviewer for highlighting this intriguing observation in our study. The absence of hepatic steatosis in SEC16B-deficient liver is likely attributed to the inhibition of SREBP pathway, which stems from the selective blockade of lipoprotein export at the late, post-ER stage, as supported by the data in Fig. 3J-L. a

similar phenomenon has been also reported in hepatic *Surf4*-KO mice, which exhibited a marked reduction in plasma lipids while developing only mild lipid accumulation and no overt liver injury (PMID: 36193893). Together, these independent findings unmask a pivotal regulatory node where lipid biogenesis and transport are delicately coordinated to maintain systemic lipid homeostasis, opening a promising new avenue for future research.

“Since SEC16B is also expressed in the gut, can the authors speculate on whether SEC16B might be involved in chylomicron secretion, and how this process might be regulated? Additionally, does the studied variant have any effect in the gut?”

Authors’ response: The reviewer is exactly right. Prior phenotypic work from the Wang group (authors of the accompanying submission) has implicated intestinal SEC16B in regulating chylomicron secretion. However, the upstream regulatory cues that modulate SEC16B function in the gut—particularly dietary states, microbial metabolites, or inflammatory signals—remain essentially unexplored. We suspect that the variants reported in our study may also serve as useful tools for future studies in the intestinal contexts and thank the reviewer for the insightful question.

Dear Xiao-Wei,

Thank you for submitting a revised version of your manuscript. We have now received input from all original reviewers. While reviewers #1, 3 and 4 find that their main concerns have been addressed satisfactorily and recommend acceptance of the manuscript, reviewer #2 indicates a number of remaining concerns. As discussed previously, experimental investigation of the further role of SREBP is not required. Please address the points by reviewer #2 textually with appropriate caveating and toning down of the conclusions.

There now remain only a few editorial and formatting points that need to be addressed before I can extend official acceptance of the manuscript:

1. It is The EMBO Journal policy for the transcript of the editorial process (containing referee reports and your response letters) to be published as an online supplement to each paper. If you should prefer removal of any referee-only figures included in the point-by-point response(s), e.g. because they may still be used for future publication or because they have been reproduced from published work by others, please do let us know immediately via response email. More information is available here: https://www.embopress.org/transparent-process#Review_Process.
2. Please submit keywords for your manuscript.
3. Please check that the funding information is correct and identical both in the manuscript and our online system. Currently, grant 32400985 is missing in our online system.
4. For author affiliations, please include full addresses of the corresponding institutions.
5. We are missing the ORCID iD for the co-corresponding author Xiao Wang. In order to link the ORCID iD to the account in our manuscript tracking system, the author in question has to do the following:
 - Click the 'Modify Profile' link at the bottom of your homepage in our system.
 - On the next page you will see a box halfway down the page titled ORCID*. Below this box is red text reading 'To Register/Link to ORCID, click here'. Please follow that link: you will be taken to ORCID where you can log in to your account (or create an account if you don't have one)
 - You will then be asked to authorise Wiley to access your ORCID information. Once you have approved the linking, you will be brought back to our manuscript system.Unfortunately, we cannot do this linking on the author's behalf for security reasons.
6. Please move "Data Availability" section to the end of "Methods".
7. Please remove Dataset legends from the manuscript text file.
8. During our routine source data check, I noted the following issues:
 - The source data for the following Western blot panels appears to differ between the figure and the source data: Fig. 5I (Sec 13 - a different exposure?); Fig. 3L (Tub).
 - There is a numerical repetition in the source data for figure 3E. I have attached the corresponding file with the detected duplications labelled in colour. I appreciate that such a duplication can also occur due to specific measurement or calculation methods used.Please take a look and correct if needed. A brief explanation would be very helpful.
9. During our routine image checks, we noted the following issues:
 - the blot images and their associated source data across the figure set appear pixelated under analysis. This is a common outcome when original 16-bit TIFF files are converted to RGB format for publication. In their current form, the pixelation prevents us from completing our integrity assessment.To resolve this, please update the source data for all blot images to include the original captured 16-bit files, prior to conversion to RGB TIFF at 300 dpi. Providing the unprocessed 16-bit images will allow us to review the blots at full resolution and complete the integrity checks.
 - Please provide all microscopy source data for EV Figures 1-5.
 - Figure EV2H (top row, GFP SEC16B) contains an empty panel. Please provide the corresponding source data for this figure so that we can verify the signal.
10. Our data editors have flagged the following issues in figure legends that need correcting:
 - Please correct the labelling the panels for figure EV2 H, I and J in the figure legend.
 - Please provide the exact p values in the legends of figures EV3 E, H-J; EV4 I.
 - Please indicate the statistical test used for data analysis in the legends of figures 1L, 3I-K; 6A, B, C, H; 7M.
 - Please define the box plots in terms of minima, maxima, centre, bounds of box and whiskers, and percentile in the legend of figure 6C.
 - Please provide information on the number and nature of replicates in the legends of figures 4F, H; 6F, H, EV3 E, G; EV4 C, D; EV5 A, B, F.
 - Please define the asterisk in the legend of figure 5B.
 - Please define the circular dashed borders in the legend of figure EV3 D.
 - Please define the red arrow heads in the legend of figure EV4 C, D.
11. Papers published in The EMBO Journal are accompanied online by a 'Synopsis' to enhance discoverability of the manuscript. It consists of A) a short (1-2 sentences) summary of the findings and their significance, B) 3-4 bullet points highlighting key results and C) a synopsis image that is 550x300-600 pixels large (width x height, jpeg or png format). You can

either show a model or key data in the synopsis image. Please note that the image size is rather small and that text needs to be readable at the final size. Please send us this information together with the revised manuscript.

With best wishes,

Ieva

Please remember: Digital image enhancement is acceptable practice, as long as it accurately represents the original data and conforms to community standards. If a figure has been subjected to significant electronic manipulation, this must be noted in the figure legend or in the 'Methods' section. The editors reserve the right to request original versions of figures and the original images that were used to assemble the figure.

We realize that it is difficult to revise to a specific deadline. In the interest of protecting the conceptual advance provided by the work, we recommend a revision within 3 months (16th Apr 2026). Please discuss the revision progress ahead of this time with the editor if you require more time to complete the revisions.

Referee #1:

The authors have satisfactorily addressed my previous concerns.

Referee #2:

General summary and opinion:

The authors have addressed my comments with new experiments and significantly improved the principle significance of manuscript. Their findings demonstrate the atherogenic effect of hepatic SEC16B and revealed a unique mechanism involving lipoprotein secretion, which leads to general significance and high priority for the wider readership of The EMBO Journal. However, a few issues remain to address in the revision. In particular, the data of SEC16B-regulated lipogenesis is too weak to support their conclusion.

Specific major concerns:

1) Authors tried to explain that the anti-ASCVD benefit and the overall liver health in SEC16B KO mice was due to the concerted effect of reduced lipoprotein secretion and suppressed lipogenesis. However, the role of SEC16B in regulating lipogenesis has not been fully investigated and highlighted in the revised manuscript. In the current abstract, authors only addressed SEC16B as a specialized modulator of the general secretory (SEC) pathway, while its dual-function of suppressing lipogenesis was not mentioned. Instead, they stated "SEC16B co-emerges with core lipoprotein biosynthetic genes", which was not enough to demonstrate the critical role of SEC16B-regulated lipogenesis in anti-ASCVD benefit and liver health. Authors should edit the abstract or even the title since lipoprotein secretion is not the sole mechanism.

2) If authors are unable to generate the animal model with inactive SREBP pathways, at least some experiments with siRNA or inhibitors could be conducted in hepatocytes. To support their conclusion that SEC16B KO also impaired lipogenesis and

contributes to reduced lipid export and accumulation in hepatocytes, the lipid/lipoprotein secretion as well as lipid content should be evaluated with an inhibition of SREBP in control or SEC16B KO hepatocytes.

3) The new data of HNF4 α indicates that the upregulation of SEC16B in pathological conditions is independent of HNF4 α protein levels. It's an interesting finding but difficult to explain the data that HNF4 α inhibitor downregulated both expression of HNF4 α and SEC16B (Fig. 6I-J). In addition, the transcriptional level of SEC16B in Fig. 6I and 6K should be measured to support authors' explanation.

Referee #3:

I thank the authors for their thorough and thoughtful revisions to the manuscript. The revised version has comprehensively addressed my original concerns, and I am now satisfied that the study meets the standards for publication in The EMBO Journal.

Referee #4:

The authors sufficiently responded to my comments. I have no other questions

Point-to-Point Response

We sincerely thank the reviewer for the time and insightful comments. We have followed these constructive comments with textual clarifications and additional experimental data. Herein, the reviewers' comments are quoted in blue and italicized.

Referee #2:

“General summary and opinion:

The authors have addressed my comments with new experiments and significantly improved the principle significance of manuscript. Their findings demonstrate the atherogenic effect of hepatic SEC16B and revealed a unique mechanism involving lipoprotein secretion, which leads to general significance and high priority for the wider readership of The EMBO Journal. However, a few issues remain to address in the revision. In particular, the data of SEC16B-regulated lipogenesis is too weak to support their conclusion.

Specific major concerns:

1) Authors tried to explain that the anti-ASCVD benefit and the overall liver health in SEC16B KO mice was due to the concerted effect of reduced lipoprotein secretion and suppressed lipogenesis. However, the role of SEC16B in regulating lipogenesis has not been fully investigated and highlighted in the revised manuscript. In the current abstract, authors only addressed SEC16B as a specialized modulator of the general secretory (SEC) pathway, while its dual-function of suppressing lipogenesis was not mentioned. Instead, they stated "SEC16B co-emerges with core lipoprotein biosynthetic genes", which was not enough to demonstrate the critical role of SEC16B-regulated lipogenesis in anti-ASCVD benefit and liver health. Authors should edit the abstract or even the title since lipoprotein secretion is not the sole mechanism.”

Authors' response: We thank the reviewer for highlighting the dual metabolic benefits in *Sec16b*-deficient mice: 1) protection against ASCVD, largely due to a blockade in late-stage lipoprotein secretion from the ER, and 2) preserved liver health, primarily attributed to the consequential suppression of hepatic lipogenesis. This suppression is triggered by initial ER lipid accumulation stemming from the lipoprotein secretion blockade. We fully agree with the reviewer on the significance of this concerted feedback mechanism on lipogenesis, and have now highlighted this point in the 'Synopsis' section. We also appreciate the reviewer for pointing out a potential misunderstanding of the phrase “core lipoprotein biosynthetic genes”. We intended this to refer specifically to genes involved in lipoprotein assembly, such as those encoding apolipoproteins and MTTP, and have edited the abstract to improve clarity as advised.

“2) If authors are unable to generate the animal model with inactive SREBP pathways,

at least some experiments with siRNA or inhibitors could be conducted in hepatocytes. To support their conclusion that SEC16B KO also impaired lipogenesis and contributes to reduced lipid export and accumulation in hepatocytes, the lipid/lipoprotein secretion as well as lipid content should be evaluated with an inhibition of SREBP in control or SEC16B KO hepatocytes.”

Authors’ response: We thank the reviewer for this thoughtful comment. Our data reveal that *Sec16b* deficiency causes profound blockage in ER export of lipoproteins, resulting in reduced lipid secretion. This ER retention coincides with a significant suppression of SREBP activation and its target genes (fig. 3i-l), suggesting the pathway is substantially inactivated. Since the SREBP pathway is already significantly suppressed in *Sec16b*-deficient livers, imposing additional pharmacological or genetic inhibition of SREBP in this context would be unlikely to yield further mechanistic insight, particularly considering that the well-established protective role of such inhibition in animal models. Editorial consultation has also suggested such experiments may not be necessary for the current study. We fully agree that elucidating the precise mechanistic link connecting ER-retained lipoproteins to SREBP inactivation is a compelling question, opening an important avenue for future investigation.

“3) The new data of HNF4 α indicates that the upregulation of SEC16B in pathological conditions is independent of HNF4 α protein levels. It's an interesting finding but difficult to explain the data that HNF4 α inhibitor downregulated both expression of HNF4 α and SEC16B (Fig. 6I-J). In addition, the transcriptional level of SEC16B in Fig. 6I and 6K should be measured to support authors' explanation.”

Authors’ response: We thank the reviewer for this insightful comment. The reviewer rightly points out that HNF4 α inhibitor treatment reduces both HNF4 α and SEC16B expression levels. As HNF4 α activates its own gene promoter, inhibiting its transcriptional activity disrupts this self-sustaining feedback, thereby downregulating its expression (PMID: 30191603). The reduction of SEC16B protein by the HNF4 α inhibitor establishes that HNF4 α is a key physiological regulator of SEC16B. This is further strengthened by the significant downregulation of hepatic *Sec16b* mRNA levels in liver-specific *Hnf4 α* knockout mice (please see Fig. 5H) as independent evidence of the transcriptional regulation suggested by the reviewer.

In pathological conditions (Fig. 6K), we observed that SEC16B is upregulated without elevations in HNF4 α protein. This may suggest that HNF4 α activity could be modulated by multiple mechanism besides upregulation at the protein level (e.g., by post-translational modifications or translocation), or the activation of parallel, HNF4-independent pathways that control SEC16B at transcriptional or post-transcriptional levels. While we reason that the protein-level data is the more direct and functionally relevant endpoint for assessing the pathophysiological outcome, we have included the requested *Sec16b* mRNA data from the inhibitor experiment as requested (please see Fig EV5A in the revised manuscript), which confirms the transcriptional

downregulation. We have also revised the manuscript text to present this experiment with greater clarity (please see page 16, line 416-418) and thank the reviewer for the constructive suggestion.

Dear Xiao-Wei,

Thank you for incorporating the final editorial and formatting requests in the manuscript. I am now pleased to inform you that your manuscript has been accepted for publication. Congratulations with a nice study!

Before we forward your manuscript to our publishers, we would like to propose some edits in the manuscript title, abstract and synopsis. I have also written a short blurb that will accompany the title of your manuscript in our online system. Please take a look at the proposed text changes in the attached text file and let me know if any corrections are needed.

You may qualify for financial assistance for your publication charges - either via a Springer Nature fully open access agreement or an EMBO initiative. Check your eligibility: <https://link.springer.com/journal/44318/how-to-publish-with-us>

If you have any questions, please do not hesitate to contact the Editorial Office. Thank you for this interesting contribution to The EMBO Journal!

Best wishes,

Ieva

Please note that it is The EMBO Journal policy for the transcript of the editorial process (containing referee reports and your response letters) to be published as an online supplement to each paper. If you should prefer removal of any referee-only figures included in the point-by-point response(s), e.g. because they may still be used for future publication or because they have been reproduced from published work by others, please do let us know immediately via response email.

More information is available here: <https://link.springer.com/partners/embo-press/editorial-policies#Peer%20review>